

# Isotopic approaches to quantifying root water uptake and redistribution: a review and comparison of methods

Youri Rothfuss[1], Mathieu Javaux[1,2,3]

[1]Institute of Bio- and Geosciences, IBG-3 Agrosphere, Forschungszentrum Jülich GmbH, Jülich, 52425, Germany
[2]Earth and Life Institute, Environnemental Sciences, Université catholique de Louvain (UCL), Louvain-la-Neuve, 1348, Belgium
[3]Department of Land, Air and Water Resources, University of California Davis, Davis, California, 95616, USA.

*Correspondence to*: Youri Rothfuss (y.rothfuss@fz-juelich.de)

**Abstract.** Plant root water uptake (RWU) and release (i.e., hydraulic redistribution – HR, and its particular case hydraulic
lift – HL) have been documented for the past five decades from water stable isotopic analysis. By comparing the (hydrogen or oxygen) stable isotopic composition of plant xylem water to those of potential contributive water sources (e.g., water from different soil layers, groundwater, water from recent precipitation or from a nearby stream) authors could determine the relative contributions of these water sources to RWU. Other authors have confirmed the existence of HR and HL from the isotopic analysis of the plant xylem water following a labelling pulse.

In this paper, the different methods used for locating / quantifying relative contributions of water sources to RWU (i.e., graphical inference, statistical (e.g., Bayesian) multi-source linear mixing models) are reviewed with emphasis on their respective advantages and drawbacks. The graphical and statistical methods are tested against a physically based analytical RWU model during a series of virtual experiments differing in the depth of the groundwater table, the soil surface water status, and the plant transpiration rate value. The benchmarking of these methods illustrates the limitations of the graphical
and statistical methods (e.g., their inability to locate or quantify HR) while it underlines the performance of one Bayesian mixing model, but only when the number of considered water sources in the soil is the highest to closely reflect the vertical distribution of the soil water isotopic composition. The simplest two end-member mixing model is also successfully tested when all possible sources in the soil can be identified to define the two end-members and compute their isotopic compositions. Finally, future challenges in studying RWU with stable isotopic analysis are evocated with focus on new
isotopic monitoring methods and sampling strategies, and on the implementation of isotope transport in physically based RWU models.





**Keywords**

Root water uptake; hydraulic redistribution; hydraulic lift; water stable isotopologues; isotope mixing model; physically based root water uptake model



# 1 INTRODUCTION

Root water uptake (RWU) is defined as the amount of water abstracted by a root system from soil over a certain period of time. Understanding the relation between the distributions of soil water, roots, RWU location and magnitude, and root hydraulic properties is important for managing soil water and plant water status (e.g., by irrigation), developing new plant

genotypes more tolerant to drought or tackling ecological questions in water-limited ecosystems, such as the competition for soil water by different plants.

RWU is principally driven by evaporative flux taking place in the leaves (i.e., transpiration) and its magnitude depends on the atmospheric evaporative demand and stomatal opening. The latter depends amongst others on leaf water status and stress hormonal signals from the roots transported to the leaves (e.g., Huber et al., 2015; Tardieu and Davies, 1993). Leaf water

status and hormonal signals are related to the soil water potential distribution and to the plant hydraulic architecture (Huber et al., 2015). The spatial distribution of RWU is very variable in time in space, depends on the presence of roots but also on the root's ability to extract water. This ability is a function of radial conductivity but axial conductance may also limit water flow in younger roots or when cavitation occurs. The flux of water depends also on soil water availability: a highly conductive root segment will not be able to extract water from a dry soil. Locally, this is the difference of water potential

between the root and the soil which drives RWU, and its magnitude is controlled by the radial hydraulic resistances in the rhizosphere, at the soil root interface and in the root system (Steudle and Peterson, 1998). The actual RWU profile is thus a combination of different aspects: the root ability to extract water (characterized by the amount of roots and their hydraulic properties), the ability of the soil to fulfill the plant water demand, and the water potential difference between soil and root (Couvreur et al., 2014).

Plants have numerous mechanisms to cope with heterogeneous soil water distribution, and adapt their RWU rate distribution: adaptive root growth, adaptive root conductivity (Javaux et al., 2013), exudation (Carminati et al., 2016). A particular process, which has attracted the attention of plant breeders and ecologists is the ability of plants to extract water from non or less water limited soil areas with potentially low root lengths densities, known as root water uptake compensation (Heinen, 2014). To describe the RWU rate in soils, we will use the root water uptake flow per volume of soil, defined as $S$ [$L^3 L^{-3} T^{-1}$]

by reference to the sink term of the Richardson (1922) equation (Vereecken et al., 2016). According to Couvreur et al. (2012), root compensation is defined in the present article as the process that decreases or increases RWU at a certain location compared to the water uptake from that location when the soil water potential would be uniform in the root zone. Thus, the distribution of the $S(x,y,z)$ is a sum of two spatially distributed components:

$$S(x, y, z) = S_{uniH}(x, y, z) + S_{comp}(x, y, z) \qquad (1)$$

where $x$, $y$ and $z$ are the 3-D spatial coordinates, $S_{uniH}$ is a term proportional to the root distribution and $S_{comp}$ the compensatory part of the RWU distribution. The first term on the right-hand side of Eq. (1) is always positive while the second one can be either positive or negative. Figure 1 illustrates how this equation affects $S$ distribution in a one-dimensional (1D) space. When there is no compensation ($S_{comp}(x,y,z) = 0$), the RWU distribution follows the root distribution





(i.e., highest at the surface and lowest in the deepest layer, Fig. 1a). When $S_{comp}(x,y,z) < 0$ but its absolute value is lower than $S_{uniH}(x,y,z)$, then $S(x,y,z)$ is positive and different from the root vertical distribution. In case $S_{uniH}(x,y,z)$ is small, as in Fig. 1c, $S_{comp}(x,y,z)$ can locally be higher in absolute value and $S(x,y,z)$ can be locally negative which implies that there is a water efflux out of the root.

The water efflux at certain locations is called root hydraulic redistribution (HR, Burgess et al., 1998) or hydraulic lift (HL, Richards and Caldwell, 1987) as a specific case of HR in which fluxes in the root system are vertically upward. In their review, Neumann and Cardon (2012) discussed that the magnitude of HR observed in different studies varied from 0.03 mm d$^{-1}$ (brasilian Cerrado, Scholz et al., 2010) to 3.50 mm d$^{-1}$ (Artemisia Tridenta, Ryel et al., 2003). Several authors have also raised the question of the "ecohydrological interest" for a plant to release water to the upper/dryer soil layers, therefore

potentially providing water to shallow-rooted plants and enhancing competition for space and nutrients. Some studies suggested that HL could increase nutrient mobility and enhance biogeochemical processes by providing moisture to the dryer soil layers (Caldwell et al., 1998; Prieto et al., 2012; Snyder et al., 2008).

Despite its importance, there is a lack of measurements of RWU, related to the difficulty of measuring root and soil water fluxes. Often soil water content change is used as a proxy for RWU. Yet, as change of soil water content with time is not due

to root extraction only (i.e., soil water redistribution can also occur), the assessment of RWU based on water content distribution alone is not possible in conductive soils (Musters and Bouten, 2000). Rather, the full soil water flow equation accounting for root uptake and soil water redistribution must be solved in an inverse mode, and, with an accurate knowledge of soil and root properties RWU distribution can be inferred (Guderle and Hildebrandt, 2015; Hupet et al., 2002; Musters and Bouten, 1999; Vandoorne et al., 2012). Nuclear magnetic resonance (NMR) imaging has been suggested as an adequate

technique to measure water flow velocity in xylem vessels but no application exists yet on living roots in soils (Scheenen et al., 2000). More recently, Zarebanadkouki et al. (2012) could measure for the first time RWU in porous media by combining a tracer experiment monitored by neutron tomography with inverse modelling of a transport equation. Yet, this was done under controlled conditions while there is no standard method to monitor three dimensional water uptake distribution of growing roots *in situ*. In woody plants, in which roots are thick enough, Nadezhdina et al. (2010; 2012; 2015) used sap flow

measurements in roots to quantify hydraulic redistribution.

Since the seminal work of Zimmermann et al. (1967) which reported that RWU of *Tradescantia fluminensis* occurred in the absence of fractionation against water oxygen stable isotope, water stable isotopologues ($^{1}H^{2}H^{16}O$ and $^{1}H_{2}^{18}O$) have been frequently used to identify and quantify root water uptake and redistribution in soils through the measurements of their natural (and artificial) isotopic abundances. Methods include simple graphical inference to more sophisticated statistical

methods, i.e., two-end members and multi-source linear mixing models. While the former attempts to locate the "mean root water uptake" in the soil, the latter category of methods provides profiles of relative contributions to transpiration flux across a number of defined soil layers.





This present paper (i) aims at reviewing these methods and (ii) proposes to compare them against each-other during a series of virtual experiments differing in the water and isotopic statuses in the soil and the plant. Prior to the review and inter-comparison, the paper reports on the mechanisms at the origin of the spatiotemporal dynamics of natural isotopic abundances in soil and on the background knowledge of isotopic transfer of soil water to and from roots. Finally, we address future challenges to be undertaken such as the dynamic isotopic assessment of HR. We also evoke opportunities offered by novel isotopic monitoring tools which provide unpreceded high frequency isotopic measurements.

## 2 THEORETICAL BACKGROUNDS

The temporal and spatial variations in natural isotopic abundances observed within the soil-plant system allow for reconstruction of $S$ profiles. These variations result from isotope-specific fractionation between different phases at thermodynamic equilibrium and during non-equilibrium phase transition, i.e., when there is a net flux between different phases as for instance during evaporation. In this section, we briefly review process based analytical models accounting for isotopic fractionation that were first proposed for (i) free water (section §2.1) and (ii) later for matric-bound water in a bare soil (section §2.2). Finally, (iii) we report on the absence of isotopic fractionation during RWU for most of the documented plant species and on the simple mixing model which is at the basis of any isotopic study on RWU (section §2.3).

### 2.1 Isotopic effects during free water evaporation

In a closed liquid water–water vapor isothermal system at water vapor saturation (relative humidity = 100 %) or at thermodynamic equilibrium, the difference between the liquid and vapor (hydrogen or oxygen) isotopic compositions ($\delta_l$ and $\delta_v$ [-, expressed in ‰ relative to the Vienna-Standard Median Ocean Water international isotope reference scale], Gonfiantini, 1978) is a function of the system temperature solely and is named "equilibrium isotopic fractionation factors" ($\alpha_{eq}$ [-], for a complete list of symbols see Appendix A). Majoube (1971) and Horita and Wesolowski (1994), among other authors, gave empirical expressions (i.e., closed-form temperature dependent equations) for these equilibrium fractionation factors.

When the system is no longer closed and a difference in water vapor partial pressure exists between the air layer in direct contact with the liquid surface and the atmosphere above (referred to as "free" atmosphere), water vapor is transferred from the liquid phase to the air layer, i.e., evaporation ($E$) occurs. In analogy to an electrical circuit (i.e., a Rideal–Langmuir linear-resistance model, Brutsaert, 1982), $E$ can be calculated from the vapor pressure difference and a transfer resistance ($r$) to vapor transport across the air layer between the evaporating surface and the free atmosphere. Following the same electrical analogy, vapor transport of isotopologues ($E_i$) is a function of the difference of vapor isotopic composition between the air layer in contact with liquid water and the free atmosphere and a transfer resistance ($r_i$). In a simple yet comprehensive model Craig and Gordon (1965) divided the air layer into two consecutive layers with different aerodynamic conditions. In a first



sub-layer (with transfer resistances $r_{diff}$ and $r_{diff,i}$ for lighter and heavier isotopologues, respectively), vapor transfer is purely diffusive whereas in a second layer (with transfer resistances $r_{ad}$ and $r_{ad,i}$ for lighter and heavier isotopologues, respectively) it is purely advective, i.e., controlled by turbulence. Since the diffusion coefficient of the heavier isotopologues are smaller than that of $^1H_2^{16}O$, $r_i$ is greater than $r$. Note that the difference between $r_i$ and $r$ originates from the difference between $r_{diff}$

and $r_{diff,i}$ whereas $r_{ad} = r_{ad,i}$. In this model, thermodynamic equilibrium conditions still prevail in the air layer in contact with the evaporating surface, termed "liquid-vapor interface" of isotopic composition $\delta_{l\text{-}v}$. The isotopic composition of the evaporated water vapor ($\delta_E$), defined as the ratio $E_i/E$, depends on the ratio of the resistances $r_i/r$. The latter ratio, named "kinetic isotopic fractionation factor" ($\alpha_K$, [-]) depends on the relative importance (or development) of each sub-layer and contributes producing an evaporated vapor depleted in heavy isotopologues with respect to the vapor at the liquid-vapor

interface (i.e., $\delta_E < \delta_{l\text{-}v}$). In turn, and depending on the turnover of the system (ratio $E/V$ with $V$ being the volume of the evaporating liquid), but also on the evaporation state (i.e., permanent or transient), the liquid phase enriches itself in the heavy isotopologues. Finally, when both $E$ and $V$ are constant over time, meaning that the loss of water is compensated by a source of constant isotopic composition $\delta_{source}$, an "isotopic steady state" might be reached where, by mass balance, $\delta_E = \delta_{source}$. For a thorough review of the evaporation model of Craig and Gordon (1965), the reader is referred to Gat (1996) and

to the more recent paper by Horita et al. (2008).

In a two-dimensional ($\delta^{18}O$, $\delta^2H$) space, meteoric waters (e.g., precipitation, river water, groundwater) formed by equilibrium processes (i.e., condensation of water vapor) fall onto a line whose slope equals approximately eight and whose theoretical value is the ratio $(\alpha_{eq}^{2H} - 1)/(\alpha_{eq}^{18O} - 1)$ at the temperature of condensation. On the other hand, the water vapor produced during a non-equilibrium process, such as evaporation, fall onto a so-called "evaporation line" with a slope of generally

lower than six and greater than 2. This is explained by the fact that $\alpha_K^{18O} > \alpha_K^{2H}$ which leads to a greater depletion of $^1H_2^{18}O$ with respect to $^1H^2H^{16}O$ in the produced water vapor. Gat (1971) showed that the value of this slope was fairly approximated by the Craig and Gordon (1965) model, which was recently tested by Rothfuss et al. (2015).

## 2.2 Isotopic effects during bare soil evaporation and leaf transpiration

The Craig and Gordon (1965) model, originally developed for free evaporating water was later adapted to bound-to-matrix

soil water. In a study that laid the basis for future work in isotopic ecohydrology, Zimmerman et al (1967) provided a steady-state analytical solution for soil water isotopic composition ($\delta_S$) in a water-saturated isothermal bare sand profile from which water evaporated at a constant rate. Under these steady-state and isothermal conditions, the upward (convective) liquid flux of isotopologues, triggered by evaporation and rising from deeper layers equals the downward (diffusive) isotopic flux from the evaporating surface which is enriched in the heavy stable isotopologues due to evaporation. A profile is obtained (Fig.

2a, black line) whose exponential shape depends on boundary conditions, i.e., the source water (e.g., groundwater) and surface water isotopic compositions ($\delta_{source}$ and $\delta_{surf}$), the diffusion coefficient of the isotopologues in water, and of a soil "tortuosity factor", conceptually defined as the ratio of the geometrical to actual water transport distance. Barnes and Allison



(1983) extended this formulation to a non-saturated sand column evaporating at isotopic steady state ($\delta_E = \delta_{\text{source}}$). In this case, the evaporating surface (i.e., the liquid-vapor interface) can be located below the soil surface and splits the profile into two regions where isotopic transport predominantly occurs either in the vapor phase above or in the liquid phase below it. At isotopic steady state, the maximal isotopic enrichment is at the evaporation front ($\delta_{\text{EF}}$ at soil depth $z_{\text{EF}}$) and can be simulated

with the Craig and Gordon (1965) model. The isotopic composition of the soil residual adsorbed water in the "vapor region" above the evaporation front can be obtained by assuming thermodynamic equilibrium conditions and by applying Fick's law, and is shown to decrease linearly towards the value of the liquid water at the soil surface which is at thermodynamic equilibrium with the ambient atmospheric water vapor (Fig. 2a, gray line). Finally, note that at transient state ($\delta_E \neq \delta_{\text{source}}$), the maximal isotopic enrichment in the soil profile does not point to the location of the evaporation front as was

demonstrated by Rothfuss et al. (2015). Instead, the depth where the steepest gradient in the isotopic profile is observed corresponds to the evaporation front.

In a two-dimensional ($\delta^{18}$O, $\delta^2$H) space, liquid soil water sampled below the evaporation front will plot on an "evaporation line" with a slope typically lower than six and greater than two, depending on atmospheric and isotopic forcing, as a result of kinetic processes during evaporation. Above the evaporation front and at isotopic steady-state, soil liquid water is in

equilibrium with a mixture of atmospheric water vapor ($\delta^{18}$O-$\delta^2$H slope ~8) and evaporated soil water vapor rising from the evaporation front ($2 < \delta^{18}$O-$\delta^2$H slope $< 6$) (Barnes and Allison, 1988; Brunel et al., 1995; DePaolo et al., 2004). As a result, an intermediate value for the slope is expected, depending on the mixing ratio of atmospheric water vapor to evaporated soil vapor at a given soil depth.

### 2.3 Isotopic transfer to and from roots

As opposed to the removal of water vapor by evaporation, RWU has been described in a number of studies and over a wide variety of plant species not to be associated with (kinetic) isotopic fractionation (Bariac et al., 1994; Dawson and Ehleringer, 1993; Thorburn et al., 1993; Walker and Richardson, 1991; Washburn and Smith, 1934; White et al., 1985; Zimmermann et al., 1967). Consequently, for plants growing in homogeneous external conditions, e.g., in hydroponic solution, root xylem sap water and external water have the same isotopic compositions. In natural soils where the liquid phase is not

homogeneous and a vertical gradient of isotopic composition due to evaporation exists, the root system takes up water at different depths having thus different isotopic compositions.

Assuming that water transport time in roots is negligible, the isotopic concentration of the xylem sap water at the root tiller ($C_{\text{Ti}}$ [M L$^{-3}$]) can be modeled as the weighted average of the product of the soil water isotopic concentration ($C_{\text{S}}$ [M L$^{-3}$]) and $S$ ($x,y,z$):

$$C_{\text{Ti}} = \frac{\int\limits_{x,y,z} C_{\text{S}}(x,y,z) \cdot S(x,y,z) \cdot dx \cdot dy \cdot dz}{\int\limits_{x,y,z} S(x,y,z) \cdot dx \cdot dy \cdot dz} = \frac{\int\limits_{x,y,z} C_{\text{S}}(x,y,z) \cdot S(x,y,z) \cdot dx \cdot dy \cdot dz}{J_{\text{Ti}}} \tag{2}$$




with $J_{Ti}$ [L$^3$ T$^{-1}$] the xylem sap flux at the root tiller. Following Braud et al. (2005):

$$C = \rho \cdot R_{ref} \frac{M_i}{M_w}(\delta + 1) \tag{3}$$

with $\rho$ [M L$^{-3}$] the volumetric mass of water, $R_{ref}$ [-] the Vienna-Standard Mean Ocean Water (V-SMOW) hydrogen or oxygen isotopic ratio, $M_w$ and $M_i$ [M L$^{-3}$] the molar masses of $^1$H$_2$$^{16}$O and isotopologue ($^1$H$^2$H$^{16}$O or $^1$H$_2$$^{18}$O), respectively, the xylem sap water isotopic composition at the root tiller $\delta_{Ti}$ [-, expressed in ‰] can be expressed as:

$$\delta_{Ti} = \frac{\int\limits_{x,y,z} \delta_S(x,y,z) \cdot S(x,y,z).dx \cdot dy \cdot dz}{J_{Ti}} \tag{4a}$$

with $\delta_S(x,y,z)$ [-, expressed in ‰] the isotopic compositions of soil water at coordinates $(x,y,z)$. Mostly, a one dimensional description of root water uptake is used assuming that $\delta_S$ and RWU do not vary in the horizontal direction and $\delta_S$ is obtained for discrete soil layers of depths $z_j$ ($j \in$ [1,n]) and thickness $\Delta z_j = z_{j+1} - z_j$. It is usually further hypothesized that $J_{Ti}$ equals the transpiration flux $T$ [L$^3$ T$^{-1}$] (low to no plant capacitance or phloem-xylem contact):

$$\delta_{Ti} = \frac{\sum\limits_{j=1,n} \delta_S(z_j) \cdot S(z_j) \cdot \Delta z_j}{\sum\limits_{j=1,n} S(z_j) \cdot \Delta z_j} = \frac{\sum\limits_{j=1,n} \delta_S(z_j) \cdot S(z_j) \cdot \Delta z_j}{q_{Ti}} \tag{4b}$$

where $q_{Ti} = J_{Ti}/(\Delta x \Delta y) = T/(\Delta x \Delta y)$ represents the sap flow rate in the root tiller per unit surface area [L T$^{-1}$].

$\delta_{Ti}$ can be accessed at different locations in the plant depending on the species, but the sampling location should not be affected by evaporative enrichment in heavier isotopologues or back-diffusion of the isotopic excess accumulated at the sites of transpiration (stomatal chambers) in the leaf. For grasses and nonwoody plants, this is done by sampling the root crown (e.g., Leroux et al., 1995), the aerial nodal roots (e.g., Asbjornsen et al., 2007), the meristematic petiole, or else the collars (e.g., tillers) at the base of the plant (e.g., Dawson and Pate, 1996; Sánchez-Perez et al., 2008). In the case of ligneous plants the fully suberized stem (Asbjornsen et al., 2007) or sapwood (e.g., White et al., 1985) is sampled. On the other hand, $\delta_S$ is usually measured by sampling soil profiles destructively. Finally, water from plant and soil is predominantly extracted by cryogenic vacuum distillation (Araguás-Araguás et al., 1995; Ingraham and Shadel, 1992; Koeniger et al., 2011; Orlowski et al., 2013; West et al., 2006).

Lin and Sternberg (1992) and Ellsworth and Williams (2007), amongst other authors, reported however that for some xerophyte (plants adapted to arid environments, e.g., *Prosopis velutina* Woot.) and halophytes species (plants adapted to saline environments, e.g., *Conocarpus erecta* L.), and mangrove species (e.g., *Laguncularia racemosa* Gaert.), RWU led to fractionation of water hydrogen isotopologues. For mangrove species, it was hypothesized that the highly developed Casparian strip of the root endodermis would force water moving symplastically (i.e., inside the cells) and therefore crossing cell membranes (Ellsworth and Williams, 2007). Water aggregates are then dissociated into single molecules to move across these membranes. This demands more energy for $^1$H$^2$H$^{16}$O than for $^1$H$_2$$^{16}$O and $^1$H$_2$$^{18}$O, thus preferentially affects $^1$H$^2$H$^{16}$O



tranport and leads to a situation where xylem sap water is depleted in this isotopologue with respect to source water. Meanwhile, this affects to a much lesser extent $^1H_2^{18}O$ transport, so that no detectable isotopic fractionation of water oxygen isotopologues is observed. It can be concluded that, for the majority of the studied plant species, either RWU does not lead to isotopic fractionation or its magnitude is too low to be observable.

Hydraulic redistribution (e.g., hydraulic lift) can be conceptualized as a reverse RWU, defined as a negative $S$. In such case, Eq. (4b) should only account for the positive $S$. It can be done by assuming that in this equation $\delta_S$ is 0 when $S < 0$.

Finally, plant water samples will, similarly to soil water samples, also fall onto an "evaporation line" of a slope lower than eight in a two-dimensional ($\delta^{18}O$, $\delta^2H$) space (Javaux et al., 2016).

## 3 METHODS FOR CHARACTERIZING RWU FROM STABLE ISOTOPIC ANALYSIS

We distinguish two classes of methods: (i) the graphical method for inferring the "mean root water uptake depth" ($\bar{z}$ [L]) (§3.1), and (ii) statistical methods based on end-member mixing analysis (EMMA) (Barthold et al., 2011; Christophersen and Hooper, 1992) for identifying $x_j$ [-], the contribution to RWU of some plant water source $j$ (e.g., water in some soil layer, groundwater, water from recent precipitation, or else from a nearby stream, §3.2). All methods have in common to use an inverse modeling approach: the RWU distribution is obtained by optimizing model input parameters until the simulated $\delta_{Ti}$

and/or the simulated soil isotopic profiles fit to the isotopic measurements.

Table 1 summarizes the 21 isotopic studies reviewed in this paper that use either one of the two classes of methods.

### 3.1 Graphical inference (GI)

This straightforward approach defines the "mean root water uptake depth" $\bar{z}$, as the depth where $\delta_S = \delta_{Ti}$. $\bar{z}$ conceptually indicates to the soil depth where the plant root system, represented as one unique root, would extract water from.

There are cases where $\bar{z}$ cannot be unambiguously identified (e.g., $\bar{z}_1$ and $\bar{z}_2$ of case 2, Fig. 2b) due to the non-monotonic character of the $\delta_S$ profile (shown in black dashed line, case 2 of Fig. 2b). In order to define a mean RWU depth for such a case one can derive a monotonously decreasing $\delta_S$ profile by smoothing the profile (shown as symbols in Fig. 2b), e.g., by averaging $\delta_S$ in a number of layers using the following mass balance:

$$\delta_{S,J} = \frac{\sum_{j \leq J} \delta_S(z_j) \cdot \theta(z_j) \cdot \Delta z_j}{\sum_{j \leq J} \theta(z_j) \cdot \Delta z_j} \tag{5}$$

where $J$ represents the set of depths that belong to the $J^{th}$ soil layer, with $\theta$ [L$^3$ L$^{-3}$] and $\Delta z_j$ [L] the soil volumetric water content and thickness of the soil layer centered around depth $z_j$. Due to this smoothing, the vertical resolution may be drastically reduced. In the example presented Fig. 2b where a uniform $\theta$ profile is assumed, the $\delta_{S,J}$ profile intersects with the vertical line of value $\delta_{Ti}$ deeper than for the initially non-monotonic $\delta_S$ profile, i.e., $\bar{z}$ (case 2, integrated $\delta_S$ profile) $< \bar{z}_2 < \bar{z}_1$.



Some authors rule out solutions in case of multiple mean root water uptake depths, e.g., by excluding the $\bar{z}$ solutions where soil water content was low and/or soil water potential was high in absolute value (e.g., Li et al., 2007; see Table 1).

Note that while Eq. (5) provides a representative value for the isotopic composition that would be measured in soil layer $J$ as a function of those of the water in the set of depths, $\delta_{S,J}$ is however equivalent to the isotopic composition "sensed by the plant" only if the root profile is homogeneous, i.e., when $RLD$ is constant with depth in that particular soil layer J.

The method of graphical inference may not only provide $\bar{z}$ but also its uncertainty caused by the uncertainty in measuring $\delta_{Ti}$ (e.g., based on the precision of the isotopic analysis and/or sampling natural variability, shown as gray stripe in Fig. 2b). The steeper the soil water isotopic profile, the larger is the uncertainty in determining $\bar{z}$ is. Figure 2b illustrates this with estimated minimum and maximum $\bar{z}$ for the monotonic $\delta_S$ profile and for the vertically averaged profile. In the latter case, the possible range of $\bar{z}$ is the largest. These ranges give first quantitative indication of variance around $\bar{z}$. Finally, for a complete "graphical assessment" of the variance of $\bar{z}$, one should also consider the uncertainty associated with measurements of the $\delta_S$ profile (not shown).

### 3.2 Statistical approaches

### 3.2.1 Two end-member (TM) mixing model

The TM method is a particular case of end-member mixing analysis (EMMA) and is based on the concept that (i) a plant extracts water from two predominant water sources A and B (e.g., water in distinct upper and lower soil layers, or groundwater and recent precipitation water etc.) in given proportions, (ii) there is no isotopic fractionation during water uptake, and (iii) there is a complete mixing inside the plant of the contributing water sources A and B to RWU. The mass conservation for isotopologues gives:

$$J_{Ti}^i = J_i^A + J_i^B \tag{6a}$$

$$C_{Ti} \cdot J_{Ti} = C_A \cdot J_A + C_B \cdot J_B \tag{6b}$$

with $J_A$, $J_B$, and $J_{Ti,}$ [L$^3$ T$^{-1}$] (respectively $J_i^A$, $J_i^B$, and $J_{Ti}^i$ [M T$^{-1}$]) the fluxes of water (respectively isotopologues) originating from water sources A and B, and at the plant tiller. $C_A$, $C_B$, and $C_{Ti}$ [M L$^{-3}$] are the water sources A and B, and xylem sap water measured isotopic concentrations. By introducing $x = J_A / J_{Ti}$ and following Eq. (3), Eq. (6b) becomes:

$$\delta_{Ti} = x \cdot \delta_A + (1-x) \cdot \delta_B \tag{7}$$

In this approach, $\delta_{Ti}$ is therefore defined as the mean value of the isotopic compositions of water sources A and B ($\delta_A$ and $\delta_B$) weighted by the proportions to $J_{Ti}$ of water volume extracted by the plant from water sources A and B, i.e., $x$ and $(1 - x)$, respectively. The error associated with the estimation of $x$ ($\sigma_x$ [-, expressed in ‰]) can be calculated following Phillips and Gregg (2001):





$$\sigma_x^2 = \left(\frac{\partial x}{\partial(\delta_A)}\right)^2 \cdot \sigma_{\delta_A}^2 + \left(\frac{\partial x}{\partial(\delta_B)}\right)^2 \cdot \sigma_{\delta_B}^2 + \left(\frac{\partial x}{\partial(\delta_{xyl})}\right)^2 \cdot \sigma_{\delta_{Ti}}^2 \tag{8a}$$

$$\sigma_x = \sqrt{\sigma_x^2} = \frac{1}{(\delta_A - \delta_B)}\sqrt{\left(\sigma_{\delta_{Ti}}^2 + x^2 \cdot \sigma_{\delta_A}^2 + (1-x)^2 \cdot \sigma_{\delta_B}^2\right)} \tag{8b}$$

with $\sigma_{\delta_A}$, $\sigma_{\delta_B}$, and $\sigma_{\delta_{Ti}}$ the standard errors associated with the measurements of $\delta_A$, $\delta_B$, and $\delta_{Ti}$, respectively. The sensitivity of Eq. (8b) to different values of $\sigma_{\delta_A}$, $\sigma_{\delta_B}$, and $\sigma_{\delta_{Ti}}$ can be tested by considering either minimal possible errors,

i.e., the analytical precision of the isotopic analyser (e.g., isotope ratio mass spectrometer, laser-based spectrometer), or by taking into account additional errors involved with sampling procedure and vacuum distillation technique (see e.g., Rothfuss et al., 2010). Equation (8b) also shows that, independently of the values considered for $\sigma_{\delta_A}$, $\sigma_{\delta_B}$, or $\sigma_{\delta_{Ti}}$, $\sigma_x$ is inversely proportional to $1/(\delta_A - \delta_B)$, indicating that the two end-members should have as much as possible distinct isotopic compositions for a low standard error of $x$. Therefore, it is especially important, e.g., for partitioning between water from an

upper and lower portion of the soil profile, to properly define the thickness of these layers, so that they have distinct isotopic compositions, and that the difference is considerably larger than the precision of the isotopic measurements. Figure 3 shows for example that when (i) $x$ is evaluated at 10 % and (ii) $\sigma_{\delta_A}$, $\sigma_{\delta_B}$, and $\sigma_{\delta_{Ti}}$ are estimated being equal to 0.02 ‰ (dark blue solid line), $(\delta_A - \delta_B)$ should be greater than 0.75 ‰ (in absolute term) in order to reach a $\sigma_x$ value lower than 5 %, i.e., more than 37 times the error made on $\delta_A$, $\delta_B$, and $\delta_{Ti}$. To obtain the same standard error for $x$ in case of a higher standard error on

the estimation of $\delta_A$, $\delta_B$, and $\delta_{Ti}$ (e.g., $\sigma_{\delta_A}$, $\sigma_{\delta_B}$, and $\sigma_{\delta_{Ti}} = 0.1‰$), $(\delta_A - \delta_B)$ should be greater than 3.00 ‰ (in absolute term). This difference becomes much greater for $\sigma_{\delta_A}$, $\sigma_{\delta_B}$, and $\sigma_{\delta_{Ti}} = 1.00‰$ and reaches 42 ‰ (not shown in Figure 3). This certainly highlights the advantage of artificially labelling soil water with water enriched (or depleted) in heavy isotopologues for a more precise assessment of the relative contribution of soil water sources to RWU, as mentioned by Moreira et al. (2000). In another study, Bachmann et al. (2015) labeled the upper and lower portion of the soil profile in a natural temperate

grassland with $^{18}$O-enriched and $^2$H-enriched water, respectively. They defined two distinct (upper and lower) soil water sources, for which they calculated the corresponding $\delta^2$H or $\delta^{18}$O on the basis of measured soil water isotopic profiles and using Eq. (5). They could find evidence against the so-called hypothesis of "niche complementarity" regarding plant water use, which states that RWU of competitive plant species is spatially and temporally distinct, and that this distinction is stronger at high species richness. Figure 3 illustrates also that for given $(\delta_A - \delta_B)$, $\sigma_{\delta_A}$, $\sigma_{\delta_B}$, and $\sigma_{\delta_{Ti}}$ values, the "optimal $x$

value" for a low $\sigma_x$ is 50% (showed by the orange lines).

Table 1 displays a sample of studies that used the two end-member mixing approach. Authors could distinguish between uptake of irrigation and precipitation water (Goebel et al., 2015), precipitation and groundwater (White et al., 1985), soil water and groundwater (McCole and Stern, 2007), or else between stream water and soil water (Dawson and Ehleringer, 1991; McDonnell, 2014). Thorburn and Ehleringer (1995) could for instance locate the dominant source for RWU, i.e.,





groundwater for their mountain and floodplain test-site and water from the soil between 0.3 and 0.4 meters depth for their cold desert test-site. Other authors (e.g., Brunel et al., 1995) combined two mixing equations, i.e., one for each isotopologue, into a single one. As infrared laser-based spectrometry now enables simultaneous measurements of $\delta^{18}$O and $\delta^{2}$H at lower cost, we believe that this dual-isotope approach (referred as "D" in Table 1) will or should gain in importance in isotopic

studies, especially in the context of pulse labelling experiments, which can "disconnect" the strong correlation between soil water $\delta^{18}$O and $\delta^{2}$H, therefore provide two independent mixing equations, one for each isotopologue.

### 3.2.2 Multi-source (MS) mixing models

When there are more than two identified plant water sources contributing to RWU, e.g., water from different layers $j$ ($j \in [1, N]$) in soil the profile, Eq. (7) becomes:

$$\delta_{Ti} = \sum_{j=1}^{N} x_j \cdot \delta_{S,j} \tag{9}$$

with N the number of plant water sources (e.g., soil layers) and $\sum_{j=1}^{N} x_j = 1$. As there are more water sources than (number of

mixing equations + 1), there is not a unique solution but an infinite range of possible solutions. However, some of these solutions are not likely or possible based on background information or knowledge. A range of solutions that is most likely based on prior information can be obtained using Bayesian methods. In the method proposed by Phillips and Gregg (2003),

the isotopic composition calculated for each considered $x_j$ combination ($\delta_{Ti}$) is compared with the measured value ($\delta_{Ti,m}$). The number of combinations depends on the value of contribution increment ($i$, %, typically 5 or 10 %) and the combinations for which $\delta_{Ti}$ meets the following requirement are selected:

$$\delta_{Ti} \leq \left| \delta_{Ti,m} \pm \tau \right| \tag{10}$$

where $\tau$ [-, expressed in ‰], standing for "tolerance", usually accounts for precision of the isotopic measurements or possible

errors during sampling and vacuum distillation steps. This multi-source mixing model approach strongly depends on $\tau$ and $i$, which therefore should be carefully chosen by the user. A smaller $i$ also refines the analysis. For this, the program "IsoSource" (https://www.epa.gov/sites/production/files/2015-11/isosourcev1_3_1.zip) is available (Phillips et al., 2005). Wang et al. (2010) compared the outcome of the GI and MS approaches and came to the conclusion that even though the latter did not solve the non-uniqueness problem and provided diffuse patterns of frequency that were difficult to interpret in

some cases (e.g., in case of a non-monotonic isotopic profile), it had the advantage over the former method of providing a systematic and quantitative assessment of ranges of relative contributions.

Parnell et al. (2010) proposed to overcome two limitations of the approach of Phillips and Gregg (2003), i.e., its inability to (i) account for uncertainty in the estimations of $\delta_{Ti}$ and of the water sources isotopic compositions $\delta_{S,j}$, and (ii) provide a optimal solution rather than ranges of feasible solutions. For this they use a Bayesian framework (for details see also Erhardt




and Bedrick, 2013; Moore and Semmens, 2008; Parnell et al., 2013), which allows uncertainty in the $x_j$ proportions and incorporates a residual error term $\varepsilon_j$ (normally distributed with mean equal to zero and variance $\sigma^2$):

$$\delta_{\text{Ti}} = \sum_{j=1}^{N} x_j \cdot \delta_{\text{S},j} + \varepsilon_j \qquad (9')$$

Note that the terms of (i) trophic enrichment factor (TEF [-, expressed in ‰], see, e.g., meta-analysis of Vanderklift and Ponsard, 2003) and (ii) isotope concentration dependency (Koch and Phillips, 2002; Phillips and Koch, 2002) originally incorporated in the formulation of Parnell et al. (2010) for other applications are not present in Eq. (9') since (i) no isotopic fractionation during RWU is assumed and (ii) isotope concentration dependency applies only for situations where isotopic compositions of different elements are measured and available.

Parnell et al. (2010) developed the program "Stable Isotope Analysis in R" (SIAR, https://cran.r-project.org/src/contrib/siar_4.2.tar.gz) in which the initial (*a priori*) $x_j$ distribution is by default the Dirichlet distribution, of which information can be partly specified by the user. *A posteriori* $x_j$ distribution is obtained by fitting the linear model to data via a Metropolis-Hasting (Hastings, 1970; Metropolis et al., 1953) Markov Chain Monte Carlo algorithm.

Prechsl et al. (2015) apply both graphical and Bayesian approaches to evaluate the shift in $\bar{z}$ and change of RWU profile following drought treatments (approx. 20 to 40 % precipitation reduction with transparent rainout shelters) in both extensively and intensively managed grasslands. From both approaches it appeared that a shift in $\bar{z}$ was inexistent or not observable from isotopic analyses. Another recent application of the Bayesian approach was performed by Volkmann et al. (2016b), who took advantage of a newly developed soil isotopic monitoring method to confront high frequency $\delta_{\text{S}}$ profiles time series to time series of $\delta_{\text{Ti}}$ (indirectly obtained from the isotopic measurement of the transpired water vapor and assuming isotopic steady state, i.e., $\delta_{\text{Ti}} = \delta_T$) following a labelling pulse (see Table 1 for details on the study).

## 4 INTER-COMPARISON OF METHODS

We tested and compared the different methods (GI, TM, MS) during a series of virtual experiments. Mean RWU depths (provided by the GI method) and $x_j$ distribution (provided by the TM and MS methods) were determined from soil and xylem water oxygen isotopic composition distributions. While the former information was prescribed to the different methods, the latter was calculated with the physically based analytical RWU model (referred to as AM) of Couvreur et al. (2012) (see Appendix B1 for a description of the model and Appendix B2 on how it was run for the inter-comparison).

### 4.1 Methodology

### 4.1.1 Scenario definition

We developed eight virtual plausible scenarios of soil-plant systems under different environmental conditions. Each environmental condition was defined as a combination of different total soil water potential distributions (resulting from the



location of the groundwater table and weather conditions), soil water oxygen isotopic composition profiles, and actual transpiration rate. The groundwater table was either shallow at -1.25 m depth (prefix "Sh") or deep at -6 m depth (prefix "De"); the soil water potential was considered to be at static equilibrium below the groundwater level; the soil surface was either dry under evaporative conditions (suffix "Dr"), or wet, e.g., shortly after a rain event (suffix "We"); the transpiration

rate was either low (e.g., relevant at night, $T = 0.01$ mm h$^{-1}$, suffix "_lT") or high ($T = 0.30$ mm h$^{-1}$, suffix "_hT"). They all relied on a common measured root length density vertical distribution of *Festuca arundinacea*. Table 2 reports the input data. Note that, as hypothetized in Eq. (4b), transpiration and sap flow rates (i.e., per unit of surface area [L T$^{-1}$]) were considered as equal.

### 4.1.2 Setup of the models

The two end-member mixing approach (TM) was tested against the isotopic data for two different cases: (i) two conjoint soil layers spreading from $0 – 0.225$ m and $0.225 – 2.00$ m and (ii) two disjoint soil layers spreading from $0 – 0.225$ m and $1.75 – 2.00$ m. The latter case was designed to evaluate the impact of lacunar soil isotopic information on the calculation of *x*, i.e., when not all potential water sources are properly identified. Representative values of water oxygen isotopic compositions for these soil layers ($\delta_{S,J}$, $J \in$ [I,II]) were obtained from the mass balance (Eq. (5)) after interpolation of the measured soil water

content and $\delta_S$ profiles at a 0.01 m vertical resolution.

For the multi-source mixing approaches of Phillips and Gregg (2003) (MSPG) and Parnell et al. (2013) (MSPa), the number of potential water sources was initially fixed to three, i.e., water from the soil layers I (0.000-0.050 m), II (0.050-0.225 m), and III (0.225-2.000 m). Upper and lower boundaries of these layers were defined to reflect the exponentially shaped (monotonic) $\delta_S$ profiles (experiments ShDr and DeDr) or to smooth the non-monotonic $\delta_S$ profiles observed during

experiments ShWe and DeWe. MSPG and MSPa were also tested for eight soil layers (i.e., as many layers as measurement points, I: 0.000-0.020, II: 0.020-0.050, III: 0.050-0.110, IV: 0.110-0.225, V: 0.225-0.400; VI: 0.400-0.750, VII: 0.750-1.500, and VIII: 1.500-2.000 m). Increment and tolerance of the MSPG method were fixed at 10 % and 0.25 ‰, respectively. Similarly to the TM approach, profiles of $\delta_{S,J}$ ($J \in$ [I,III] or [I,VIII]) were obtained from the mass balance (Eq. (5)) after interpolation of the measured soil water content and $\delta_S$ profiles at a 0.01 m vertical resolution.

Finally for the MSPa method, uncertainty associated with $\delta_S$ measurements was set to 0.2 ‰ and the function *siarmcmcdirichletv4* of the SIAR R-package was run 500000 times (of which 50000 runs where discarded).

For a detailed description of the inter-comparison methodology, refer to Appendix C.

### 4.2 Results and discussion

Figure 4 displays $x_{AM}$, the simulated $\dfrac{S_{AM}(z)dz}{T/(\Delta x \cdot \Delta y)}$ ratios (solid colored lines) simulated by the analytical model of Couvreur

et al. (2012) for the eight scenarios together with uncertainty (shaded areas) and the corresponding $\delta_{Ti\_AM}$ (±1sd) (for a





description on how uncertainty was assessed, refer to Appendix C). In general, at high $T$ the compensation was negligible and the $S_{AM}$ profile was mainly proportional to the $RLD$ profile (Fig. 4b, d, f, and h). The only exception was a soil with deep groundwater table and dry surface, where this dry layer limited root water uptake (DeDr_hT). At lower transpiration demand, the $S$ profile predicted by the Couvreur et al. (2012) model generally differed from the $RLD$ profile (Fig.4a, c , e,

and g) due to the fact that the second term of Eq. (1) (i.e., $S_{comp}$, see also Eq. (B4) and (B4') in Appendix B) was proportionally larger. Water uptake from the upper layer was always more than proportional to the $RLD$, when this layer was wetter, and vice versa. Water release to the soil (i.e., HR) was observed only for the soil with the deep groundwater table and dry upper layer (DeDr_hT, Fig. 4e). From the graphical method GI, either a single or two distinct solutions for $\bar{z}$ (displayed as gray-shaded horizontal stripes) could be retrieved, depending on the monotonic/non-monotonic character of the $\delta_S$ profile,

and ranged between –0.02 and –0.95 m.

Figure 5a displays the relative contribution to $T$ of the upmost layer 0 – 0.225 m in case of two conjoint soil layers as computed with the TM approach and a comparison with the results of the analytical model. Except for the very last two virtual experiments (i.e., DeWe_lT and DeWe_hT), there was a very good agreement between TM and AM results: absolute difference between $x_{TM}$ and $x_{AM}$ ranged between 1.5 % (ShDr_lT) and 6.3 % (ShDr_hT). During experiment DeDr_lT, The

TM approach estimated that $x$ was equal to 12.3 % while the analytical model simulated hydraulic redistribution, i.e., excluded the layer 0 – 0.225 m as potential source. The significant difference between TM and AM results during experiments DeWe_lT and DeWe_hT and the higher standard error associated with $x_{TM}$ ($\sigma_x$, displayed in the form of error bars in Figure 5) were due to the small difference between the isotopic compositions of the defined soil water sources $\delta_{S,I}$ (– 6.0 ‰) and $\delta_{S,II}$ (–5.3 ‰) as illustrated in section §3.2.1. Figure 5b gives the relative contribution to $T$ of the layer 1.75 –2.00

m in case of two disjoint soil layers, i.e., when not all potential water sources are accounted for into the calculation of $\delta_{S,I}$ and $\delta_{S,II}$. In this case there were important disparities between $x_{TM}$ and $x_{AM}$. The mean absolute difference between these two estimates was equal to 43,5 (±17.8) %. Omitting some of the potential water sources contributing to $T$ had in this second case the consequence of artificially overestimate the contribution of the lowest layer. We therefore suggest to always attempting to fully characterize the soil isotopic profile before aggregating the isotopic information when defining the two water

sources.

Figure 6 gives the relative contributions from soil layers I, II, and III (upper, middle, and lower panel, respectively) to $T$ following the method of Phillips and Gregg (2003) ($x_{MSPG}$, in %, displayed in the form of gray histograms) and the Bayesian method of Parnell et al. (2010) ($x_{MSPa}$, in %, gray probability density curves). The colored vertical lines are $x_{I\_AM}$, $x_{II\_AM}$, and $x_{III\_AM}$, the simulated $\dfrac{S_{AM}(z)dz}{T/(\Delta x \cdot \Delta y)}$ ratios from layers I, II, and III. The color-shaded areas associated with $x_{I\_AM}$, $x_{II\_AM}$, and

$x_{III\_AM}$ refer to their uncertainty by accounting for the uncertainty of the input data. As for Fig. 5, $\delta_{Ti\_AM}$ is reported above each plot along with its standard deviation. $x_{J\_MSPG}$ probability distribution was observed to be either narrow (e.g., DeDr_lT / layer I, Fig. 5m) or broad (e.g., DeWe_hT / layer I), i.e., the range of the possible solutions for $x_{J\_MSPG}$ was relatively small



or large (10 and 100 % respectively for these two examples). In general, both MSPG and MSPa statistical methods agreed well with each other: the $x_{\mathrm{MSPa}}$ most frequent value (MFV, at the peak of the density distribution curve) was in most case either located near the median value of the $x_{\mathrm{MSPG}}$ probability range (e.g., ShWe_lT / layer I, Fig. 5g) or matched exactly the $x_{\mathrm{MSPG}}$ unique value (i.e., DeDr_lT / layer I, Fig. 5m). In contrast, the statistical methods succeeded best in providing $x$

estimates similar to those of the model of Couvreur et al. (2012) in case of a shallow groundwater table and at low $T$ only (Fig. 5a-c and g-i), thus when water availability was high and root compensation was low. In these cases, $x_{I\_AM}$ was included in the estimated $x_{I\_MSPG}$ range and the mean absolute difference (MD) between $x_{J\_AM}$ and $x_{\mathrm{MSPa}}$ MFV was equal to 8.6 %. This difference was the greatest (129.2 %) for experiment DeDr_lT when HR was simulated by the analytical model (Fig. 5m-o).

Considering eight soil layers instead of three added uncertainty in the assessment of their relative contribution to $T$ as determined by the MSPG method: the estimated probability ranges increased in most of the cases (results not shown). However it considerably improved the results of the MSPa method: the mean absolute difference between $x_{J\_AM}$ and the most frequent $x_{\mathrm{MSPa}}$ value was equal to 4.7 % for the scenarios with a shallow groundwater table and low transpiration rate and equal to 52.1 % in case of HR (Table 3).

Independent of the number of defined soil layers, lowering the value of increment to 5 % in the MSPG method refined the analysis where the probability distribution was already narrow (i.e., in the case of a well identified $x_{\mathrm{MSPG}}$ value, e.g., Fig. 5m) while it produced distributions that were flatter and contained less gaps when no clear solutions had emerged before (results not shown). Artificially increasing the value of tolerance had the consequence that more solutions to Eq. (10) were found for each experiment / transpiration value / layer combinations and *vice versa* (results not shown). An increase or decrease of a

factor 2 of the number of runs as well as the number of runs to be discarded from the analysis had only a marginal impact on the density distribution curves obtained with the MSPa method in the case of three or four soil layers.

The modelling exercise illustrated the disparities of outcome between the graphical method on the one hand and the statistical and mechanistic methods on the other: there simply cannot be a single or multiple "root water uptake depths" but rather a continuous RWU profile (AM) or statistical solutions of contribution to transpiration (MSPG and MSPa). Significant

changes of $\delta_{Ti}$ do not necessarily mean important changes in the depth of RWU but rather slight (nevertheless significant) modification of the RWU profile. The authors believe that the relatively novel statistical tools MSPG and MSPa presented in this review should be therefore preferred over the GI method, especially since the two former are available as user-friendly programs and packages and do not require significant computing time, therefore can be run locally on a personal computer. As highlighted in this series of virtual experiments, the Bayesian method showed much more convincing results than the

method of Phillips and Gregg (2003), especially in the case of eight soil layers, illustrating the interest of reaching the best vertical resolution and maximizing the number of identified potential sources.

One can also show from this inter-comparison of methods that labelling of soil water in either $^{18}$O or $^{2}$H has potentials for improving the different methods presented here theoretically if water is taken up by the roots from the labeled region





predominantly. However this was never the case looking at the results of the analytical model. A dual isotope ($^{18}$O or $^2$H) labelling pulse experiment that would artificially disconnect the strong link between $\delta^{18}$O and $\delta^2$H would on the other hand much more constrain the inverse problem and provide accurate estimate of contribution of $S$ to transpiration flux.

## 5 CHALLENGES AND PROGRESSES

### 5.1 Isotopic assessment of hydraulic redistribution

HR (e.g., HL) has been observed using isotopic measurements in a number of studies (e.g., Caldwell and Richards, 1989; Dawson, 1993; Kurz-Besson et al., 2006). However, in contrast to nondestructive "traditional" methods allowing for direct monitoring of redistribution dynamics (i.e., psychrometry, time domain reflectometry, and frequency domain capacitance, Brooks et al., 2002; Dawson, 1996; Richards and Caldwell, 1987; Wan et al., 2000), isotopic methods provide a destructive and indirect assessment. These methods are based on (i) labelling of soil or roots of deep-rooted plants at a given depth in the soil or at a certain location in the experimental field and (ii) measuring the $\delta_{Ti}$ of plants not having access to labeled water (i.e., of which the roots do not reach the isotopic labeled depth or location). When HR occurs, the xylem sap water (of measured isotopic composition $\delta_{Ti,m}$) of these plants can be conceptualized as a mixture of antecedent soil water (at natural isotopic abundance) and isotopically enriched water released to the soil by the deep-rooted vegetation. From simple mass balance at the release location, $\delta_S$ at a given depth $z$ in the soil and at time $(t+\Delta t)$ deviates from that at time $t$ as a function of the (negative) $S$ (i.e., HR or HL) at time $t$ and change of soil volumetric water content ($\theta$):

$$\delta_S(z, t+\Delta t) \cdot \theta(z, t+\Delta t) - \delta_S(z, t) \cdot \theta(z, t) = \delta_{Ti,m} \cdot V_{HR}(t)/V(z) = \delta_{Ti,m} \cdot |S(z,t)| \cdot \Delta t \qquad (11)$$

If $\theta$ and $\delta_S$ at times $t$ and $t+\Delta t$, and $\delta_{Ti}$ are measured, the water volume transported by the roots ($V_{HR}$, L$^3$) can be calculated knowing the volume of soil representative of the hydric and isotopic measurements ($V$, L$^3$). Note that HR is observable at a certain soil depth if and only if uptake and release locations in the soil have distinct water isotopic compositions. Finally, the obtained volume can be compared with the water volume transpired by the vegetation on the following day.

To the authors' knowledge, no precise observation (other than the study of Zegada-Lizarazu and Iijima, 2004) of change of soil water isotopic composition has been attributed with certainty to hydraulic redistribution and simultaneously provided amount of water involved in the process. Such observations however should be feasible under controlled experimental conditions where (i) the initial soil water isotopic profile before labelling is known and (ii) natural isotopic changes (due to, e.g., soil redistribution and moisture input from a precipitation event) can be avoided, and (iii) the lateral heterogeneity of soil water and isotopic composition profiles can be minimized (see for instance the setups of Armas et al., 2012; Querejeta et al., 2012). As highlighted in section 2.3, HR can be conceptualized as a negative $S$ (Eq. (4b)) and should therefore be exempt of isotopic fractionation. However, to the authors' knowledge this point has not yet been proven experimentally.





## 5.2 High frequency isotopic data and sampling strategies

For determination of $\delta_S$, soil profiles are usually destructively sampled, typically with an auger down to a depth of a few centimeters (Rothfuss et al., 2010) to a few meters (Moreira et al., 2000) (see Table 1), depending on the depths of the root system and of the water table. The sampling depth interval should, when possible, match the exponential decrease of isotopic

composition (Wang et al., 2010) due to fractionating evaporation and it should capture sudden variations with time at the soil surface due to precipitation, i.e., be minimal at the surface and maximal deeper in the soil profile where isotopic dynamics are less pronounced. A minimal sampling interval at the surface is also crucial as it provides the isotopic composition of the layer contributing the most to transpiration in the case of a low $T$ flux (e.g., morning transpiration) under non-limiting water availability. Not measuring this maximum soil isotopic composition (between precipitation event) can lead to a situation

where source partitioning is not feasible from isotopic measurements. Under field conditions (i.e., ~95 % of the studies reviewed in this work, summarized in Table 1) soil material is generally not a limiting factor, thus can be sampled twice or thrice to average out or characterize lateral heterogeneity without significant disturbance of the soil (Leroux et al., 1995).

Water from plant and soil materials is predominantly extracted by cryogenic vacuum distillation (Araguás-Araguás et al., 1995; Ingraham and Shadel, 1992; Koeniger et al., 2011; Orlowski et al., 2013; West et al., 2006). This consists in (i)

introducing the plant or soil sample into an extraction flask attached to one end of the extraction line, while at the other end a collection tube is connected, (ii) freezing the sample by immersing the collection flask into liquid nitrogen (temperature ~ − 200°C), (iii) pumping the extraction line down to a pressure of ~$10^{-3}$ mbar, (iv) heating the sample to a certain temperature ( ~60 < T < ~100°C) depending on its nature while immersing the trap into liquid nitrogen. The water vapor produced condenses in the trap following a stepwise procedure (~lasting one to a few hours), in order to avoid condensation elsewhere

on the water vapor path between sample and collection trap. Accuracy of this extraction method was shown to be maximal at higher water content and for sandy soils and lower for soils with high clay content. In the latter case, extraction times should be longer and temperatures higher to mobilize water strongly bound to clay particles, which has a distinct isotopic composition from that of pore "bulk" water (Araguás-Araguás et al., 1995; Ingraham and Shadel, 1992; Oerter et al., 2014; Sofer and Gat, 1972). In other studies, plant and soil waters are extracted following azeotropic distillation with kerosene as

solvent (e.g., Brunel et al., 1995; Thorburn and Ehleringer, 1995), or direct equilibration with $CO_2$ (Asbjornsen et al., 2007) following the method of Scrimgeour (1995), or else the mild vacuum method (Dawson and Pate, 1996; Jeschke and Pate, 1995).

Certainly one of the main limitations of all isotopic approaches for quantifying RWU and HR is the destructive character of isotopic sampling (see section 3.1) and associated offline analyses (sections 2.2 and 2.3). This usually leads to poor spatial

(maximum a few $cm^2$) as well as temporal (minimum hourly) resolution of the inferred results, when comparing with measuring frequency of other soil and plant state variables, e.g., soil water content and potential, and leaf water potential (section 3.2.2). In addition, one may question the representativeness of plant samples, in which tissues (and thus water) with





very different water residence time is mixed. Similarly, given the expected high lateral and temporal variability of the HR process, the representativeness of $\delta_S$ should be questioned for soils, in particular when combined with 1D models.

Recently developed methods take advantage of laser-based spectroscopy which allows on-line and continuous isotopic measurements in the gas phase. These methods rely on coupling a laser spectrometer with specific soil gas sampling probes

consisting of gas-permeable microporous polypropylene membranes or tubing. These membranes or tubing exhibit strong hydrophobic properties, while their microporous structures allow the intrusion and collection of soil water vapor. Several authors (Gaj et al., 2015; Gangi et al., 2015; Herbstritt et al., 2012; Rothfuss et al., 2013; Sprenger et al., 2015; Volkmann and Weiler, 2014) could determine the soil liquid water isotopic composition in a nondestructive (yet invasive) manner from that measured in the collected soil water vapor considering thermodynamic equilibrium between vapor and liquid phase in

the soil. In contrast to "traditional" isotopic methods, these novel isotopic monitoring methods have also the distinct advantage of determining soil liquid water isotopic composition at very low water content, since water vapor, in contrast to soil liquid water, is not limiting for analysis. These novel methods allow a vertical resolution down to 1 cm and an approximately hourly time-resolution. However, they do not allow horizontal resolution along the tube and are greatly sensitive to the carrier gas used (Gralher et al., 2016). In their opinion papers, McDonnell (2014) and Orlowski et al. (2016)

also urged for a comparison between methods, which was addressed by Gaj et al. (2015) and Pratt et al. (2015).

Leaf and plant gas chamber systems provide indirect means for a nondestructive determination of $\delta_{Ti}$, i.e., by either assuming full steady-state conditions at the evaporative sites of the leaves ($\delta_{Ti} = \delta_T$) (e.g., Dubbert et al., 2014; Volkmann et al., 2016b). In the coming years, effort should be made towards developing novel methods for a direct and nondestructive determination of $\delta_{Ti}$ based on the use of gas-permeable membranes, which was recently initiated for trees (Volkmann et al.,

2016a). This should be further investigated to test applicability to other (non-woody) plant species. This will imply the major challenge of not disrupting the water columns in the active xylem vessels when installing such a membrane-based system. Another potential issue to be investigated is the species-specific extent of water exchange between xylem and phloem conductive tissues which might lead to isotopic "contamination" of the xylem sap water.

### 5.3 Call for a coupled experiments-modelling approach for determination of plant water sources and redistribution
### on the basis of isotopic data

In order to fully benefit from the potential of water stable isotopologue analysis as tools for partitioning transpiration flux, the authors call for a generalization of coupled approaches based on the confrontation of experimental data with a physically based understanding of RWU processes.

Simple analytical models, such as the formulation of Couvreur et al. (2012), can be applied and confronted with isotopic

data. In comparison with statistical tools, such physical models provide profiles with high spatial resolution and lower uncertainty, on the condition that all required (isotopic) data is available. We recognize that in comparison with the statistical and conceptual methodologies presented in this review, using a physical (analytical or numerical) model implies the



measurements of additional state variables to be fed as input to the model, and of one parameter ($K_{plant}$) (when considering the assumption $K_{plant} = K_{comp}$ valid, see Appendix B). Some of these variables are laborious to obtain (e.g., *RLD*) or not straightforward to measure ($H_S$, $H_L$, and $T$) – especially in the field – but are mandatory to be able to determine contributions to $T$ across a set of identified water sources. In addition, they are necessary to gain insights into soil-plant interactions, e.g.,

5    dynamics of root function (active versus non-active roots in the soil profile) in water uptake and thus quantify the disconnection between measured *RLD* and the prognostic variable *SSF* (see Appendix B1) For doing this, controlled conditions in state-of-the-art climatic chambers are ideal, as they allow reducing the inherent spatial heterogeneity present under natural conditions and, thus, the deconvolution of environmental effects on RWU. Experimental facilities that not only control atmospheric forcing (soil upper boundary conditions for latent and heat flow), but impose lower boundaries for the

10   soil compartment (e.g., drainage and capillary rise dynamics) and provide means to close the hydrological balance are required. Moreover, macrocosm experiments (~m$^3$ scale) should be favored over mesocosm (~dm$^3$ scale) experiments to avoid or reduce inherent side effects that would ultimately hamper mimicking natural conditions.



# 6 CONCLUSION

Root water uptake is a key process in the global water cycle. More than 50% of total terrestrial evapotranspiration crosses plant roots to go back to the atmosphere (Jasechko et al., 2013). Despite its importance, quantification of root water uptake remains difficult due to the opaque nature of the soil and the spatial and temporal variability of the uptake process.

Water stable isotopic analysis is powerful and valuable tool for the assessment of plant water sources and for the identification of hydraulic redistribution. In an inverse modelling framework, isotopic analysis of plant tissues and soil also allow for obtaining species-specific parametrization of physically-based analytical and numerical RWU models. They provide at the plant scale a unique way to tackle the difficulty of disentangling actual RWU profiles with root traits and characteristics.

In this review we tried to highlight the importance of systematically reporting uncertainties along with estimates of contribution to $T$ of given plant water sources. The inter-comparison exercise could quantify the impact of the definition of the plant water sources (i.e., whether they are spatially disjoint or not and whether their isotopic compositions values are significantly different or not) on the outcome of the two end-member mixing model. The inter-comparison also illustrated the limitations of the graphical inference method and the multi-source mixing model of Phillips and Gregg (2003), whereas it

underlined the performance of the Bayesian approach of (Parnell et al., 2013), which uses a more rigorous statistical framework, if the number of considered water sources matches the number of isotopic measurements in the soil profile. However, contrary to the analytical model none of the graphical and statistical methods could locate and quantify hydraulic redistribution of water.

Finally, the authors call for (i) the further development of nondestructive and on-line isotopic measurement methodologies to

circumvent the necessity of sampling soil material and plant organs destructively, and (ii) a generalization of coupled approaches relying on the confrontation between labelling experiments under controlled conditions and three dimensional RWU numerical modelling.





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





## AUTHOR CONTRIBUTION

Y. Rothfuss reviewed the published literature. Y. Rothfuss and M. Javaux designed the virtual experiments, analysed, and discussed the obtained results.

## ACKNOWLEDGEMENTS

5  This study was conducted in the framework of and with means from the Bioeconomy Portfolio Theme of the Helmholtz Association of German Research Centers. The authors would like to thank Harry Vereecken, Jan Vanderborght, and Nicolas Brüggemann for their insightful comments on the manuscript.




**TABLES**

**Table 1**

**Table 1: Summary of the reviewed studies that use one of either the three methods (graphical inference (GI), two-end members mixing model (TM), and "multi-source mixing models (MS)) for plant water sources partitioning.**

| Authors | Experimental conditions (field: F / laboratory: L) |
|---|---|
| **Graphical inference (GI)** | |
| Leroux et al., 1995 | F (tropical) |
| Thorburn and Ehleringer, 1995 | F (semi-arid; cold desert, subhumid) |
| Weltzin and McPherson, 1997 | F (temperate semi-arid savanna) |
| Jackson et al., 2009 | F (tropical forest / savanna) |
| Moreira et al., 2000 | F (eastern Amazon) |
| Chimner and Cooper, 2004 | F (desert) |
| Kulmatiski et al., 2006 | F (shrub-steppe) |
| Li et al., 2007 | F (cold continental-semiarid) |
| Wang et al., 2010 | F (warm temperate / monsoon climate) |
| Stahl et al., 2013 | F (tropical) |
| **Two-end members mixing model (TM)** | |
| White et al., 1985 | F (temperate) |
| Dawson and Ehleringer, 1991 | F (riparian zone) |
| Brunel et al., 1995 | F (aeolian sand dune) |
| Dawson and Pate, 1996 | F (mediterranean) |
| McCole and Stern, 2007 | Field conditions (subtropical) |
| Goebel et al., 2015 | F (semi-arid) |
| **Multi-source mixing model (MS)** | |
| Asbjornsen et al., 2007 | F (cornfield, prairie, oak savanna, and woodland) |
| Wang et al., 2010 | F (warm temperate / monsoon climate) |
| Huang and Zhang, 2015 | F (desert) |
| Prechsl et al., 2015 | F (temperate) |
| Volkmann et al., 2016 | F (temperate) |





| Plant species |
| --- |
| Hyparrhenia diplandra, Andropogon schirensis, Imperata cylindrica (grasses), Cussonia barteri, Crossopteryx febrifuga, Bridelia ferruginea (shrubs) |
| Eucalyptus largiflorens, camaldulensis, Acer negundo and grandidentatum, Atriplex canescens, Chrysothamnus, nauseosus, Vanclevea stylosa |
| Quercus emoryi Torr., Trachypogon montufari (H.B.K.) Nees. (deciduous) Qualea grandiflora Mart., Q. parviflora Mart., Kielmeyera coriacea (Spr.) Mart., Pterodon pubescens Benth., Dalbergia myscolobium Benth. / (evergreen) Didymopanax macrocarpum, Sclerolobium paniculatum, Miconia ferruginata, Roupala montana |
| (invasive) Solanum crinitum Lamb. (native) Panicum maximum Jacq. |
| Sarcobatus vermiculatus, Chrysothamnus nauseosus, and Chrysothamnus greenei |
| (invasive) Centaurea diffusa, (native) Pseudoroegneria spicata; Bromus tectorum L. |
| Larix sibirica |
| Summer corn and cotton (species not specified) |
| tropical rainforest trees |
| Taxodium distichum / Pinus strobus |
| Acer grandidentatum Nutt., A. negundo L., Quercus gambelii Nutt. |
| "Mallee Tree" (Eucalyptus sp.) |
| Banksia prionotes, Dryandra sessilis, Grevillea (species unknown) |
| Juniperus ashei |
| Gossypium hirsutum L. |
| Quercus macrocarpa, Umus americana L. (trees), Zea mays L. (crop), and Andropogon gerardii (grass) |
| Summer corn and cotton (species not specified) |
| Caragana korshinskii and Artemisia ordosica |
| Phleum pratense, Lolium multiflorum, Poa pratensis, Taraxacum officinale, Trifolium repens, Rumex obtusifolius, Trisetum flavescens, Phleum rhaeticum, Carum carvi, and Achillea millefolium, Rumex alpestris, Taraxacum officinale and Trifolium, pratense. Quercus petraea and Fagus sylvatica |


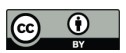

| SAMPLES FOR ISOTOPIC MEASUREMENTS | |
| --- | --- |
| Soil profile (soil depth: SD [m] / increment: l [m] / number of profiles: Nps / Replicates) | Roots profile (soil depth: SD [m] / increment: l [m] / number of profiles: Npr) |
| $0.10 < SD < 2.00 / 0.01 < l < 0.10 / R = 3$ | $SD = 1.80 / l = 0.10 / R = 16$ |
| No profiles, but soil directly surrounding roots are sampled | No profiles, but single roots sampled after excavation |
| $SD = 1.50 / l = 0.05 / 3 < R < 4$ | none |
| $SD = 5.00$ (depending on the site) $R = 2 / 0.05 < l < 0.20$ | none |
| $4 < SD < 6 / 0.05 < l < 2.00$ | $SD = 4 / 0.25 < l < 0.50 / R = 3$ |
| (1st campaign) $SD = 0.6 / l = 0.10 / R= 2$ at each site / $4 < Nps$ (per site) $< 6$ / (2nd campaign) $SD = 2.1 / 0.2 < l < 0.30 / Nps = 10$ | none (reference made to Cooper and Chimner, unpublished data) |
| $1.05 < SD < 2.20 / 0.10 < l < 0.60$ | $1.20 < SD < 2.20 / l = 0.15$ |
| $SD = 1.00 / 0.05 < l < 0.30$ | none |
| $SD = 1.50 / 0.05 < l < 0.30 / Nps = 7$ | none |
| $SD = 2.00 / 0.20 < l < 0.30 / R = 6$ | none (reference made to other literature) |
| none | none |
| $SD = 0.50$ | none |
| $50 < SD < 4.2 / 0.10 < l < 0.25$ | none |
| none (sand around the root system) | none ( qualitative observation) |
| $SD = 0.30 / 0.05 < l < 0.10 / 4 < Nps$ (per site) $< 5$ | none (reference made to other literature) |
| $SD = 0.3 / l \sim 0.02 / Nps = 4$ per irrigation treatment | none |
| $1.40 < SD < 2.00 / 0.05 < l < 0.20 / R=2$ per site | none |
| $SD = 1.50 / 0.05 < l < 0.30 / Nps = 7$ | none |
| $SD = 2.00 / 0.05 < l < 0.50$ | none |
| $0.30 < SD < 0.40 / 0.04 < l < 0.10 / R = 3$ | $SD = 0.30 / 0.075 < l < 0.125 / 6 < Npr < 7$ |
| $SD = 0.60 / 0.05 < l < 0.10 / Nps > 17$ | $SD = 0.60 / 0.05 < l < 0.10 / R = 4$ |



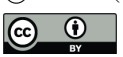

| Plant (organ: O / number of samples: Ns / replicates: R / temporal resolution: TR [hrs]) | Additional isotopic meas. (groundwater: G / spring: S / river: R / precipitation: P) |
|---|---|
| O: sapwood trunks (shrub), Crown (grasses) / 8 < Ns < 24 / 0.5 < TR < 1 | none |
| O: non-green, suberised stems / Ns = 3 | none |
| O: stem with phloem tissue / R = 4 (tree and sapling) / O: stem without green tissue / R = 4 (seedling) / O: culm base with sheaths removed / 3 < R < 4 | P |
| O: wood or suberized, mature, stem segments (outer bark and phloem are removed) / R =2 | none |
| O: well suberized stems (trees) / thick fleshy culms covered with dry leaves (grass) / 3 < R < 5 | none |
| O: fully suberized stem sections from the base of plants / (1st campaign, S. Vermiculatus) 4 < Ns (per site) < 8 / R =3 / (1st campaign, C. nauseosus) 5 < Ns (per site) < 6 / R = 3 / (2nd campaign) 3 < Ns < 5 (depending on the species) | GW / P / R |
| O: stem | P / Snow |
| O: stem / Ns (per sampling date) = 5 | P / R |
| O: stems (epidermis contacted with air was removed) / Ns = 7 / TR = 1 per vegetation stage | P/ GW / Irrigation water |
| O: branch (length = 0.07 m; diameter = 0.1 – 0.3 m). Bark tissue is immediately removed | none |
| O: wood samples taken at breast height | P / GW / R |
| O: mature suberized stems | P / Stream / S |
| O: twigs (bark is removed) / R = 2 | none |
| O: roots, trunks, stem base | GW / P |
| O: stem | P / S |
| O: meristematic petiole reduced in size to 5mm. / 14 < Ns < 17 (depending on the irragtion treatment) / R = 2 / TR = 1 | P |
| O: stem (trees); aerial nodal roots just above the soil surface (Zea mays L.); stem (grass) / Ns = 2 per specie | none |
| O: stems (epidermis contacted with air was removed) / Ns = 7 / TR = 1 per vegetation stage | P/ GW / Irrigation water |
| O: twigs (1–2 cm of stem with bark immediately removed) / R = 3 | P |
| O: root crown / 2 < Ns < 10 | none |
| none (measurement of transpiration isotopic composition) | none |





| Natural Abundance: NAb / labelling Experiment: LE | Identified plant water sources | Water extraction method (cryogenic vacuum distillation: Cr / azeotropic distillation Az / direct equilibration: GI / mild vacuum distillation: Mi) | NON-ISOTOPIC MEASUREMENTS soil water Content: SWC / soil water potential: SWP / soil temperature: ST / leaf water potential: LWP / stem water potential: StWP / sap flow (heat pulse technique): SF / transpiration flux T / precipitation amount: Pa / leaf area index: LAI / depth of water table: DWT |
|---|---|---|---|
| Nab | Soil water across depths | Cr | SWC / LWP (not shown) |
| Nab | Soil water across depths | Cr / Az | SWP / StWP / SF |
| Nab | Soil water across depths | Cr | SWC / Pa |
| Nab | Soil water across depths | Cr | SWP / ST / StWP / SF |
| LE | Soil water at isotopic labeled depth | Cr | none |
| Nab | Soil water across depths | Cr | Pa / LAI |
| Nab | Soil water across depths | Cr | SWC / Pa / DWT |
| Nab | Soil water across depths | Cr | Pa |
| Nab | Soil water across depths | Cr | SWP |
| LE | Soil water across depths | Cr | SWC / Predawn LWP |
| Nab | GW / P | none (Sap flow water) | none |
| Nab | GW / Stream water | Cr | none |
| Nab | Upper / lower soil profile | Az | DWT |
| Nab | Upper / lower soil profile | Mi | none |
| Nab | Upper / lower soil profile | Cr | Pa |
| Nab | Precipitation / Irrigation | Cr | Pa |
| Nab | Soil water across depths | Di | none |
| Nab | Soil water across depths | Cr | SWP |
| Nab | Soil water across depths | Cr | Pa |
| NAb | Soil water across depths | Cr | SWC / Pa / Aboveground biomass dry weight |
| LE | Soil water across depths | none | SWC / T |





| Single: S / Dual: D / "Double Single": DS isotope approach |
| --- |
| S ($\delta^{18}$O) |
| S ($\delta^{2}$H) |
| D |
| S ($\delta^{2}$H) |
| S ($\delta^{2}$H) |
| S ($\delta^{18}$O) |
| DS |
| DS |
| DS |
| D |
| S ($\delta^{2}$H) |
| S ($\delta^{2}$H) |
| D |
| S ($\delta^{2}$H) |
| S ($\delta^{18}$O) |
| S ($\delta^{18}$O) |
| S ($\delta^{18}$O) |
| DS |
| DS |
| DS |
| S ($\delta^{2}$H) |



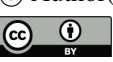

**Main results (RWU depth: $z_{RWU}$ [m] / soil depth z [m] / fraction of transpiration: x / Source: So)**

$0.00 < z_{RWU} < 0.05$ (grasses, early morning) / $0.05 < z_{RWU} < 0.10$ (grasses, midday) / $z_{RWU} = 0.30$ or $z_{RWU} > 1.50$ (shrubs, no unique solution)

Dominant source: groundwater (Mountain and floodplain) / $0.3 < z_{RWU} < 0.4$ (cold desert)

$z_{RWU} > 0.50$ (trees and sapling) / $z_{RWU} < 0.15$ (2-months-old seedling) / $0.20 < z_{RWU} < 0.35$ (1- and 2-year-old seedlings and grasses)

$z_{RWU} > 2.00$ (four evergreen and one deciduous species) / $z_{RWU} > 2.00$ (three deciduous and one evergreen species)

Fraction of root water uptake from the labeled region $x = 0.20$ (Solanum crinitum Lamb.) $z_{RWU} < 1.00$ (P. maximum Jacq.)

$0.00 < z_{RWU} < 0.50$ (C. nauseosus, pre-monsoon and monsoon periods) / (S. vermiculatus and C. nauseosus) dominant source: groundwater (pre-monsoon period) and switched to precipitation recharged water ($0.30 < z_{RWU} < 0.40$, during mansoon).

$z_{RWU} = 0.15$ (Bromus tectorum L., early season) / $z_{RWU} = 1.20$ (Centaurea diffusa Lam, late season)

$0.10 < z_{RWU} < 0.40$ from $\delta^{18}O$ meas / $0 < z_{RWU} < 0.80$ from $\delta^2H$ meas $z_{RWU} = 0.10$ (Corn, jointing stage) / $z_{RWU} = 0.50$ (Corn, flowering stage) /

$z_{RWU} = 0.10$ (Corn, full ripe stage) / $z_{RWU} = 0.40$ (Cotton, seedling stage) / $z_{RWU} = 0.50$ (Cotton, bud stage) / $z_{RWU} = 1.10$ (Cotton, bolls open stage)

$z_{RWU} > 1.00$ (dry periods, tall trees) / More diffuse $z_{RWU}$ for shorter trees

Groundwater: $0.46 < x < 0.64$ wet site / $0.16 < x < 0.25$ (intermediate site)

So: Groundwater (streamside mature trees) / So: stream water (younger streamside trees) / So: Precipitation (younger non-streamside trees)

$0 < z_{RWU} < 0.4$ with $0.7 < x < 0.9$ (1st site) / $0 < z_{RWU} < 1.5$ with $x = 0.07$ (2nd site)

So: deeper soil layers (dry season) / So: shallow soil layers (wet season)

$z_{RWU} > 0.30$ m (hot, dry summer) / $0.10 < z_{RWU} < 0.30$

Evidence for shifting to rainwater predominantly

$0.0 < z_{RWU} < 0.20$ m with $x < 45\%$ and $x < 36\%$ (crop and grass, resp.) / $0.00 < z_{RWU} < 0.20$ m with $x < 40\%$ and $x < 20\%$ and $z_{RWU} > 0.60$ m with $x < 60\%$ and $x < 80\%$ (Q. macrocarpa and U. americana L., resp.)

$0 < z_{RWU} < 0.20$ with $96 < x < 99$ (Corn, jointing stage) / $0.20 < z_{RWU} < 0.50$ cm with $58 < x < 85$ (Corn, flowering stage), $0.00 < z_{RWU} < 0.20$ with $69 < x < 76$ (Corn, full ripe stage) // $0.00 < z_{RWU} < 0.20$ with $27 < x < 49$ (Cotton, seedling stage), $0.20 < z_{RWU} < 0.50$ cm with $79 < x < 84$ (Cotton, bud stage) / $0.50 < z_{RWU} < 0.90$ with $30 < x < 92$ (Cotton, blooming stage) / $z_{RWU} > 90$ cm with $69 < x < 92$ (Cotton, boll open stage)

$0.1 < z_{RWU} < 1.0$ (wet seasons) / $z_{RWU}$ not affected by small rainfall events / $z_{RWU} > 1.00$ with $x$ observed to increase from $2 (\pm 0.7)$ to $10 (\pm 1.4)$ % for both plants after large rainfall event.

$0 < z_{RWU} < 0.10$ with $0.43 < x < 0.68$ (Drought treatment) / $0 < z_{RWU} < 0.10$ with $0.04 < x < 0.37$ and $0.20 < z_{RWU} < 0.35$ with $0.29 < x < 0.48$ (control treatment)

constant RWU depth profile with $0.15 < x < 0.18$ (beech) / $x < 0.15$ for $z < 0.20$ and $0.15 < x < 0.25$ for $z > 0.30$ (beech/oak mixture and oak monoculture)




1    **Table 2**

| SOIL DATA | Shallow groundwater table (Sh) | | | | | | Deep groundwater table (De) | | | | | | RLD (cm cm⁻³) |
|---|---|---|---|---|---|---|---|---|---|---|---|---|---|
| z (m) | Dry surface conditions (ShDr) | | | Wet surface conditions (ShWe) | | | Dry surface conditions (DeDr) | | | Wet surface conditions (DeWe) | | | |
| | $\theta$ (cm³ cm⁻³) | $H_S$ (cm) | $\delta_S$ (‰) | $\theta$ (cm³ cm⁻³) | $H_S$ (cm) | $\delta_S$ (‰) | $\theta$ (cm³ cm⁻³) | $H_S$ (cm) | $\delta_S$ (‰) | $\theta$ (cm³ cm⁻³) | $H_S$ (cm) | $\delta_S$ (‰) | |
| -0.01 | 0.235 | -454 | 5 | 0.372 | -2 | -7 | 0.044 | -9875 | 11 | 0.372 | -51 | -5 | 6.0 |
| -0.03 | 0.325 | -267 | 3 | 0.372 | -8 | -6 | 0.055 | -3581 | 7 | 0.371 | -77 | -5,5 | 3.0 |
| -0.07 | 0.347 | -215 | 1 | 0.372 | -11 | -5 | 0.081 | -1661 | 1 | 0.372 | -14 | -7 | 2.0 |
| -0.15 | 0.360 | -179 | -4 | 0.372 | -70 | -6 | 0.105 | -1165 | -3,5 | 0.135 | -869 | -3,5 | 0.8 |
| -0.30 | 0.367 | -155 | -6 | 0.370 | -125 | -6,5 | 0.122 | -989 | -4 | 0.134 | -889 | -4 | 0.5 |
| -0.50 | 0.371 | -135 | -7 | 0.371 | -135 | -7 | 0.165 | -730 | -5 | 0.165 | -730 | -5 | 0.4 |
| -1.00 | 0.372 | -125 | -7 | 0.372 | -125 | -7 | 0.210 | -620 | -6 | 0.210 | -620 | -6 | 0.3 |
| -2.00 | 0.372 | -125 | -7 | 0.372 | -15 | -7 | 0.259 | -600 | -7 | 0.259 | -600 | -7 | 0.2 |
| PLANT DATA | $T$ (mm h⁻¹) | $H_L$ (cm) | | $T$ (mm h⁻¹) | $H_L$ (cm) | | $T$ (mm h⁻¹) | $H_L$ (cm) | | $T$ (mm h⁻¹) | $H_L$ (cm) | | |
| IT | 0.01 | -587 | | 0.01 | -491 | | 0.01 | -2347 | | 0.01 | -918 | | |
| hT | 0.30 | -12330 | | 0.30 | -12234 | | 0.30 | -14090 | | 0.30 | -12661 | | |

3    **Table 2: Soil, plant, and isotopic input data for the different modelling approaches (depth ($z$) profiles of soil water**
4    **content $\theta$, total soil water potential $H_S$, soil water oxygen isotopic composition $\delta_S$, root length density $RLD$,**
5    **transpiration rate $T$, and leaf water potential $H_L$) "collected" during eight virtual experiments differing in the depth**
6    **of the groundwater table (Shallow –Sh / Deep – De) and the water status at the soil surface (Dry – Dr / Wet – We).**





**Table 3**

| Soil layer (m) | Shallow groundwater table (Sh) | | | | | | | | | | | |
| --- | --- | --- | --- | --- | --- | --- | --- | --- | --- | --- | --- | --- |
| | Dry surface conditions (ShDr) | | | | | | Wet surface conditions (ShWe) | | | | | |
| | Low $T$ (ShDr_lT) | | | High $T$ (ShDr_hT) | | | Low $T$ (ShWe_lT) | | | High $T$ (ShWe_hT) | | |
| | $x_{MSPa}$ mfv(range) (%) | $x_{AM}$ (1sd) (%) | Abs. diff. (%) | $x_{MSPa}$ mfv(range) (%) | $x_{AM}$ (1sd) (%) | Abs. diff. (%) | $x_{MSPa}$ mfv(range) (%) | $x_{AM}$ (1sd) (%) | Abs. diff. (%) | $x_{MSPa}$ mfv(range) (%) | $x_{AM}$ (1sd) (%) | Abs. diff. (%) |
| 0-0.02 | 1,1(0-34,6) | 4,8(0,7) | 3,8 | 6(0-37) | 10,9(1,3) | 4,9 | 17,5(0-48) | 13,4(1,4) | 4,1 | 15,8(0-52,9) | 11,2(1,3) | 4,7 |
| 0.02-0.05 | 1,1(0-35,4) | 7,3(0,9) | 6,2 | 5(0-37,8) | 8,7(0,9) | 3,7 | 13,1(0-41,5) | 10,4(1) | 2,7 | 7,4(0-42,8) | 8,8(0,9) | 1,3 |
| 0.05-0.11 | 2,5(0-40,8) | 10,7(1,2) | 8,3 | 9,8(0-48,4) | 11,3(1,1) | 1,5 | 10,9(0-40,5) | 13,2(1,1) | 2,2 | 7,1(0-41,2) | 11,4(1,1) | 4,4 |
| 0.11-0.225 | 14,5(0-56,6) | 9,9(0,7) | 4,6 | 13,9(0-47,2) | 9,7(0,5) | 4,2 | 11,3(0-45,6) | 10(0,5) | 1,4 | 3,4(0-42,9) | 9,7(0,5) | 6,3 |
| 0.225-0.4 | 19,4(0-56,5) | 10,6(0,3) | 8,8 | 16(0-54,6) | 9,8(0,1) | 6,1 | 15,8(0-52,8) | 9(0) | 6,9 | 16,1(0-48,5) | 9,8(0,1) | 6,3 |
| 0.4-0.75 | 16,2(0-54,9) | 16,3(0,2) | 0,0 | 16,7(0-47,9) | 14,4(0,4) | 2,2 | 17,5(0-44) | 12,7(0,5) | 4,8 | 15,2(0-48,4) | 14,3(0,4) | 0,8 |
| 0.75-1.5 | 17,1(0-52,1) | 26,7(1,8) | 9,6 | 18,4(0-46,1) | 23,2(1,9) | 4,9 | 15,8(0-47,9) | 20,8(1,9) | 4,9 | 16,2(0-52,8) | 23(1,9) | 6,8 |
| 1.5-2 | 16,6(0-59,4) | 13,7(1,8) | 2,8 | 17,1(0-47,2) | 11,9(1,7) | 5,1 | 15,3(0-52,3) | 10,7(1,6) | 4,6 | 16(0-51,4) | 11,8(1,7) | 4,1 |
| | | MD | 5.5 | | | 4.1 | | | 3.9 | | | 4.3 |

| Soil layer (m) | Deep groundwater table (De) | | | | | | | | | | | |
| --- | --- | --- | --- | --- | --- | --- | --- | --- | --- | --- | --- | --- |
| | Dry surface conditions (DeDr) | | | | | | Wet surface conditions (DeWe) | | | | | |
| | Low $T$ (DeDr_lT) | | | High $T$ (DeDr_hT) | | | Low $T$ (DeWe_lT) | | | High $T$ (DeWe_hT) | | |
| | $x_{MSPa}$ mfv(range) (%) | $x_{AM}$ (1sd) (%) | Abs. diff. (%) | $x_{MSPa}$ mfv(range) (%) | $x_{AM}$ (1sd) (%) | Abs. diff. (%) | $x_{MSPa}$ mfv(range) (%) | $x_{AM}$ (1sd) (%) | Abs. diff. (%) | $x_{MSPa}$ mfv(range) (%) | $x_{AM}$ (1sd) (%) | Abs. diff. (%) |
| 0-0.02 | 1(0-42,2) | -169,5(15,8) | 170,4 | 1(0-40,9) | 5,1(0,7) | 4,1 | 1,5(0-48,9) | 23,6(2,3) | 22,2 | 10,4(0-52,3) | 11,5(1,3) | 1,1 |
| 0.02-0.05 | 1(0-41,7) | -16,9(1,4) | 17,8 | 2,2(0-45,2) | 7,9(0,9) | 5,6 | 16,6(0-54,8) | 18,4(1,6) | 1,9 | 13,2(0-54) | 9(1) | 4,1 |
| 0.05-0.11 | 1,2(0-44) | 18,9(6) | 17,7 | 5,3(0-47,2) | 11,6(1,2) | 6,3 | 16,4(0-57,6) | 21(1,5) | 4,5 | 16(0-50,5) | 11,7(1,1) | 4,3 |
| 0.11-0.225 | 2,6(0-55) | 27,7(5) | 25,1 | 11,3(0-51,2) | 10,3(0,7) | 1,1 | 1,2(0-38,9) | 3(0,2) | 1,8 | 12(0-43) | 9,4(0,5) | 2,6 |
| 0.225-0.4 | 6,9(0-75) | 33,3(3,8) | 26,4 | 17(0-50,5) | 10,6(0,2) | 6,3 | 0,8(0-38) | 1,3(0,4) | 0,4 | 8,7(0-38,4) | 9,5(0,1) | 0,9 |
| 0.4-0.75 | 14,5(0-67,7) | 57,3(3,3) | 42,9 | 17,1(0-55,9) | 15,8(0,3) | 1,3 | 4,5(0-46,2) | 6,9(0,8) | 2,4 | 15,4(0-52,5) | 14,1(0,4) | 1,3 |
| 0.75-1.5 | 16,4(0-73,8) | 98(0,6) | 81,6 | 16,2(0-54,4) | 25,6(1,8) | 9,3 | 16,1(0-51,2) | 16,6(2,3) | 0,5 | 16(0-45,4) | 22,9(1,9) | 6,9 |
| 1.5-2 | 16,7(0-76,1) | 51,1(3,5) | 34,4 | 17,5(0-53,3) | 13,2(1,7) | 4,3 | 17,9(0-53,2) | 9,2(1,8) | 8,8 | 15,9(0-46,2) | 11,8(1,7) | 4,1 |
| | | MD | 52.1 | | | 4.8 | | | 5.3 | | | 3.2 |

**Table 3: Most frequent value (mfv) and range of the density distribution curve of the relative contribution to transpiration across eight defined soil layers as determined by the Bayesian method of Parnell et al. (2010) ($x_{MSPa}$, %) and mean relative contribution (with standard deviation) provided by the analytical model of Couvreur et al. (2012) ($x_{AM}$, %). Profiles of relative contribution were computed for eight soil-plant virtual experiments differing in the depth of the groundwater table (shallow – Sh / deep –De), the soil surface water status (dry – Dr / wet – We), and the plant transpiration rate (low – lT / high – hT). The absolute difference between the $x_{MSPa}$ mfv and $x_{AM}$ for each soil layer (Abs. diff, %) and the mean absolute difference (MD) for each soil-plant experiment are also reported.**





1    **FIGURES**

2    **Figure 1**

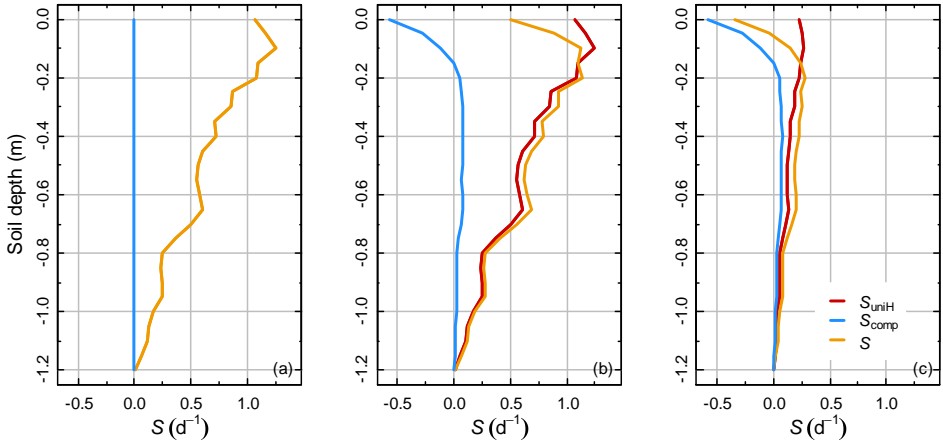

**Figure 1: Some examples of root water uptake sink term ($S$, in $d^{-1}$) profiles (orange lines) conceptualized as the sum of**
**two vertically distributed components, the root water uptake term proportional to root distribution ($S_{uniH}$, blue line)**
**and the compensatory root water uptake ($S_{comp}$, red line) accounting for heterogeneous soil water potential**
**distribution. (a) $S_{comp} = 0$ (no root compensation, i.e., soil water potential profile is homogeneous) leading to a situation**
**where $S = S_{uniH}$. (b) $S_{comp}$ is (i) different than zero and (ii) becomes negative at the surface but remains smaller (in**
**absolute term) than $S_{uniH}$. (c) $S_{comp}$ is (i) different than zero and (ii) becomes negative at the surface while becomes**
**greater (in absolute term) than $S_{uniH}$ at the surface ($z > -0.08$ m). In the last case, $S$ is negative at the surface, meaning**
**hydraulic redistribution – more specifically hydraulic lift – is observed.**





1    **Figure 2**

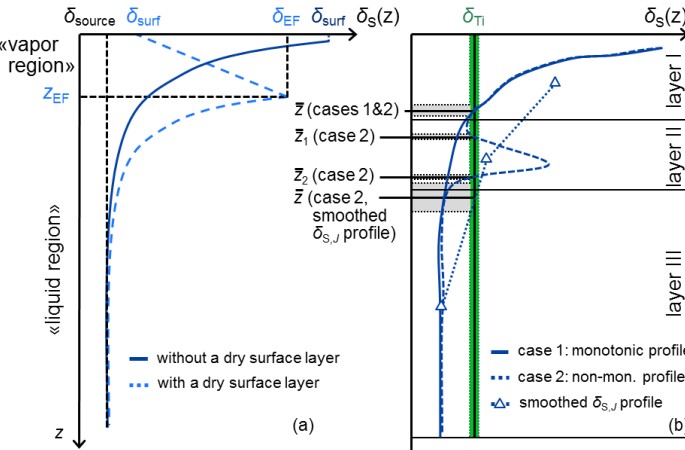

**Figure 2: (a) Simulated soil water isotopic composition ($\delta_S$) profiles under isothermal conditions for a water saturated**
**(black line) and unsaturated (gray line) soil following Zimmermann et al. (1967) and Barnes and Allison (1983).**
**Indices "surf" and "EF" refer to soil surface and Evaporation Front. "vapor" and "liquid" regions refer to soil**
**regions where water flow occurs predominantly in the liquid and vapor phase, respectively. (b) Illustration of the**
**"graphical inference" (GI) method for determining the "mean root water uptake depth" ($\bar{z}$) as the soil depth where**
**$\delta_S = \delta_{Ti}$ with "Ti" standing for the sap xylem water at the plant tiller. Case 1 represents the condition for which a**
**unique solution is found and case 2 the condition with more than one solution due to a heterogeneous $\delta_S$ profile. In the**
**latter case, a possibility is to smooth the $\delta_S$ profile (using Eq. (5)) until it is monotonous (smoothed profile is**
**designated by the symbols). In this example, the $\delta_S$ profile is smoothed over three layers and a uniform soil water**
**content profile is implied. $\bar{z}$ range (gray horizontal stripes) is determined by taking into account the uncertainty**
**associated with measurement of $\delta_{Ti}$ (i.e., precision of the isotopic analysis; green vertical stripe).**





1    **Figure 3**

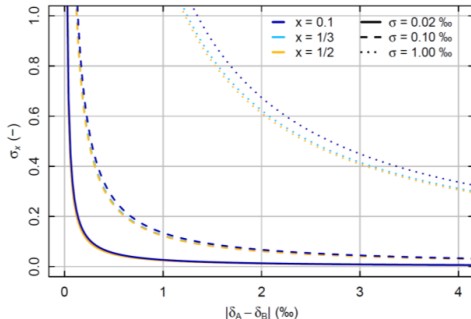

Figure 3: Standard error ($\sigma_x$) associated with the estimation of the relative contribution ($x$) of source A water to root
water uptake in case of two distinct sources (A and B of isotopic compositions $\delta_A$ and $\delta_B$). Following Eqs. (8a) and (8b)
(Phillips and Gregg, 2001), $\sigma_x$ is a function of the difference ($\delta_A - \delta_B$) (represented in absolute term in this Figure), of
the value of $x$ (three values are tested here: 0.1, 1/3, and 1/2, corresponding to dark blue, light blue and orange lines),
and of the standard errors associated with sampling and measurement of $\delta_A$, $\delta_B$, and of the isotopic composition of the
tiller sap water ($\delta_{Ti}$) (three values are tested, i.e., 0.02, 0.10, and 1.00, corresponding to solid and dashed lines).





1    **Figure 4**

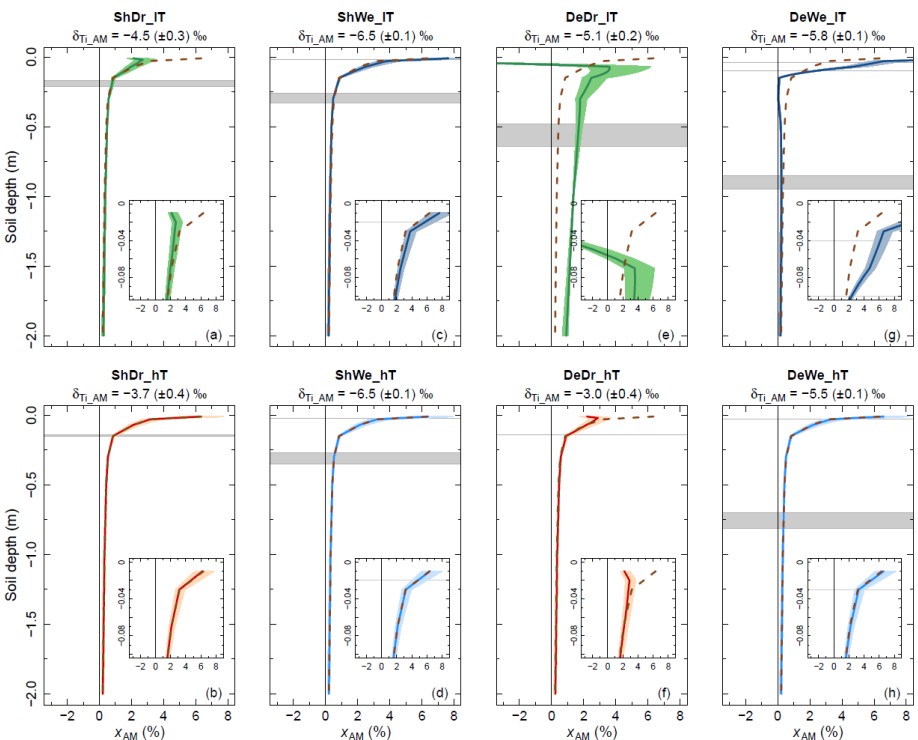

Figure 4: Simulated depth ($z$, in m) profiles of $x_{AM}$ (%) (solid colored lines), the simulated ratio $\dfrac{S_{AM}(z)dz}{T/(\Delta x \cdot \Delta y)}$

**provided by the model of Couvreur et al. (2012) (Eq. (B4'), see Appendix B) on the basis of input data reported in**
**Table 2 for experiments "ShDr" (soil with shallow groundwater table and relatively dry soil surface), "ShWe" (soil**
**with shallow groundwater table after a rainfall event)", "DeDr" (soil with deep groundwater table with relatively dry**
**soil surface), "DeWe" (soil with deep groundwater table after a rainfall event). Suffices "IT" and "hT" refer to "low"**
**and "high" transpiration rate simulations. Note that negative $x_{AM}$ means hydraulic redistribution (HR) by the roots.**
**The color-shaded areas depict the results of 1000 model runs where for each input data variable (soil water potential,**
**$\delta^{18}O$, and root length density – RLD) a single offset randomly selected between –5 and +5 cm, –0.2 and +0.2 ‰, and –**
**0.1 and +0.1 cm cm$^{-3}$ respectively for each variable was added to the initial values reported in Table 2. This should**
**represent the uncertainty of the model estimates on the account of the precision of the measurements. The horizontal**
**gray-shaded areas delimit the soil layers that contribute to RWU as obtained by the "graphical inference" (GI)**
**method, i.e., by locating the depth of the intersection between a vertical line of value $\delta_{Ti\_AM}$ and the soil water isotopic**
**profile. Layers' lower and upper boundaries are inferred from the uncertainty of the $\delta_{Ti\_AM}$ estimates for each**
**scenario ($\delta_{Ti\_AM}$ is given above each plot along with its standard deviation). At the bottom right corner of each plot is**
**represented a detail for $z \geq -0.10$ m. Finally, results from the first term of the model of Couvreur et al. (2012) which**
**considers uptake proportional to $RLD$ ($S_{uniH}$, Eq. (B4)) is plotted as a dashed brown line for comparison.**



**Figure 5**

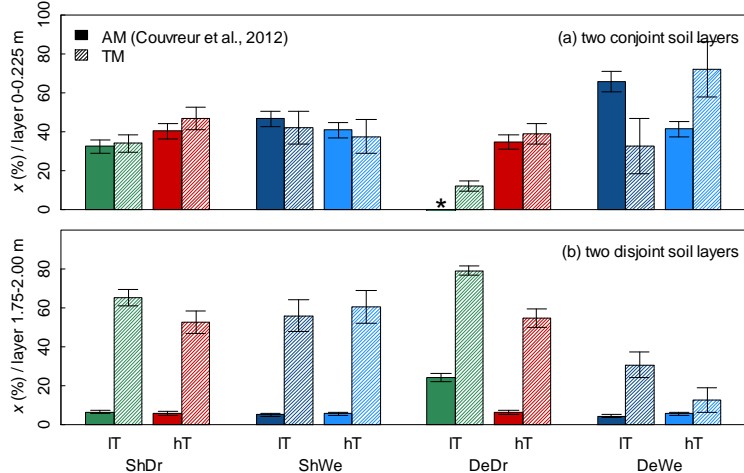

**Figure 5: Comparison between relative contributions to transpiration ($x$, in %) from one soil layer simulated by the analytical RWU model of Couvreur et al. (2012) and the two end-member mixing model (TM, see section §3.2.1) in case of two defined soil layers. Figure 5a displays $x$ from the topmost soil layer (0 – 0.225 m) in case of a two conjoint soi layers (0 – 0.225 m and 0.225 – 2.00 m) whereas Figure 5b displays $x$ from the lowest soil layer (1.75 – 2.00 m) in case of a two disjoint soil layers (0 – 0.225 m and 0.225 – 2.00 m), i.e., information on soil water isotopic composition is lacking between 0.225 and 1.75 m. "Sh" ("De") stands for the virtual experiments where the soil has a shallow (deep) groundwater table while "Dr" and "We" stand for when the soil is dry or wet at the surface (e.g., shortly after a rain event). Suffices "lT" and "hT" refer to "low" and "high" transpiration rate simulations. "\*" refers to when hydraulic redistribution is simulated by the analytical model, leading to a negative $x$. Error bars refer to either one standard deviation (for the RWU analytical model) or one standard error (for the TM approach).**





**Figure 6**

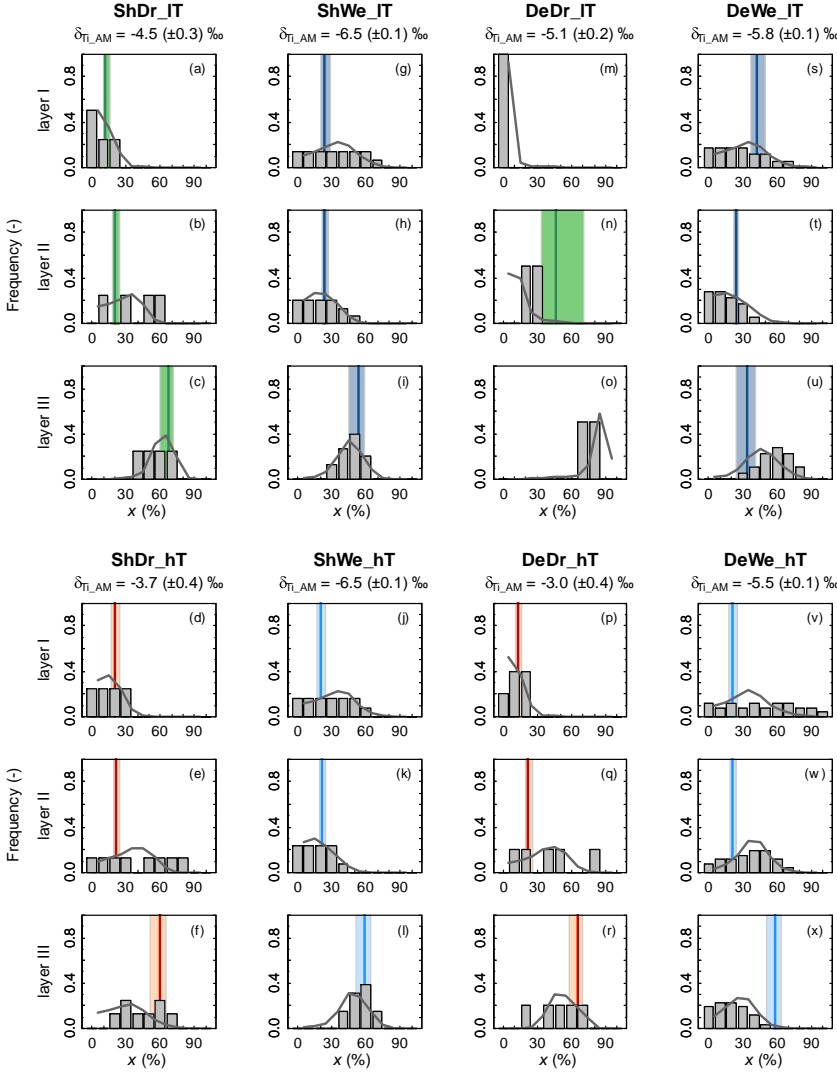

**Figure 6: Simulated ranges of possible relative contributions to transpiration from three defined soil layers (I: 0.00 – 0.05 m, II: 0.050 – 0.225 m, and III: 0.225 – 2.000) following the method of Phillips and Gregg (2003) ($x_{MSPG}$, in %, displayed in the form of gray histograms). Density distributions functions following the Bayesian approach of Parnell et al. (2010) ($x_{MSPa}$, gray lines). $x_{MSPG}$ and $x_{MSPa}$ were obtained from the confrontation of soil water $\delta^{18}O$ profiles (Table 1) and the $\delta_{Ti}$ simulated by the model of Couvreur et al. (2012) (i.e., $\delta_{Ti\_AM}$, given above each plot along with its standard deviation). Tolerance of the MSPG was set equal to the standard deviation of $\delta_{Ti\_AM}$. "Sh" ("De") stands for the virtual experiments where the soil has a shallow (deep) groundwater table while "Dr" and "We" stand for when the soil is dry or wet at the surface (e.g., shortly after a rain event). Suffices "lT" and "hT" refer to "low" and "high" transpiration rate simulations. The colored vertical lines give $x_{I\_AM}$, $x_{II\_AM}$, and $x_{III\_AM}$, the ratios $\dfrac{S_{AM}(z)dz}{T/(\Delta x \cdot \Delta y)}$ from layers I, II, and III to transpiration rate as simulated by the analytical model of Couvreur et al. (2012). The color-shaded areas associated with $x_{I\_AM}$, $x_{II\_AM}$, and $x_{III\_AM}$ vertical lines refer to their uncertainty associated with input data uncertainty (i.e., ±5 cm for soil water potential, ±0.2 ‰ for soil water $\delta^{18}O$, and ±0.1 cm cm$^{-3}$ for root length density).**




# APPENDIX

## Appendix A: List of symbols

| Symbol | Description | Dimension | Equation number | Measured (m) / simulated (s) / prescribed (p) |
|---|---|---|---|---|
| $C$, $C_S$, $C_A$, $C_B$, $C_{Ti}$ | Water stable isotopic concentration, soil water stable isotopic concentration, sources A and B water stable isotopic concentrations, xylem sap water isotopic concentration, root water uptake isotopic concentration | $M\,L^{-3}$ | 2, 3, 6a, 6b | m s |
| $E$, $E_i$ | Evaporation rate for $^1H_2^{16}O$ isotopologue, Evaporation rate for $^1H^2H^{16}O$ or $^1H_2^{18}O$ isotopologue | $L^3\,T^{-1}$ | B1-B4 | m/s |
| $h$ | Matric head | $L$ | | m |
| $H_{eq}$, $H_L$, $H_S$ | Soil water equivalent and leaf water potentials, total soil water potential | $P$ | | m |
| $J_A$, $J_B$, $J_{Ti}$ | Fluxes of water originating from water sources A and B, and at the plan tiller | $L^3\,T^{-1}$ | 6b | m |
| $J_i^A$, $J_i^B$, and $J_{Ti}^i$ | Fluxes of isotopologues originating from water sources A and B, and at the plan tiller | $M\,T^{-1}$ | 6a | m |
| $K_{plant}$, $K_{comp}$ | Plant and compensatory conductances to water flow | $L^3\,P^{-1}\,T^{-1}$ | B1-B4 | m/p |
| $M_w$, $M_i$ | Molar masses of water and isotopologue ($^1H^2H^{16}O$ or $^1H_2^{18}O$) | $M\,L^{-3}$ | 3 | m |
| $RLD$ $RLD_{1D}$ | Root length density Root length density per unit of surface area | $L\,L^{-3}$ $L\,L^{-1}$ - | B3 | m/p m/p |
| $R_{ref}$ | Vienna-Standard Mean Ocean Water (V-SMOW) hydrogen or oxygen stable isotopic ratio | - | 3 | m |
| $S$, $S_{uniH}$, $S_{comp}$, $S_{AM}$ | Root water uptake sink term, Root water uptake sink term under uniform soil water potential distribution, compensatory root water uptake sink term, total root water uptake sink term as simulated by the analytical model of Couvreur et al. (2012) | $L^3\,L^{-3}\,T^{-1}$ | 1-4, 6b, B4,B5, 6b | s |
| $SSF$ | Standard sink fraction | - | B2, B4, B4' | m/p |
| $t$, $\Delta t$ | Time, time step | $T$ | 11 | m |
| $T$ | Transpiration flux | $L^3\,T^{-1}$ | 2, 4a, 4b, B1, B3, B4 | m |
| $x$, $x_j$ $x_{AM}$, $x_{J\_AM}$, $x_{J\_MSPG}$, $x_{J\_MSPa}$ | Contributive proportion to transpiration, source $j$ contributive proportion to transpiration, continuous and integrated (layer $J$) contributive proportions to transpiration as simulated by the analytical model of Couvreur et al. (2012), integrated (layer $J$) contributive proportions to transpiration as determined by the statistical approaches of Phillips and Gregg (2003) and Parnell et al. (2010). Contributive proportion to transpiration under conditions of uniform soil water potential | - | 7, 8b, 9, 9' | s |
| $z$, $z_j$, $z_{j+1}$, $\Delta z_j$, $z_{max}$, $\overline{z}_{RWU}$ | Soil depth, soil depth of layers $j$ and $j$+1, thickness of soil layer $j$, depth of the root system, "mean root water uptake depth" | $L$ | 4b, 5, B2-B4' | m/p |





| | | | | |
|---|---|---|---|---|
| $\alpha_{eq}$, $\alpha_K$, $\alpha_{eq}^{^2H}$, $\alpha_{eq}^{^{18}O}$, $\alpha_K^{^2H}$, $\alpha_K^{^{18}O}$ | Equilibrium and kinetic isotopic fractionation factors, hydrogen and oxygen equilibrium isotopic fractionation factors, hydrogen and oxygen kinetic isotopic fractionation factors | - | | m/s |
| $\delta$, $\delta^2H$, $\delta^{18}O$, $\delta_l$, $\delta_v$, $\delta_{l\text{-}v}$, $\delta_{source}$, $\delta_{surf}$, $\delta_{sim}$, $\delta_S$, $\delta_{S,j}$, $\delta_{S,J}$, $\delta_A$, $\delta_B$, $\delta_{Ti}$, $\delta_{Ti,m}$, $\delta_{Ti\_AM}$, $\delta_E$, $\delta_T$ | Water stable isotopic composition, water hydrogen and oxygen stable isotopic compositions, liquid, vapor, liquid-vapor interface, source, soil surface, and simulated water isotopic compositions, soil water isotopic composition, soil layer $j$ and $J$ water isotopic composition, sources A and B water stable isotopic compositions, isotopic composition of xylem sap water at the plant tiller, isotopic composition of xylem sap water measured at the plant tiller, isotopic composition of xylem sap water at the plant tiller as simulated by the model of Couvreur et al. (2012), isotopic composition of transpiration | - (expressed in ‰) | 3-5, 7-9, 11 | m/s |
| $\varepsilon_j$ | Residual error term | - (expressed in ‰) | 9' | s |
| $\theta$ | Soil volumetric water content | $L^3 L^{-3}$ | 5, 11 | m |
| $\rho$ | Volumetric mass of water | $M L^{-3}$ | 3 | m |
| $\sigma_x$, $\sigma_{\delta_A}$, $\sigma_{\delta_B}$, $\sigma_{\delta_{Ti}}$, $\sigma_{\delta_{Ti\_AM}}$, $\sigma_x$ | Sandard errors associated with the measurements of $x$, $\delta_A$, $\delta_B$, $\delta_{Ti}$ and estimated uncertainty of $\delta_{Ti\_AM}$ as simulated by the analytical model of Couvreur et al. (2012), error associated with the estimation of the contributive proportion to $T$ of water source A in the case of two distinct sources | - (expressed in ‰) | 8a, 8b | s |
| $\tau$ | Isotopic tolerance | - (expressed in ‰) | 10 | p |





## Appendix B: The macroscopic RWU model of Couvreur et al. (2012)

### B1: Presentation of the model

In the approach of Couvreur et al. (2012), RWU is based on physical equations describing the water flow processes but without the need of the full knowledge of the root system architecture and local hydraulic parameters. Instead, three

macroscopic parameters are needed. The first equation defines plant transpiration:

$$J_{Ti} = K_{plant} \cdot \left( H_{eq} - H_L \right) \tag{B1}$$

where $J_{Ti}$ [L³ T⁻¹] is the sap flow rate in the root tiller and considered to be equal to the transpiration rate, $K_{plant}$ [L³ P⁻¹ T⁻¹] is the plant conductance to water flow (the first macroscopic parameter of Couvreur et al., 2012's model). $H_L$[P] is the leaf water potential and $H_{eq}$ [P] the "plant averaged soil water potential" defined as the mean soil water potential "sensed" by the

plant root system in the one dimensional (vertical) space:

$$H_{eq} = \int_z SSF(z) \cdot H_S(z) \tag{B2}$$

where $z$ is the soil depth, $H_S$ [P] is the total soil water potential, and $SSF$ [-] the standard sink term fraction (the second macroscopic parameter of the model of Couvreur et al., 2012). $SSF$ is defined as the RWU fraction under the condition of totally uniform soil water potential (i.e., when $H_S(x,y,z) = H_S = cst$). Under such conditions, if all the root segments had the

same radial conductivity (and the xylem conductance would not be limiting), the RWU distribution in a uniform soil water potential profile would be exactly the same as the root length density per unit of surface area ($RLD_{1D}$ of dimension [L L⁻¹]) profile. $SSF$ could be defined as:

$$SSF(z) = \frac{S_{uniH}(z)dz}{q_{Ti}} \approx \frac{RLD_{1D}(z) \cdot dz}{\int_z RLD_{1D}(z) \cdot dz} \tag{B3}$$

where $q_{Ti} = J_{Ti}/(\Delta x \cdot \Delta y)$ represents the sap flow rate in the root tiller per unit surface area [L T⁻¹], $S_{uniH}$ [T⁻¹] is the RWU sink

term under uniform soil water potential profile. The RWU under conditions of heterogeneous soil water potential is described with the following equation:

$$S(z) = S_{uniH}(z) + S_{comp}(z) = q_{Ti} \cdot SSF(z) + K_{comp} \cdot \frac{\left( H_s(z) - H_{eq} \right) \cdot SSF(z)}{V(z)} \tag{B4}$$

where $K_{comp}$ [L³ P⁻¹ T⁻¹] is the compensatory conductance and $S_{comp}$ [L³ T⁻¹] the compensatory RWU accounting for the non-uniform distribution of the soil water potential and $V(z)$ is the volume of soil considered. If the soil water potential is

uniform, this term vanishes from the equation, as $H_S = H_{eq}$ for any $z$, and water is extracted from the soil proportionally to $RLD$. When the water potential at a certain location is smaller (more negative, which means drier) than $H_{eq}$, less water is extracted from this location. On the other hand, when the soil is wetter ($H_S$ less negative), a larger amount of water can be taken up from the same location as compared. Note that if $H_S < H_{eq}$ and if the compensatory term is higher than the first one,





$S$ can become positive, and water is released to the soil (i.e., hydraulic redistribution – HR occurs). From Eq. (B4), it can be concluded that HR will preferably occur when $q_{Ti}$ is small and when large soil water potential gradients exist. Plant root hydraulic characteristics will control compensation through the $K_{comp}$ term. The importance of the compensatory RWU term has been discussed in the literature for a long time (e.g., Jarvis, 1989). Except if plants activate specific mechanisms to avoid

it, compensation always takes place under natural conditions due to the spatially heterogeneous distribution of soil water potential (Javaux et al., 2013).

A simplifying hypothesis that can be made (Couvreur et al., 2014; Couvreur et al., 2012) is to consider that $K_{plant}$ and $K_{comp}$ are equal, which substituted in Eq. (B4) leads to:

$$S(z) = SSF(z) \cdot K_{plant} \cdot \left(H_s(z) - H_L\right)/V(z) \qquad (B4')$$

Finally, the uptake of water stable isotopologues, i.e., the "isotopic sink term" ($S_i$ [M T$^{-1}$]) is defined as:

$$S_i(z) = S(z) \cdot C(z) \qquad (B5)$$

where $C$ [M L$^{-3}$] is the water isotopic concentration.

**B2: Running the model for the inter-comparison**

The root water uptake ($S_{AM}$) depth profiles and corresponding $\delta_{Ti\_AM}$ were simulated using the model of Couvreur et al.

(2012) (Eq. (B4')) for all eight scenarios. For this, $H_S$, $\delta_S$, and $RLD$ input data were interpolated at a 0.01 m vertical resolution and the resistance of the xylem vessels was assumed to be neglible so that $H_{Ti} = H_L$. A $K_{plant}$ value of 2.47 10$^{-6}$ h$^{-1}$ was taken and was determined based on concomitant $T$, $H_{eq}$ and $H_L$ data measured for *Festuca arundinacea*. $\delta_{Ti\_AM}$ was then calculated from Eq. (4b) (section §2.3). From these simulations, the depth profiles of $x_{AM}$ (%), the ratio $\dfrac{S_{AM}(z)dz}{T/(\Delta x \cdot \Delta y)}$ at each interpolated depth $z$ was determined, and $x_{J\_AM}$, the ratio $\dfrac{S_{AM}(z)dz}{T/(\Delta x \cdot \Delta y)}$ from each of the integrated soil layers $J$ ($J \le$ III or $J \le$

VIII) were calculated. In order to account for uncertainty of the input data (i.e., total soil water potential and oxygen isotopic composition $H_S$ and $\delta_S$, and root length density $RLD$), the model was run a 1000 times where a single offset randomly selected between –5 and +5 cm, –0.2 and +0.2 ‰, and –0.1 and +0.1 cm cm$^{-3}$ was added to the initial values (reported Table 2) of $H_S$, $\delta_S$, and $RLD$, respectively. By doing this we obtained *a posteriori* distributions of $S_{AM}$ and corresponding $\delta_{Ti\_AM}$ standard deviations ($\sigma_{\delta_{Ti\_AM}}$);




**Appendix C: Inter-comparison methodology**

The graphical inference method (GI), the two end-member mixing model (TM), and the statistical methods of Phillips and Gregg (2003) (MSPG) and Parnell et al. (2010) (MSPa) were compared to each-other in the following manner for each of the eight virtual experiments:

(i)  Single (or multiple) mean RWU depth(s) ($\bar{z}$) were graphically identified following the GI method as the depth(s) of the intersection between a vertical line of value $\delta_{\text{Ti\_AM}}$ with the $\delta_S$ profile. The uncertainty of method GI was determined on the basis of the $\delta_{\text{Ti\_AM}}$ *a posteriori* distribution: by taking into account $\sigma_{\delta_{\text{Ti\_AM}}}$, $\bar{z}$ results were translated into "RWU layers";

(ii)  relative contribution of RWU to transpiration ($x_{\text{TM}}$, %) to two defined soil layers (either conjoint: 0 – 0.225 m and
0.225 – 2.00 m or disjoint: 0 – 0.225 m and 1.75 – 2.00 m) were determined using the TM approach. For this, representative values for the water oxygen isotopic compositions of these soil layers were computed using Eq. (5) which uses soil volumetric water content ($\theta$, in m$^3$ m$^{-3}$) as input data. $\theta$ distribution was obtained from $H_S$ distribution and the van Genuchten (1980) closed-form equation. Values for its different parameters, i.e., the soil residual and saturated water contents ($\theta_{\text{res}}$ and $\theta_{\text{sat}}$), and the shape parameters related to air entry potential and pore-size distribution
($\alpha$ and $n$) were equal to 0.040 and 0.372 m$^3$ m$^{-3}$, 0.003 cm$^{-1}$, and 3.3, respectively;

(iii)  Possible range of $x_{J\_\text{MSPG}}$, the relative contribution of RWU to transpiration for each of the integrated soil layers following the MSPG method was computed based on smoothed $\delta_{S,J}$ profile and $\delta_{\text{Ti\_AM}}$ by solving the following equation:

$$\sum_J x_{J\_\text{MSPG}} \cdot \delta_{S,J} \leq \left| \delta_{\text{Ti\_AM}} \pm t \right| \qquad (C1)$$

with $\tau = \sigma_{\delta_{\text{Ti\_AM}}}$.

$\delta_{S,J}$ was computed similarly to for the TM method;

(iv)  Density distribution of $x_{J\_\text{MSPa}}$, the relative contribution of RWU to transpiration for each of the three (or eight) soil layers following the MSPG method was determined based on smoothed $\delta_{S,J}$ profile and $\delta_{\text{Ti\_AM}}$ data as well. To compare with the MSPG method (i) the number of $\delta_{\text{Ti}}$ replicates was fixed to three and equal to $\delta_{\text{Ti\_AM}} - \sigma_{\delta_{\text{Ti\_AM}}}$, $\delta_{\text{Ti\_AM}}$, and
$\delta_{\text{Ti\_AM}} + \sigma_{\delta_{RWU\_AM}}$, and (ii) $x_{J\_\text{MSPa}}$ was computed at a 10 % increment ($i$);

(v)  Results obtained at steps (i)-(iv) were compared to each other;

(vi)  Sensitivity of the MSPG method to the values of $i$ and $\tau$, and of the MSPa method to number of $\delta_{\text{Ti}}$ replicates, and to values of arguments *iterations* and *burnin* were finally briefly tested.