# Peer review of "Isotopic approaches to quantify root water uptake: a review and comparison of methods"

_Biogeosciences, 2016_

## Referee Comment (RC1) · M. Sprenger (Referee) · 7 Nov 2016

General comments

The manuscript by Rothfuss and Javaux on quantifying root water uptake by the means of isotopic approaches aims to provide both an overview of methods and a comparison of the methods with regard to their limitations for the interpretation. They further propose to include modeling approaches to better estimate the root water uptake patterns.

The manuscript is generally well prepared, with mostly sufficient references, in depth information, and proper visualizations. Due to its nature as a review, it is pretty long and I am not sure if the sections 2.1 and 2.2 are really necessary. I agree that it is

necessary to understand the soil water isotopic composition in order to interpret the root water uptake with the means of stable isotopes. The authors focus on evaporation fractionation as one process to alter the soil water isotopic composition. From my point of view, also the precipitation input (and its variability in time) would then need to be considered. However, this has been reviewed recently and would blow up the manuscript.

I think that the manuscript is a good contribution to current issues in ecohydrology and will be of interest to a broad readership. Therefore, I suggest a publication after a minor revision.

Specific comments

On page 5, where you introduce into the theoretical backgrounds, I do not think that the isotope depth profiles are solely a result of fractionation effects. It seems that you miss the importance of the variability of the isotopic signal of the precipitation input and its consequences for the spatial variability of the soil water isotopes over depth. As you write, the evaporation fractionation has been reviewed by Horita et al. (2008) and it was more recently reviewed by Soderberg et al. (2012). In order to streamline the manuscript and keep its focus on the root water uptake, I doubt that the section 2.1 and 2.2 are really necessary. However, I agree that it is necessary to account for the spatial and temporal variability of the soil water isotopes, but this is influenced by more than soil evaporation (see also my review Sprenger et al. (2016)).

I am wondering if the authors are aware of the work by Ogle et al. (2004) and Ogle et al. (2014), where they suggest to include biophysical conditions in a process-based mixing model ("root area profile and isotope deconvolution, RAPID"). I think it would be worth including this in the review of methods, since it goes in the same direction as the author's proposal of including more physical basis of the root water uptake estimates.

Please state that the recently published dependency on the carrier gas was found for a WS-CRDS (Picarro); I did not find this CO2 dependency for Off-axis ICOS (Los Gatos)

(currently in review).

Technical corrections

P1 L12/13: I suggest using "studies" rather than "authors"

P3 L11: Insert "and" for "in space," and replace "but also on the root's" with "and their"

P3 L25: Not sure what you mean with "by reference"

P3 L32: I suggest "distribution of S" instead of "S distribution

P3 L23: Introduce RLD here.

P4 L21: Why not stating directly that Zarebanadkouki et al. (2012) used deuterated water?

P4 L29: I suggest (or artificial enriched/depleted)

P5 L1: Please state here once again what "these methods" will be.

P5 L8: Is this not a bit too simplified at this point? You would always also need some kind of info about vegetation isotopes. I don't think the first sentence is necessary here.

P5 L9: Are you referring to S profiles or isotope depth profiles? I do not think that the isotope depth profiles are solely a result of fractionation effects.

P5 L 16: Consider splitting this long sentence.

P6 L 30: This line is not black anymore.

P7 L5: I think it would be worth including the definition by Barnes, Allison (1983) for the vapor region to be of total water potential about 15bar, while at lower potential there would be little connected water.

P7 L8: This line is not grey anymore

P7 L15: Is this supported by data? I don't see it like that in Rothfuss et al. (2015),

where you have shown that the slope for the depths above the EF (max -0.06 m) is still clearly below 8.

P7 L16: I have shown that for several studies in Sprenger et al. (2016).

P10 L8: delete second "is".

P10 L12: I summarized the uncertainty of the different methods to derive soil water isotope data in Sprenger et al. (2015).

P14 L25: I assume this is for d18O? Please clarify.

P14 L24: It would be interesting which rooting depth and density profile was assumed for the modeling. Please provide.

P16 L2: cases

P17 L28: What about the study by Dawson (1993) who provides valumes?

P18 L27: Why do you not include the direct-equilibration method by Wassenaar et al. (2008)?

P19 L7: I did not use in-situ in that study, but consider including a recent paper by Oerter et al. (2016).

P19 L14: Please not that this is the case for a WS-CRDS (Picarro); I did not find this CO2 dependency for Off-axis ICOS (Los Gatos) (currently in review)

P19 L25: I suggest referring to Farquhar et al. (2007). Table 1: I believe there are more studies than the listed ones. I suggest considering the following: Meinzer et al. (1999); Kulmatiski et al. (2010); Kulmatiski, Beard (2013); Evaristo et al. (2016); Goldsmith et al. (2012); Liu et al. (2011); Bertrand et al. (2012); Meißner et al. (2012); Dawson (1996); Bijoor et al. (2012)

Table 2: Why did you limit your analysis here to d18O, while emphasizing that dual isotope approaches would be preferable on page 12 L4?

Figure 1: Caption "negative towards the surface"

Figure 2: Update the caption according to the color of the lines (blue).

Figure 4: Caption "a detail is is presented for"

Figure 5: Why standard deviation and standard error for the different approaches?

Publication bibliography

Barnes, C.J; Allison, G.B (1983): The distribution of deuterium and 18O in dry soils. In Journal of Hydrology 60 (1-4), pp. 141–156. DOI: 10.1016/0022-1694(83)90018-5.

Bertrand, Guillaume; Masini, Jean; Goldscheider, Nico; Meeks, Jessica; Lavastre, Véronique; Celle-Jeanton, Hélène et al. (2012): Determination of spatiotemporal variability of tree water uptake using stable isotopes ($\delta$18O, $\delta$2H) in an alluvial system supplied by a high-altitude watershed, Pfyn forest, Switzerland. In Ecohydrol. 7 (2), pp. 319–333. DOI: 10.1002/eco.1347.

Bijoor, Neeta S.; McCarthy, Heather R.; Zhang, Dachun; Pataki, Diane E. (2012): Water sources of urban trees in the Los Angeles metropolitan area. In Urban Ecosyst 15 (1), pp. 195–214. DOI: 10.1007/s11252-011-0196-1.

Dawson, T. E. (1996): Determining water use by trees and forests from isotopic, energy balance and transpiration analyses: the roles of tree size and hydraulic lift. In Tree Physiology 16 (1-2), pp. 263–272. DOI: 10.1093/treephys/16.1-2.263.

Dawson, Todd E. (1993): Hydraulic lift and water use by plants: implications for water balance, performance and plant-plant interactions. In Oecologia 95 (4), pp. 565–574. DOI: 10.1007/BF00317442.

Evaristo, Jaivime; McDonnell, Jeffrey J.; Scholl, Martha A.; Bruijnzeel, L. Adrian; Chun, Kwok P. (2016): Insights into plant water uptake from xylem-water isotope measurements in two tropical catchments with contrasting moisture conditions. In Hydrol. Process. 30 (18), pp. 3210–3227. DOI: 10.1002/hyp.10841.

Farquhar, Graham D.; Cernusak, Lucas A.; Barnes, Belinda (2007): Heavy Water Fractionation during Transpiration. In Plant Physiol. 143 (1), pp. 11–18. DOI: 10.1104/pp.106.093278.

Goldsmith, Gregory R.; Muñoz-Villers, Lyssette E.; Holwerda, Friso; McDonnell, Jeffrey J.; Asbjornsen, Heidi; Dawson, Todd E. (2012): Stable isotopes reveal linkages among ecohydrological processes in a seasonally dry tropical montane cloud forest. In Ecohydrol. 5 (6), pp. 779–790. DOI: 10.1002/eco.268.

Horita, Juske; Rozanski, Kazimierz; Cohen, Shabtai (2008): Isotope effects in the evaporation of water: a status report of the Craig-Gordon model. In Isotopes in Environmental and Health Studies 44 (1), pp. 23–49. DOI: 10.1080/10256010801887174.

Kulmatiski, Andrew; Beard, Karen H. (2013): Root niche partitioning among grasses, saplings, and trees measured using a tracer technique. In Oecologia 171 (1), pp. 25–37. DOI: 10.1007/s00442-012-2390-0.

Kulmatiski, Andrew; Beard, Karen H.; Verweij, Richard J. T.; February, Edmund C. (2010): A depth-controlled tracer technique measures vertical, horizontal and temporal patterns of water use by trees and grasses in a subtropical savanna. In New Phytologist 188 (1), pp. 199–209. DOI: 10.1111/j.1469-8137.2010.03338.x.

Liu, Yuhong; Xu, Zhen; Duffy, Rodney; Chen, Wenlian; An, Shuqing; Liu, Shirong; Liu, Fude (2011): Analyzing relationships among water uptake patterns, rootlet biomass distribution and soil water content profile in a subalpine shrubland using water isotopes. In European Journal of Soil Biology 47 (6), pp. 380–386. DOI: 10.1016/j.ejsobi.2011.07.012.

Meinzer, F. C.; Andrade, José Luis; Goldstein, Guillermo; Holbrook, N. Michele; Cavelier, Jaime; Wright, S. Joseph (1999): Partitioning of soil water among canopy trees in a seasonally dry tropical forest. In Oecologia 121 (3), pp. 293–301. DOI: 10.1007/s004420050931.
Meißner, M.; Köhler, M.; Schwendenmann, L.; Hölscher, D. (2012): Partitioning of soil water among canopy trees during a soil desiccation period in a temperate mixed forest. In Biogeosciences 9 (8), pp. 3465–3474. DOI: 10.5194/bg-9-3465-2012.

Oerter, Erik J.; Perelet, Alexei; Pardyjak, Eric; Bowen, Gabriel (2016): Membrane inlet laser spectroscopy to measure H and O stable isotope compositions of soil and sediment pore water with high sample throughput. In Rapid communications in mass spectrometry : RCM. DOI: 10.1002/rcm.7768.

Ogle, Kiona; Tucker, Colin; Cable, Jessica M. (2014): Beyond simple linear mixing models. Process-based isotope partitioning of ecological processes. In Ecological Applications 24 (1), pp. 181–195. DOI: 10.1890/12-1970.1.

Ogle, Kiona; Wolpert, Robert L.; Reynolds, James F. (2004): Reconstruction plant root area and water uptake profiles. In Ecology 85 (7), pp. 1967–1978. DOI: 10.1890/03-0346.

Rothfuss, Y.; Merz, S.; Vanderborght, J.; Hermes, N.; Weuthen, A.; Pohlmeier, A. et al. (2015): Long-term and high-frequency non-destructive monitoring of water stable isotope profiles in an evaporating soil column. In Hydrol. Earth Syst. Sci. 19 (10), pp. 4067–4080. DOI: 10.5194/hess-19-4067-2015.

Soderberg, Keir; Good, Stephen P.; Wang, Lixin; Caylor, Kelly (2012): Stable Isotopes of Water Vapor in the Vadose Zone: A Review of Measurement and Modeling Techniques. In Vadose Zone Journal 11 (3), p. 0. DOI: 10.2136/vzj2011.0165.

Sprenger, Matthias; Herbstritt, Barbara; Weiler, Markus (2015): Established methods and new opportunities for pore water stable isotope analysis. In Hydrol. Process. 29 (25), pp. 5174–5192. DOI: 10.1002/hyp.10643.

Sprenger, Matthias; Leistert, Hannes; Gimbel, Katharina; Weiler, Markus (2016): Illuminating hydrological processes at the soil-vegetation-atmosphere interface with water stable isotopes. In Rev. Geophys. 54 (3), pp. 674–704. DOI: 10.1002/2015RG000515.

Wassenaar, L.I; Hendry, M.J; Chostner, V.L; Lis, G.P (2008): High Resolution Pore Water $\delta$2H and $\delta$18O Measurements by H 2 O (liquid) −H 2 O (vapor) Equilibration Laser Spectroscopy. In Environ. Sci. Technol. 42 (24), pp. 9262–9267. DOI: 10.1021/es802065s.

―――――――――――――――――――――――――

---

## Referee Comment (RC2) · Anonymous Referee #2 · 8 Nov 2016

General comments:

The manuscript aims to compare different methods to locate root water uptake depth. I understand that it is a review paper, however, it is rather long, way too complex, and unfortunately hard to follow (it even comes with an appendix). This is mainly due to the many formulas that are presented and which disturb the text flow. I agree that a method comparison is needed but the manuscript is blown up with a lot of "basic isotope knowledge" which for my feeling is not necessary in such extent. I would suggest to reduce the length of the whole manuscript and focus on what differentiates the three methods to be compared. Further, the authors should only present equations which are really needed to understand the method comparison. I believe that this would increase

the readability. Concerning the presented figures, I would recommend to keep them simpler as they should generally be self-explaining and not as complex as they are now. I would consider the manuscript ready for publication after major revision.

Specific comments:

Title: "quantify" instead of "quantifying"

p. 3ff: Introduction needs a better/clearer structure

p. 3 l. 2-6: References are missing

p.3 l. 7: "driven by transpiration taking place..."

p. 3 l.11: "spatial distribution...is very variable in time and space"; spatial in space → avoid duplication

p. 3 l. 13-14: Reference missing; permanent wilting point concept; what is a dry soil in this context?

p. 5 l. 1: "each other" without hyphen

p. 5 l. 3-6: Repetition from abstract

p. 5 chp. 2.1: This chapter is too detailed; fundamentals of isotope hydrology do not have to be explained in such detail

p. 6 l. 20: Reference missing

p. 7 l. 8: "grey" instead of "gray"

p. 7 l. 12-14: Repetition

p. 7 l. 24: Mention this earlier in the manuscript

p. 9 l. 16: 21 studies: Based on which criteria have these studies been selected? Literature review using ISI web of knowledge? Please mention briefly.

p. 9 l. 20: "unambiguously identified" → What about issues with regard to water extraction techniques which might be a cause for this?

p. 10 l. 7: grey → correct throughout the manuscript

p. 11 l. 26: Table 1 should rather go into the Introduction section, also it is too detailed

p. 12 l. 22: Replace www. by an abbreviation for example EPA, 2015; same for p. 13 l. 10

p. 13 l. 20ff: Why did the authors not intercompare the methods based on a dual isotope approach? How reliable/meaningful is a single isotope approach?

p. 14 l. 26: Is it necessary to mention the function?

p. 18: Think about renaming the subsection e.g. method uncertainties and. . .

p. 18 l. 1-20: This does not belong into the discussion section

p. 18 l. 21: Reference missing

p. 18 l. 21-24: Is this water plant available? Does it make sense to extract at such conditions if plant available soil water pools are of interest? Please discuss briefly.

p. 18 l. 24ff: Methods are also not intercomparable and each method comes with a huge uncertainty (e.g. Sprenger et al., 2015; Orlowski et al., 2016). How reliable is such data in the end when utilized for RWU calculations? How would RWU depth vary if e.g. extraction method uncertainty is accounted for?

p. 19 l. 3ff: Again, does this represent plant available water?

p. 19 l. 15: Gaj et al. (2015) is not a method comparison paper. Pratt et al. (2015) is wrong → Orlowski et al. (2016) and please cite Sprenger et al. (2015) as review paper about extraction method comparisons.

p. 19 l. 27: "generalization of coupled approaches" → What does that mean?

Table 1: Too much information

Table 3: Not sure if all these numbers are necessary to understand the method comparison/virtual experiments

---

## Author Comment (AC1) · 24 Nov 2016

General comments

The manuscript by Rothfuss and Javaux on quantifying root water uptake by the means of isotopic approaches aims to provide both an overview of methods and a comparison of the methods with regard to their limitations for the interpretation. They further propose to include modeling approaches to better estimate the root water uptake patterns.

The manuscript is generally well prepared, with mostly sufficient references, in depth information, and proper visualizations. Due to its nature as a review, it is pretty long and I am not sure if the sections 2.1 and 2.2 are really necessary. I agree that it is necessary to understand the soil water isotopic composition in order to interpret the root water uptake with the means of stable isotopes. The authors focus on evaporation fractionation as one process to alter the soil water isotopic composition. From my point of view, also the precipitation input (and its variability in time) would then need to be considered. However, this has been reviewed recently and would blow up the manuscript.

I think that the manuscript is a good contribution to current issues in ecohydrology and will be of interest to a broad readership. Therefore, I suggest a publication after a minor revision.

Specific comments

On page 5, where you introduce into the theoretical backgrounds, I do not think that the isotope depth profiles are solely a result of fractionation effects. It seems that you miss the importance of the variability of the isotopic signal of the precipitation input and its consequences for the spatial variability of the soil water isotopes over depth.

> *Dear Matthias, we fully agree with this. In a revised version, a text will be added on how soil upper-boundary isotopic condition (i.e., $\delta_{surf}$) - one of the determinants of the soil water isotopic composition profile – is highly impacted both spatially and temporally by input precipitation isotopic composition.*

As you write, the evaporation fractionation has been reviewed by Horita et al. (2008) and it was more recently reviewed by Soderberg et al. (2012). In order to streamline the manuscript and keep its focus on the root water uptake, I doubt that the section 2.1 and 2.2 are really necessary. However, I agree that it is necessary to account for the spatial and temporal variability of the soil water isotopes, but this is influenced by more than soil evaporation (see also my review Sprenger et al. (2016)).

> *The authors still think that sections 2.1 and 2.2 are necessary for the readers who, for instance, already study RWU yet are interested in using water stable isotopes to characterize it. These paragraphs will however be significantly shortened. About the remark that "spatial and temporal variability is influenced by more than soil evaporation", please refer to our answer to your specific comment below.*

I am wondering if the authors are aware of the work by Ogle et al. (2004) and Ogle et al. (2014), where they suggest to include biophysical conditions in a process-based mixing model ("root area profile and isotope deconvolution, RAPID"). I think it would be worth including this in the review of methods, since it goes in the same direction as the author's proposal of including more physical basis of the root water uptake estimates.

> *The authors are aware of the work of Ogle et al. (2004) where they could reconstruct "active root area" and RWU profiles from isotopic measurements (assuming, amongst other things, normal a priori distributions for the xylem water oxygen and hydrogen isotopic compositions and considering prior knowledge on x). Since we wanted to focus on the most used and cited methods for partitioning plant water sources, we choose not to incorporate that of Ogle in the model comparison (and for that same reason neither that of Romero-Saltos et al. (2005)). Nevertheless, we propose that in the revised version of the manuscript mention of Ogle et al. study is made in Section 5 (Challenges and progresses) as it nicely fits there as you point out.*

Please state that the recently published dependency on the carrier gas was found for a WS-CRDS (Picarro); I did not find this $CO_2$ dependency for Off-axis ICOS (Los Gatos) (currently in review).

> *Done. Thanks!*

Technical corrections

P1 L12/13: I suggest using "studies" rather than "authors"

*Done*

P3 L11: Insert "and" for "in space," and replace "but also on the root's" with "and their"

*Done*

P3 L25: Not sure what you mean with "by reference"

*We meant that the letter "S" makes reference to the "sink term". We will use the correct formulation "in reference to" instead of the incorrect "by reference to".*

P3 L32: I suggest "distribution of S" instead of "S distribution

*Done*

P3 L23: Introduce RLD here.

*Indeed, this was not properly introduced in the original manuscript. Done.*

P4 L21: Why not stating directly that Zarebanadkouki et al. (2012) used deuterated water?

*Done*

P4 L29: I suggest (or artificial enriched/depleted)

*Since "artificial" stands for "isotopic abundance", it cannot be followed by "enriched or depleted" but only "higher of lower" but then it reads a bit funny. We would like to keep the more general "artificial"*

P5 L1: Please state here once again what "these methods" will be.

*Done*

P5 L8: Is this not a bit too simplified at this point? You would always also need some kind of info about vegetation isotopes. I don't think the first sentence is necessary here.

*The authors write that for reconstruction of S profiles, one needs both soil and plant isotopic information ("within the soil-plant system").*

P5 L9: Are you referring to S profiles or isotope depth profiles? I do not think that the isotope depth profiles are solely a result of fractionation effects.

*We are referring to isotopic variations. It is implied here that these isotopic variations are solely due to the difference of physical properties of the different isotopologues (ultimately leading to isotopic fractionation) only in-between precipitation events, i.e., we rule out the partial to total reset of the isotopic profile due to each rain event.*

*We added "in-between precipitation events" at the beginning of the sentence and "isotopic" before "variations" to clarify the sentence.*

P5 L 16: Consider splitting this long sentence.

*We moved "are expressed in ‰ relative to the Vienna Standard Median Ocean Water international (VSMOW) isotope reference scale (Gonfiantini, 1978)" to the introduction of section 2 to lighten this long (!) sentence. Thanks.*

P6 L 30: This line is not black anymore.

*Thanks! Done*

P7 L5: I think it would be worth including the definition by Barnes, Allison (1983) for the vapor region to be of total water potential about 15bar, while at lower potential there would be little connected water.

*Done*

P7 L8: This line is not grey anymore

*Thanks! Done*

P7 L15: Is this supported by data? I don't see it like that in Rothfuss et al. (2015), where you have shown that the slope for the depths above the EF (max -0.06 m) is still clearly below 8.

*We write that "As a result, an intermediate value for the slope is expected, depending on the mixing ratio of atmospheric water vapor to evaporated soil vapor at a given soil depth." (P7*

*L16-18), meaning that the value for slope should lie between 2 and 8 (which is in accordance with Rothfuss et al. (2015)).*

P7 L16: I have shown that for several studies in Sprenger et al. (2016).

*Indeed. The list of three references has been simply replaced with your review "Sprenger et al. (2016)".*

P10 L8: delete second "is".

*Thanks! Done*

P10 L12: I summarized the uncertainty of the different methods to derive soil water isotope data in Sprenger et al. (2015).

*The reference is cited now. Done*

P14 L25: I assume this is for $\delta^{18}O$? Please clarify.

*Yes. We added "($\delta_S$)" after "soil water oxygen isotopic composition" (as well as "(T)" after "actual transpiration rate") above in section 4.1.1*

P14 L24: It would be interesting which rooting depth and density profile was assumed for the modeling. Please provide.

*This is explained P14 L5-6:*

*"[All scenarios] relied on a common measured root length density vertical distribution of Festuca arundinacea",*

*and in Appendix B2 (Running the model for the inter-comparison):*

*"For this, $H_S$, $\delta_S$, and RLD input data were interpolated at a 0.01 m vertical resolution…"*

P16 L2: cases

*Done*

P17 L28: What about the study by Dawson (1993) who provides volumes?

*To the authors' understanding, Dawson (1993) provided a lateral gradient of the proportion of hydraulically-lifted water from maple trees used by neighboring plants, not HL water volumes.*

P18 L27: Why do you not include the direct-equilibration method by Wassenaar et al. (2008)?

*Done*

P19 L7: I did not use in-situ in that study, but consider including a recent paper by Oerter et al. (2016).

*Done*

P19 L14: Please not that this is the case for a WS-CRDS (Picarro); I did not find this $CO_2$ dependency for Off-axis ICOS (Los Gatos) (currently in review)

*Done*

P19 L25: I suggest referring to Farquhar et al. (2007).

*Done*

Table 1: I believe there are more studies than the listed ones. I suggest considering the following: Meinzer et al. (1999); Kulmatiski et al. (2010); Kulmatiski, Beard (2013); Evaristo et al. (2016); Goldsmith et al. (2012); Liu et al. (2011); Bertrand et al. (2012); Meißner et al. (2012); Dawson (1996); Bijoor et al. (2012)

*The authors agree! However we had to make a choice and focus on the non-exhaustive list of papers that were reviewed here… Note that, on the other hand, another reviewer says that Table 1 is too long in its current form.*

Table 2: Why did you limit your analysis here to $\delta^{18}O$, while emphasizing that dual isotope approaches would be preferable on page 12 L4?

*In the text, we write that a dual isotopic framework is only interesting when oxygen and hydrogen stable isotopic composition profiles are disconnected from each other, i.e., when they are not linearly linked, which is the case in between rain events. Therefore with think that*

*a dual isotopic framework only adds value in the context of isotopic ($^{18}O$ and $^2H$) labelling pulses across the soil profile to artificially deconvoluate oxygen and hydrogen stable isotopic composition profiles (see, e.g., study of Bachmann et al., 2015).*

*As we wanted to compare the different approaches at natural isotopic abundance, we choose to consider one of the isotopologues (i.e., $H_2^{18}O$) only.*

Figure 1: Caption "negative towards the surface"

*Done*

Figure 2: Update the caption according to the color of the lines (blue).

*Done*

Figure 4: Caption "a detail is is presented for"

*Done*

Figure 5: Why standard deviation and standard error for the different approaches?

*Error bars for the RWU analytical model refer to the standard deviation associated with relative contributions to transpiration (x) across the 1000 model runs. For the TM approach, error bars are standard error of x as calculated with Equation (8b). This is now specified in the caption of the figure.*

Bachmann, D., Gockele, A., Ravenek, J. M., Roscher, C., Strecker, T., Weigelt, A., and Buchmann, N.: No evidence of complementary water use along a plant species richness gradient in temperate experimental grasslands, Plos One, 10, doi: 10.1371/journal.pone.0116367, 2015.

Dawson, T. E.: Hydraulic lift and water-use by plants - implications for water-balance, performance and plant-plant interactions, Oecologia, 95, 565-574, doi: 10.1007/BF00317442, 1993.

Farquhar, G. D., Cernusak, L. A., and Barnes, B.: Heavy water fractionation during transpiration, Plant Physiol., 143, 11-18, doi, 2007.

Gonfiantini, R.: Standards for stable isotope measurements in natural compounds, Nature, 271, 534-536, doi: 10.1038/271534a0, 1978.

Oerter, E. J., Perelet, A., Pardyjak, E., and Bowen, G.: Membrane inlet laser spectroscopy to measure H and O stable isotope compositions of soil and sediment pore water with high sample throughput, Rapid Commun Mass Spectrom, doi: 10.1002/rcm.7768, 2016. doi: 10.1002/rcm.7768, 2016.

Ogle, K., Wolpert, R. L., and Reynolds, J. F.: Reconstructing plant root area and water uptake profiles, Ecology, 85, 1967-1978, doi: 10.1890/03-0346, 2004.

Romero-Saltos, H., Sternberg Lda, S., Moreira, M. Z., and Nepstad, D. C.: Rainfall exclusion in an eastern Amazonian forest alters soil water movement and depth of water uptake, Am. J. Bot., 92, 443-455, doi: 10.3732/ajb.92.3.443, 2005.

Rothfuss, Y., Merz, S., Vanderborght, J., Hermes, N., Weuthen, A., Pohlmeier, A., Vereecken, H., and Brüggemann, N.: Long-term and high frequency non-destructive monitoring of water stable isotope profiles in an evaporating soil column, Hydrol. Earth Syst. Sci., 19, 4067-4080, doi: 10.5194/hessd-19-1-2015, 2015.

Sprenger, M., Volkmann, T. H. M., Blume, T., and Weiler, M.: Estimating flow and transport parameters in the unsaturated zone with pore water stable isotopes, Hydrol. Earth Syst. Sci. Discuss., 19, 2617–2635, doi: 10.5194/hess-19-2617-2015, 2015.

Wassenaar, L. I., Hendry, M. J., Chostner, V. L., and Lis, G. P.: High resolution pore water delta2H and delta18O measurements by H2O(liquid)-H2O(vapor) equilibration laser spectroscopy, Environ. Sci. Technol., 42, 9262-9267, doi, 2008.

Zarebanadkouki, M., Kim, Y. X., Moradi, A. B., Vogel, H. J., Kaestner, A., and Carminati, A.: Quantification and modeling of local root water uptake using neutron radiography and deuterated water, Vadose Zone J., 11, doi: 10.2136/vzj2011.0196, 2012.

---

## Author Comment (AC2) · 24 Nov 2016

General comments: The manuscript aims to compare different methods to locate root water uptake depth. I understand that it is a review paper, however, it is rather long, way too complex, and unfortunately hard to follow (it even comes with an appendix). This is mainly due to the many formulas that are presented and which disturb the text flow.

*The authors present only equations that are needed to understand all three methods (graphical inference, two to n end-member mixing models) plus the physically based approach of Couvreur et al. (2012). The appendix section was also meant in this way: not to disrupt the text flow. You also noted that sections 2.1 and 2.2 sections do not – on purpose – incorporate equations.*

I agree that a method comparison is needed but the manuscript is blown up with a lot of "basic isotope knowledge" which for my feeling is not necessary in such extent. I would suggest to reduce the length of the whole manuscript and focus on what differentiates the three methods to be compared.

*We will reduce sections 2.1 and 2.2 lengths significantly in a revised version of the manuscript.*

Further, the authors should only present equations which are really needed to understand the method comparison. I believe that this would increase the readability.

*Please see the answer to you first general comment.*

Concerning the presented figures, I would recommend to keep them simpler as they should generally be self-explaining and not as complex as they are now.

*Could you give some specifics? How the results were illustrated is directly inspired from the reviewed literature. Even though some figures might not be straightforward to all readers at first, the authors tried as much as possible to make them self-explaining by adding a thorough description in the caption.*

I would consider the manuscript ready for publication after major revision.

Specific comments:

Title: "quantify" instead of "quantifying"

*Done*

p. 3ff: Introduction needs a better/clearer structure

*The Introduction section is articulated thusly:*

*1- RWU definition;*
*2- RWU controlling variables and factors (e.g., concept of compensation and extreme case of Hydraulic redistribution);*
*3- Difficulty of measuring RWU;*
*4- Water stable isotopic compositions as a way to measure RWU;*
*5- Mention of objectives: review and comparison of the isotopic methods.*

*Where/what would you propose we change?*

p. 3 l. 2-6: References are missing

*Done (Javaux et al., 2013)*

p.3 l. 7: "driven by transpiration taking place. . ."

*Done*

p. 3 l.11: "spatial distribution. . .is very variable in time and space"; spatial in space ! avoid duplication

*Done*

p. 3 l. 13-14: Reference missing; permanent wilting point concept; what is a dry soil in this context?

*Sentence was rewritten:*

*"The flux of water depends also on soil water availability, i.e., the ability of the soil to provide water at the plant imposed rate (Couvreur et al., 2014): a highly conductive root segment will not be able to extract water from a dry soil."*

p. 5 l. 1: "each other" without hyphen

*Done*

p. 5 l. 3-6: Repetition from abstract

*This sentences state two objectives of the paper addressed in the section "challenges and progresses". This is why the authors mentioned it in both Abstract and Introduction sections. We do not feel this constitute repetition.*

p. 5 chp. 2.1: This chapter is too detailed; fundamentals of isotope hydrology do not have to be explained in such detail

*These paragraphs will be significantly shortened.*

p. 6 l. 20: Reference missing

*Done. (Sprenger et al., 2016)*

p. 7 l. 8: "grey" instead of "gray"

*Thanks (color wasn't actually grey but blue).*

p. 7 l. 12-14: Repetition

*Sentence was erased.*

p. 7 l. 24: Mention this earlier in the manuscript

*What exactly should we mention earlier?*

*If you are referring to "for plants growing in homogeneous external conditions, e.g., in hydroponic solution, root xylem sap water and external water have the same isotopic compositions", the authors think this is the right place to mention this.*

*If you are referring to "In natural soils where the liquid phase is not homogeneous and a vertical gradient of isotopic composition due to evaporation exists…", it is explained earlier in Section 2.2.*

p. 9 l. 16: 21 studies: Based on which criteria have these studies been selected? Literature review using ISI web of knowledge? Please mention briefly.

*This (non-exhaustive) list of publication was chosen according to, amongst other things, number of citation / retrospective contribution to field (publication year <2015) or novelty (publication year >= 2015). This will be specified.*

p. 9 l. 20: "unambiguously identified"! What about issues with regard to water extraction techniques which might be a cause for this?

*At this point of the demonstration, there is, for sake of clarity and concision, no question of techniques-related issues but rather the concept behind the graphical method (i.e., $\bar{z}$ is the depth where the soil water isotopic profile ($\delta_S$) equals that of the tiller water ($\delta_{Ti}$)). Extraction techniques are evocated later (section 5).*

p. 10 l. 7: grey ! correct throughout the manuscript

*Done*

p. 11 l. 26: Table 1 should rather go into the Introduction section, also it is too detailed

*The authors agree that Table 1 is too detailed and will be substantially simplified (e.g., less columns). However the authors feel it should remain in Section 3 as it illustrates the studies described in this very section.*

p. 12 l. 22: Replace www. by an abbreviation for example EPA, 2015; same for p. 13 l.

*The authors' intention was to provide the readers with direct access links to the zip files for each model (as this is generally done in the literature with, e.g., R packages).*

p. 13 l. 20ff: Why did the authors not intercompare the methods based on a dual isotope approach?

> *We opted for a single isotope ($^{18}O$) approach because we wanted to compare the different methods at natural isotopic abundance. As explained in the text a dual isotope approach only adds value if there is a disconnection between the oxygen and hydrogen stable isotopic composition profiles, which can be ideally realized following $^{18}O$ and $^{2}H$ labelling pulses in different portions of the soil profile.*

How reliable/meaningful is a single isotope approach?

> *The point of the model inter-comparison is to determine if using the different approaches is meaningful in the context of a single isotopic approach. Note that the vast majority of published studies use(d) a single isotopic approach. Reliability depends on whether or not uncertainty is properly accounted for, which we address as well.*

p. 14 l. 26: Is it necessary to mention the function?

> *Mention to the function was erased.*

p. 18: Think about renaming the subsection e.g. method uncertainties and. . .

> *Subsection will be split and renamed. Done*

p. 18 l. 1-20: This does not belong into the discussion section

> *Certainly. Please note that this section is not about discussing results but reviewing materials and isotopic techniques with focus on new developed techniques.*

p. 18 l. 21: Reference missing

> *Reference will be added ("e.g., Koeniger et al., 2011; West et al., 2006")*

p. 18 l. 21-24: Is this water plant available? Does it make sense to extract at such conditions if plant available soil water pools are of interest? Please discuss briefly.

> *Although this adsorbed water might not be directly available to plants it is in equilibrium with the bulk water and needs to be accounted for as a potential source in the root zone for modeling purposes.*

p. 18 l. 24ff: Methods are also not intercomparable and each method comes with a huge uncertainty (e.g. Sprenger et al., 2015; Orlowski et al., 2016). How reliable is such data in the end when utilized for RWU calculations? How would RWU depth vary if e.g. extraction method uncertainty is accounted for?

> *Uncertainty associated with extraction techniques are explicitly accounted for in each method and addressed in details in the manuscript:*
>
> - *through the uncertainty associated with measurement of $δ_{Ti}$ (width of the vertical band) [graphical method];*
> - *through $σ_{δTi}$ , $σ_{δs}$ etc [two end-members method];*
> - *through the parametrization of the approach of Phillips and Gregg (2001) (tolerance parameter) and Parnell et al. (2013) (sources ($δ_{S,J}$) and product ($δ_{Ti}$) uncertainties) [multi-sources mixing models]*
> - *through the sensitivity analysis [the model of Couvreur et al. (2010) was run a 1000 times, see Appendix B]*

p. 19 l. 3ff: Again, does this represent plant available water?

> *See answer to previous comment.*

p. 19 l. 15: Gaj et al. (2015) is not a method comparison paper.

> *The authors do not agree: Gaj et al. (2015) attempted to compare (i) on-line isotopic measurements, i.e., obtained non-destructively by sampling the soil atmosphere and analyzing with a WS-CRDS with (ii) off-line isotopic measurements, i.e., following destructive sampling and cryogenic vacuum extraction.*

Pratt et al. (2015) is wrong!

> *It is now Pratt et al. (2016). Thanks. Done*

Orlowski et al. (2016) and please cite Sprenger et al. (2015) as review paper about extraction method comparisons.

*Sprenger in now cited. What about Orlowski et al. (2016)? It is already cited.*

p. 19 l. 27: "generalization of coupled approaches" ! What does that mean?

The sentence was reformulated as such:

In order to fully benefit from the potential of water stable isotopologue analysis as tools for partitioning transpiration flux, the authors call for the development of a novel approach making use of physically based models for RWU and isotopic fractionation to analyze experimental data.

Table 1: Too much information

*Done (number of columns will be reduced)*

Table 3: Not sure if all these numbers are necessary to understand the method comparison/virtual experiments

*We propose to simplify Table 3 by removing the columns where absolute differences between the outcome of both methods are reported (numbers are actually already mentioned in the text). Furthermore, results will be rounded to the next whole number for readability.*

Couvreur, V., Vanderborght, J., Draye, X., and Javaux, M.: Dynamic aspects of soil water availability for isohydric plants: Focus on root hydraulic resistances, Water Resour. Res., 50, doi: 10.1002/2014WR015608, 2014.

Couvreur, V., Vanderborght, J., and Javaux, M.: A simple three-dimensional macroscopic root water uptake model based on the hydraulic architecture approach, Hydrol. Earth Syst. Sc., 16, 2957-2971, doi: 10.5194/hess-16-2957-2012, 2012.

Gaj, M., Beyer, M., Koeniger, P., Wanke, H., Hamutoko, J., and Himmelsbach, T.: In-situ unsaturated zone stable water isotope ($^2$H and $^{18}$O) measurements in semi-arid environments using tunable off-axis integrated cavity output spectroscopy, hydrol. Earth Syst. Sci., 20, 715-731, doi: 10.5194/hess-20-715-2016, 2015.

Javaux, M., Couvreur, V., Vander Borght, J., and Vereecken, H.: Root Water Uptake: From Three-Dimensional Biophysical Processes to Macroscopic Modeling Approaches, Vadose Zone J., 12, doi: DOI 10.2136/vzj2013.02.0042, 2013.

Orlowski, N., Breuer, L., and McDonnell, J. J.: Critical issues with cryogenic extraction of soil water for stable isotope analysis, Ecohydrology, 9, 3-10, doi: 10.1002/eco.1722, 2016.

Parnell, A. C., Phillips, D. L., Bearhop, S., Semmens, B. X., Ward, E. J., Moore, J. W., Jackson, A. L., Grey, J., Kelly, D. J., and Inger, R.: Bayesian stable isotope mixing models, Environmetrics, 24, 387–399, doi: 10.1002/env.2221, 2013.

Phillips, D. L. and Gregg, J. W.: Uncertainty in source partitioning using stable isotopes, Oecologia, 127, 171-179, doi: 10.1007/s004420000578, 2001.

Pratt, D. L., Lu, M., Barbour, S. L., and Hendry, M. J.: An evaluation of materials and methods for vapour measurement of the isotopic composition of pore water in deep, unsaturated zones, Isotopes Environ. Health Stud., doi: 10.1080/10256016.2016.1151423, 2016. doi: 10.1080/10256016.2016.1151423, 2016.

Sprenger, M., Leistert, H., Gimbel, K., and Weiler, M.: Illuminating hydrological processes at the soil-vegetation-atmosphere interface with water stable isotopes, Review of Geophysics, 54, 674-704, doi: 10.1002/2015RG000515, 2016.

---

## Referee Comment (RC3) · M. Sprenger (Referee) · 25 Nov 2016

Response by Matthias Sprenger to a Reply cited at the end of the document:

Yes, naturally occurring isotopic compositions of hydrogen and oxygen are linearly linked. However, as you point out in your section 2.1 and 2.2 of the manuscript, this linear relationship between the two isotopes can be different depending on having unfractionated soil water (slope of the regression line in the dual isotope space is about 8) or having fractionated soil water (slopes between 2 and 8). I think that this information as the deviation from the GMWL or LMWL (e.g., described as d-excess or lc-excess) would provide additional constrains on the root water uptake depth. I can imagine a situation, where there are two soil depths matching isotopically with the plant isotopes

for deuterium, but there is not a match between soil and plant water oxygen isotopes for the shallow soil depth, because the shallow depth experienced evaporation fractionation, while there is no evaporation signal in the deeper soils. Under that naturally occurring soil water isotope profile, a dual isotope approach would help delineating the root water uptake depth.

Therefore, I am currently not convinced that a dual isotope approach is only interesting when applying artificially isotopically enriched/depleted waters for root water uptake studies. Maybe this aspect of additional information due to the dual isotope relation (d-excess or lc-excess) is worth adding in the discussion. I am not asking to include that in the current modeling.

Comment by Matthias Sprenger during first round of revisions: "Why did you limit your analysis here to d18O, while emphasizing that dual isotope approaches would be preferable on page 12 L4?"

Reply by Youri Rothfuss: "In the text, we write that a dual isotopic framework is only interesting when oxygen and hydrogen stable isotopic composition profiles are disconnected from each other, i.e., when they are not linearly linked, which is the case in between rain events. Therefore with think that a dual isotopic framework only adds value in the context of isotopic (18O and 2H) labelling pulses across the soil profile to artificially deconvoluate oxygen and hydrogen stable isotopic composition profiles (see, e.g., study of Bachmann et al., 2015). As we wanted to compare the different approaches at natural isotopic abundance, we choose to consider one of the isotopologues (i.e., H218O) only"

---

## Referee Comment (RC4) · M. Sprenger (Referee) · 25 Nov 2016

Dear Youri and Mathieu,

Thanks for the answer and thus allowing for a discussion.

"when equilibrium for soil water is reached, one evaporation line is observed, i.e., d-excess is constant across the soil profile". Having all soil water isotope data on one evaporation line does not mean that there is a constant d-excess; actually the opposite would be the case. I guess you mean that one regression line can describe the evaporation line for the entire soil profile here.

I believe that your statement that soil water plots along the evaporation line (of constant

slope) is often limited to the upper soil. In your experiment (Rothfuss et al. 2015) the observations are limited to the upper 60 cm. However, I doubt that for thicker soil profiles, the slope of the soil water isotope samples will be constant across the profile. Soil water isotopes in the subsoil often plot along the LMWL. We have shown that for a study site in Luxembourg in Figure 2 and Figure 3b in Sprenger et al. (2016). That this trend towards higher d-excess (or lc-excess) (menaing less negative) with deeper depth occurs also for other study sites and climates is shown in Figure 5 in Sprenger et al. (2016). One can see there, that the lc-excess is not constant over depth, but approaches a more stable value with depth. Under such conditions, one could group the soil water isotope data into topsoil, plotting along an evaporation line, and subsoil, plotting along the LMWL (depending on the range of precipitation input). The simulations shown in Figure 9 in Sprenger et al. (2016) support that, since there is little variation in lc-excess below ca. 50 cm. Therefore, the soil water below 50 cm will plot along the LMWL, while the topsoil that experience evaporation will be plotting along an evaporation line. I hope I could clarify what I mean and that under the above described (natural occurring) conditions, a dual isotope approach would be beneficial.

References

Rothfuss, Y.; Merz, S.; Vanderborght, J.; Hermes, N.; Weuthen, A.; Pohlmeier, A. et al. (2015): Long-term and high-frequency non-destructive monitoring of water stable isotope profiles in an evaporating soil column. In Hydrol. Earth Syst. Sci. 19 (10), pp. 4067–4080. DOI: 10.5194/hess-19-4067-2015.

Sprenger, Matthias; Leistert, Hannes; Gimbel, Katharina; Weiler, Markus (2016): Illuminating hydrological processes at the soil-vegetation-atmosphere interface with water stable isotopes. In Rev. Geophys. 54 (3), pp. 674–704. DOI: 10.1002/2015RG000515.

---

## Author Comment (AC3) · 25 Nov 2016

Yes, naturally occurring isotopic compositions of hydrogen and oxygen are linearly linked. However, as you point out in your section 2.1 and 2.2 of the manuscript, this linear relationship between the two isotopes can be different depending on having un-fractionated soil water (slope of the regression line in the dual isotope space is about 8) or having fractionated soil water (slopes between 2 and 8). I think that this information as the deviation from the GMWL or LMWL (e.g., described as d-excess or lc-excess) would provide additional constrains on the root water uptake depth. I can imagine a situation, where there are two soil depths matching isotopically with the plant isotopes for deuterium, but there is not a match between soil and plant water oxygen isotopes for the shallow soil depth, because the shallow depth experienced evaporation fractionation, while there is no evaporation signal in the deeper soils. Under that naturally occurring soil water isotope profile, a dual isotope approach would help delineating the root water uptake depth.

Therefore, I am currently not convinced that a dual isotope approach is only interesting when applying artificially isotopically enriched/depleted waters for root water uptake studies. Maybe this aspect of additional information due to the dual isotope relation (d-excess or lc-excess) is worth adding in the discussion. I am not asking to include that in the current modeling.

> *Dear Matthias, thanks your comment,*
>
> *We agree with you that a partial reset of the isotopic profile due to, e.g., a rain occurrence, has for consequence to produce at first two regions (one upper and one lower) of distinct evaporation lines (i.e., distinct d-excess). Depending on soil boundary conditions and physical properties (in addition to the (self-)diffusion properties of water stable isotopologues), this d-excess vertical (if we stay in 1D) distribution will fade out until, when equilibrium for soil water is reached, one evaporation line is observed, i.e., d-excess is constant across the soil profile. This could be observed in the study Rothfuss et al. (2015) in the days following the "irrigation" event. Ideally, one would therefore have to measure "not too long" after the rain event to capture the d-excess spatial distribution for determination of the relative RWU profiles.*
>
> *In a revised version, we propose not to state that "[] a dual isotopic framework only adds value in the context of isotopic ($^{18}$O and $^{2}$H) labelling pulses across the soil profile to artificially deconvoluate oxygen and hydrogen stable isotopic composition profiles". Rather, the content of your comment and our answer to your comment will be added to the section 2.2 and to the discussion of the inter-comparison.*

Rothfuss, Y., Merz, S., Vanderborght, J., Hermes, N., Weuthen, A., Pohlmeier, A., Vereecken, H., and Brüggemann, N.: Long-term and high frequency non-destructive monitoring of water stable isotope profiles in an evaporating soil column, Hydrol. Earth Syst. Sci., 19, 4067-4080, doi: 10.5194/hessd-19-1-2015, 2015.

---

## Author Comment (AC4) · 25 Nov 2016

I believe that your statement that soil water plots along the evaporation line (of constant slope) is often limited to the upper soil. In your experiment (Rothfuss et al. 2015) the observations are limited to the upper 60 cm. However, I doubt that for thicker soil profiles, the slope of the soil water isotope samples will be constant across the profile. Soil water isotopes in the subsoil often plot along the LMWL. We have shown that for a study site in Luxembourg in Figure 2 and Figure 3b in Sprenger et al. (2016). That this trend towards higher d-excess (or lc-excess) (menaing less negative) with deeper depth occurs also for other study sites and climates is shown in Figure 5 in Sprenger et al. (2016). One can see there, that the lc-excess is not constant over depth, but approaches a more stable value with depth. Under such conditions, one could group the soil water isotope data into topsoil, plotting along an evaporation line, and subsoil, plotting along the LMWL (depending on the range of precipitation input). The simulations shown in Figure 9 in Sprenger et al. (2016) support that, since there is little variation in lc-excess below ca. 50 cm. Therefore, the soil water below 50 cm will plot along the LMWL, while the topsoil that experience evaporation will be plotting along an evaporation line. I hope I could clarify what I mean and that under the above described (natural occurring) conditions, a dual isotope approach would be beneficial.

*Dear Matthias,*

*Thanks for this. You made yourself very clear and we will follow your comment. It is true that our laboratory experiment (Rothfuss et al., 2015) was run on rather shallow soil profiles (height of the soil columns was 60 cm). We think that, in general, the determination of relative RWU profiles in the context of a dual isotopic approach under both natural and artificial isotopic abundance could be a nice follow-up contribution to the present manuscript. We can think of several scenarios differing this time in the nature ("position" on the LMWL) and magnitude of the rain event (natural isotopic abundances scenarios) or differing in the nature (e.g., location, isotopic composition values) and amount of labeling pulses (artificial isotopic abundances scenarios).*

Rothfuss, Y., Merz, S., Vanderborght, J., Hermes, N., Weuthen, A., Pohlmeier, A., Vereecken, H., and Brüggemann, N.: Long-term and high frequency non-destructive monitoring of water stable isotope profiles in an evaporating soil column, Hydrol. Earth Syst. Sci., 19, 4067-4080, doi: 10.5194/hess-19-4067-2015, 2015.

---

## Author Response (AR1)

General comments

5 The manuscript by Rothfuss and Javaux on quantifying root water uptake by the means of isotopic approaches aims to provide both an overview of methods and a comparison of the methods with regard to their limitations for the interpretation. They further propose to include modeling approaches to better estimate the root water uptake patterns.

The manuscript is generally well prepared, with mostly sufficient references, in depth information, and proper
10 visualizations. Due to its nature as a review, it is pretty long and I am not sure if the sections 2.1 and 2.2 are really necessary. I agree that it is necessary to understand the soil water isotopic composition in order to interpret the root water uptake with the means of stable isotopes. The authors focus on evaporation fractionation as one process to alter the soil water isotopic composition. From my point of view, also the precipitation input (and its variability in time) would then need to be considered. However, this has been reviewed recently and would blow
15 up the manuscript.

I think that the manuscript is a good contribution to current issues in ecohydrology and will be of interest to a broad readership. Therefore, I suggest a publication after a minor revision.

Specific comments

On page 5, where you introduce into the theoretical backgrounds, I do not think that the isotope depth profiles are
20 solely a result of fractionation effects. It seems that you miss the importance of the variability of the isotopic signal of the precipitation input and its consequences for the spatial variability of the soil water isotopes over depth.

> *Dear Matthias, we fully agree with this. A text was added (now P6 L4-6) on how soil upper-boundary isotopic condition (i.e., $\delta_{surf}$) - one of the determinants of the soil water isotopic composition profile – is highly impacted both spatially and temporally by input precipitation isotopic composition.*

25 As you write, the evaporation fractionation has been reviewed by Horita et al. (2008) and it was more recently reviewed by Soderberg et al. (2012). In order to streamline the manuscript and keep its focus on the root water uptake, I doubt that the section 2.1 and 2.2 are really necessary. However, I agree that it is necessary to account for the spatial and temporal variability of the soil water isotopes, but this is influenced by more than soil evaporation (see also my review Sprenger et al. (2016)).

30 > *Sub-section 2.1 was removed from the manuscript, sub-sections 2.2 and 2.3 were shortened, merged together, and renamed "Flow of isotopologues in the soil-plant-system". About the remark that "spatial and temporal variability is influenced by more than soil evaporation", please refer to our answer to your specific comment below.*

I am wondering if the authors are aware of the work by Ogle et al. (2004) and Ogle et al. (2014), where they
35 suggest to include biophysical conditions in a process-based mixing model ("root area profile and isotope deconvolution, RAPID"). I think it would be worth including this in the review of methods, since it goes in the same direction as the author's proposal of including more physical basis of the root water uptake estimates.

> *The authors are aware of the work of Ogle et al. (2004) where they could reconstruct "active root area"
> and RWU profiles from isotopic measurements (assuming, amongst other things, normal a priori
40 distributions for the xylem water oxygen and hydrogen isotopic compositions and considering prior knowledge on x). Since we wanted to focus on the most used and cited methods for partitioning plant water sources, we choose not to incorporate that of Ogle in the model comparison (and for that same reason neither that of Romero-Saltos et al. (2005)). Nevertheless, we added their work in Section 5*

*(Challenges and progresses), sub-section "Call for a coupled experiments-modelling approach for determination of plant water sources and redistribution on the basis of isotopic data" as it nicely fits there as you point out.*

Please state that the recently published dependency on the carrier gas was found for a WS-CRDS (Picarro); I did not find this $CO_2$ dependency for Off-axis ICOS (Los Gatos) (currently in review).

*Done. Thanks!*

Technical corrections

P1 L12/13: I suggest using "studies" rather than "authors"

*Done*

P3 L11: Insert "and" for "in space," and replace "but also on the root's" with "and their"

*Done*

P3 L25: Not sure what you mean with "by reference"

*We meant that the letter "S" makes reference to the "sink term". We will use the correct formulation "in reference to" instead of the incorrect "by reference to".*

P3 L32: I suggest "distribution of S" instead of "S distribution

*Done*

P3 L23: Introduce RLD here.

*Done (now P3 L23).*

P4 L21: Why not stating directly that Zarebanadkouki et al. (2012) used deuterated water?

*Done (now P4 L23)*

P4 L29: I suggest (or artificial enriched/depleted)

*Since "artificial" stands for "isotopic abundance", it cannot be followed by "enriched or depleted" but only "higher of lower" but then it reads a bit funny. We would like to keep the more general "artificial"*

P5 L1: Please state here once again what "these methods" will be.

*Done*

P5 L8: Is this not a bit too simplified at this point? You would always also need some kind of info about vegetation isotopes. I don't think the first sentence is necessary here.

*The authors write that for reconstruction of S profiles, one needs both soil and plant isotopic information ("within the soil-plant system").*

P5 L9: Are you referring to S profiles or isotope depth profiles? I do not think that the isotope depth profiles are solely a result of fractionation effects.

*We are referring to isotopic variations. It is implied here that these isotopic variations are solely due to the difference of physical properties of the different isotopologues (ultimately leading to isotopic fractionation) only in-between precipitation events, i.e., we rule out the partial to total reset of the isotopic profile due to each rain event.*

*We added the following sentence (now P6L4-6): "Finally, under natural conditions, the $\delta_S$ profile is not solely a result of isotopic fractionation but is as well highly impacted both spatially and temporally by input precipitation isotopic composition through modification of the upper boundary condition ($\delta_{surf}$)."*

P5 L 16: Consider splitting this long sentence.

*The entire sentence was removed from the current version of the manuscript*

P6 L 30: This line is not black anymore.

*Thanks! Done*

P7 L5: I think it would be worth including the definition by Barnes, Allison (1983) for the vapor region to be of total water potential about 15bar, while at lower potential there would be little connected water.

*Done (now P5 L22)*

P7 L8: This line is not grey anymore

*Thanks! Done*

P7 L15: Is this supported by data? I don't see it like that in Rothfuss et al. (2015), where you have shown that the slope for the depths above the EF (max -0.06 m) is still clearly below 8.

*We write that "As a result, an intermediate value for the slope is expected, depending on the mixing ratio of atmospheric water vapor to evaporated soil vapor at a given soil depth." (now P6 L3-4), meaning that the value for slope should lie between 2 and 8 (which is in accordance with Rothfuss et al. (2015)).*

P7 L16: I have shown that for several studies in Sprenger et al. (2016).

*Indeed. The list of three references has been simply replaced with your review "Sprenger et al. (2016)". (now P6 L3)*

P10 L8: delete second "is".

*Thanks! Done*

P10 L12: I summarized the uncertainty of the different methods to derive soil water isotope data in Sprenger et al. (2015b).

*The reference is cited now. Done (now P9 L2)*

P14 L25: I assume this is for $\delta^{18}O$? Please clarify.

*Yes. We added "($\delta_S$)" after "soil water oxygen isotopic composition" (as well as "(T)" after "actual transpiration rate") above in section 4.1.1 (now P12 L24)*

P14 L24: It would be interesting which rooting depth and density profile was assumed for the modeling. Please provide.

*This is explained (P12 L28-30):*

*"[All scenarios] relied on a common measured root length density vertical distribution of Festuca arundinacea",*

*and in Appendix B2 (Running the model for the inter-comparison, P45 L15-17):*

*"For this, $H_S$, $\delta_S$, and RLD input data were interpolated at a 0.01 m vertical resolution…"*

P16 L2: cases

*Done*

P17 L28: What about the study by Dawson (1993) who provides volumes?

*To the authors' understanding, Dawson (1993) provided a lateral gradient of the proportion of hydraulically-lifted water from maple trees used by neighboring plants, not HL water volumes.*

P18 L27: Why do you not include the direct-equilibration method by Wassenaar et al. (2008)?

*Done (now P17 L30)*

P19 L7: I did not use in-situ in that study, but consider including a recent paper by Oerter et al. (2016).

*Done (now P18 L12)*

P19 L14: Please not that this is the case for a WS-CRDS (Picarro); I did not find this $CO_2$ dependency for Off-axis ICOS (Los Gatos) (currently in review)

*Done (now P18 L19-20)*

P19 L25: I suggest referring to Farquhar et al. (2007).

*Done (now P18 L29)*

Table 1: I believe there are more studies than the listed ones. I suggest considering the following: Meinzer et al. (1999); Kulmatiski et al. (2010); Kulmatiski, Beard (2013); Evaristo et al. (2016); Goldsmith et al. (2012); Liu et al. (2011); Bertrand et al. (2012); Meißner et al. (2012); Dawson (1996); Bijoor et al. (2012)

*The authors agree! However we had to make a choice and focus on the non-exhaustive list of papers that were reviewed here… Note that, on the other hand, another reviewer says that Table 1 is too long in its current form.*

Table 2: Why did you limit your analysis here to $\delta^{18}O$, while emphasizing that dual isotope approaches would be preferable on page 12 L4?

*In the text (now P10 L24-28) we indeed underline the potential of the dual isotope approach:*

*"As infrared laser-based spectrometry now enables simultaneous measurements of $\delta^{18}O$ and $\delta^2H$ at lower cost, we believe that this dual-isotope approach (referred as "D" in Table 1) will or should gain in importance in isotopic studies. This is especially useful when (i) under natural conditions the $\delta^{18}O$-$\delta^2H$ slope is not constant over depth (Sprenger et al., 2016) or (ii) in the context of pulse labelling experiments, which can artificially change the value of the $\delta^{18}O$-$\delta^2H$ slope at given locations in the soil profile. In these cases, two independent mixing equations are obtained, one for each isotopologue."*

*As we wanted to concentrate on the different methods rather than on the consequences of the quality of the input data, we decided to go with a single isotope approach. But this would constitute, without a doubt, an interesting follow-up paper.*

*A justification of the choice of single isotope approach was added (Now P12 L16-18):*

*"The inter-comparison of models was performed using a single isotope ($^{18}O$) approach as the focus here was the differences of outcomes rather than the impact of the input isotopic data on these results."*

Figure 1: Caption "negative towards the surface"

*Done*

Figure 2: Update the caption according to the color of the lines (blue).

*Done*

Figure 4: Caption "a detail is is presented for"

*Done*

Figure 5: Why standard deviation and standard error for the different approaches?

*Error bars for the RWU analytical model refer to the standard deviation associated with relative contributions to transpiration (x) across the 1000 model runs. For the TM approach, error bars are standard error of x as calculated with Equation (8b). This is now specified in the caption of the figure.*
General comments: The manuscript aims to compare different methods to locate root water uptake depth. I understand that it is a review paper, however, it is rather long, way too complex, and unfortunately hard to follow (it even comes with an appendix). This is mainly due to the many formulas that are presented and which disturb the text flow.

*The authors present only equations that are needed to understand all three methods (graphical inference, two to n end-member mixing models) plus the physically based approach of Couvreur et al. (2012). The appendix section was also meant in this way: not to disrupt the text flow.*

I agree that a method comparison is needed but the manuscript is blown up with a lot of "basic isotope knowledge" which for my feeling is not necessary in such extent. I would suggest to reduce the length of the whole manuscript and focus on what differentiates the three methods to be compared.

*Sub-section 2.1 was removed from the manuscript, sub-sections 2.2 and 2.3 were shortened, merged together, and renamed "Flow of isotopologues in the soil-plant-system". About the remark that "spatial and temporal variability is influenced by more than soil evaporation", please refer to our answer to your specific comment below.*

Further, the authors should only present equations which are really needed to understand the method comparison. I believe that this would increase the readability.

*Please see the answer to you first general comment.*

Concerning the presented figures, I would recommend to keep them simpler as they should generally be self-explaining and not as complex as they are now.

*Could you give some specifics? How the results were illustrated is directly inspired from the reviewed literature. Even though some figures might not be straightforward to all readers at first, the authors tried as much as possible to make them self-explaining by adding a thorough description in the caption. Note that the colors in Figure 1 were changed (but not the data).*

I would consider the manuscript ready for publication after major revision.

Specific comments:

Title: "quantify" instead of "quantifying"

*Done*

p. 3ff: Introduction needs a better/clearer structure

*The Introduction section is articulated thusly:*

*1- RWU definition;*
*2- RWU controlling variables and factors (e.g., concept of compensation and extreme case of Hydraulic redistribution);*
*3- Difficulty of measuring RWU;*
*4- Water stable isotopic compositions as a way to measure RWU;*
*5- Mention of objectives: review and comparison of the isotopic methods.*

*Where/what would you propose we change?*

p. 3 l. 2-6: References are missing

*Done (Javaux et al., 2013) (now P3 L6)*

p.3 l. 7: "driven by transpiration taking place. . ."

*Done*

p. 3 l.11: "spatial distribution. . .is very variable in time and space"; spatial in space ! avoid duplication

*Done*

p. 3 l. 13-14: Reference missing; permanent wilting point concept; what is a dry soil in this context?

*Sentence was rewritten (now P3 L13-15):*

*"The flux of water depends also on soil water availability, i.e., the ability of the soil to provide water at the plant imposed rate (Couvreur et al., 2014): a highly conductive root segment will not be able to extract water from a dry soil."*

p. 5 l. 1: "each other" without hyphen

*Done*

p. 5 l. 3-6: Repetition from abstract

*This sentences state two objectives of the paper addressed in the section "challenges and progresses". This is why the authors mentioned it in both Abstract and Introduction sections. We do not feel this constitute repetition.*

p. 5 chp. 2.1: This chapter is too detailed; fundamentals of isotope hydrology do not have to be explained in such detail

*See answer to your second general comment*

p. 6 l. 20: Reference missing

*Text is no more part of the manuscript*

p. 7 l. 8: "grey" instead of "gray"

*Thanks (color wasn't actually grey but blue).*

p. 7 l. 12-14: Repetition

*Sentence was erased.*

p. 7 l. 24: Mention this earlier in the manuscript

*What exactly should we mention earlier?*

*If you are referring to "for plants growing in homogeneous external conditions, e.g., in hydroponic solution, root xylem sap water and external water have the same isotopic compositions", the authors think this is the right place to mention this.*

*If you are referring to "In natural soils where the liquid phase is not homogeneous and a vertical gradient of isotopic composition due to evaporation exists…", it is explained earlier (now P5ff).*

p. 9 l. 16: 21 studies: Based on which criteria have these studies been selected? Literature review using ISI web of knowledge? Please mention briefly.

*We explain this briefly in the text (now P8 L5-7):*

*"This non-exhaustive list was drawn up according to either the number of citations and contribution importance (for studies published before 2015) or to the novelty of the publications (publication year ≥ 2015)."*

p. 9 l. 20: "unambiguously identified"! What about issues with regard to water extraction techniques which might be a cause for this?

*At this point of the demonstration, there is, for sake of clarity and concision, no question of techniques-related issues but rather the concept behind the graphical method (i.e., $\bar{z}$ is the depth where the soil water isotopic profile ($\delta_S$) equals that of the tiller water ($\delta_{Ti}$)). Extraction techniques are evocated later (section 5).*

p. 10 l. 7: grey ! correct throughout the manuscript

*Done*

p. 11 l. 26: Table 1 should rather go into the Introduction section, also it is too detailed

*The authors agree that Table 1 was too detailed and is now simplified (e.g., less columns). However the authors feel it should remain in Section 3 as it illustrates the studies described in this very section.*

p. 12 l. 22: Replace www. by an abbreviation for example EPA, 2015; same for p. 13 l.

*The authors' intention was to provide the readers with direct access links to the zip files for each model (as this is generally done in the literature with, e.g., R packages).*

p. 13 l. 20ff: Why did the authors not intercompare the methods based on a dual isotope approach?

*As we wanted to concentrate on the different methods rather than on the consequences of the quality of the input data, we decided to go with a single isotope approach. But this would constitute, without a doubt, an interesting follow-up paper.*

*A justification of the choice of single isotope approach was added (Now P12 L16-18):*

*"The inter-comparison of models was performed using a single isotope ($^{18}$O) approach as the focus here was the differences of outcomes rather than the impact of the input isotopic data on these results."*

How reliable/meaningful is a single isotope approach?

*The point of the model inter-comparison is to determine if using the different approaches is meaningful in the context of a single isotopic approach. Note that the vast majority of published studies use(d) a single isotopic approach. Reliability depends on whether or not uncertainty is properly accounted for, which we address as well.*

p. 14 l. 26: Is it necessary to mention the function?

*Mention to the function was erased.*

p. 18: Think about renaming the subsection e.g. method uncertainties and. . .

*Subsection "5.2 High frequency isotopic data and sampling strategies" was split and renamed. We now have*

*"5.2 Soil and plant water sampling strategies" (P16 L27)*

*and*

*"5.3 Off-line destructive versus on-line non destructive isotopic measurements in plant and soil waters" (P17 L15)*

p. 18 l. 1-20: This does not belong into the discussion section

*Certainly. Please note that this section is not about discussing results but reviewing materials and isotopic techniques with focus on new developed developments.*

p. 18 l. 21: Reference missing

*Reference was added ("e.g., Koeniger et al., 2011; West et al., 2006") (now P17 L24-25)*

p. 18 l. 21-24: Is this water plant available? Does it make sense to extract at such conditions if plant available soil water pools are of interest? Please discuss briefly.

*Although this adsorbed water might not be directly available to plants it is in equilibrium with the bulk water and needs to be accounted for as a potential source in the root zone for modeling purposes.*

p. 18 l. 24ff: Methods are also not intercomparable and each method comes with a huge uncertainty (e.g. Sprenger et al., 2015; Orlowski et al., 2016). How reliable is such data in the end when utilized for RWU calculations? How would RWU depth vary if e.g. extraction method uncertainty is accounted for?

*Uncertainty associated with extraction techniques are explicitly accounted for in each method and addressed in details in the manuscript:*

- *through the uncertainty associated with measurement of $\delta_{Ti}$ (width of the vertical band) [graphical method];*
- *through $\sigma_{\delta Ti}$ , $\sigma_{\delta s}$ etc [two end-members method];*
- *through the parametrization of the approach of Phillips and Gregg (2001) (tolerance parameter) and Parnell et al. (2013) (sources ($\delta_{S,J}$) and product ($\delta_{Ti}$) uncertainties) [multi-sources mixing models]*
- *through the sensitivity analysis [the model of Couvreur et al. (2010) was run a 1000 times, see Appendix B]*

p. 19 l. 3ff: Again, does this represent plant available water?

See answer to previous comment.

p. 19 l. 15: Gaj et al. (2015) is not a method comparison paper.

*The authors do not agree: Gaj et al. (2015) attempted to compare (i) on-line isotopic measurements, i.e., obtained non-destructively by sampling the soil atmosphere and analyzing with a WS-CRDS with (ii) off-line isotopic measurements, i.e., following destructive sampling and cryogenic vacuum extraction.*

Pratt et al. (2015) is wrong!

*It is now Pratt et al. (2016). Thanks. Done*

Orlowski et al. (2016) and please cite Sprenger et al. (2015) as review paper about extraction method comparisons.

*Sprenger in now cited. What about Orlowski et al. (2016)? It is already cited.*

p. 19 l. 27: "generalization of coupled approaches" ! What does that mean?

*The sentence was reformulated as such (now P19 L2-6)*

*"In order to fully benefit from the potential of water stable isotopic analysis as tools for partitioning transpiration flux, the authors call for the development of approaches making use of physically based models for RWU and isotopic fractionation to analyze experimental data, especially since several soil-vegetation-atmosphere transfer (SVAT) models are available that can simulate flow of isotopologues in the soil and the plant"*

Table 1: Too much information

*Done: three columns were removed from the Table 1.*

Table 3: Not sure if all these numbers are necessary to understand the method comparison/virtual experiments

5 *Table 3 was simplified: we removed the columns where absolute differences between the outcome of both methods are reported (numbers are actually already mentioned in the text). Furthermore, results will be rounded to the next whole number for readability.*

Couvreur, V., Vanderborght, J., Draye, X., and Javaux, M.: Dynamic aspects of soil water availability for isohydric plants: Focus on root hydraulic resistances, Water Resour. Res., 50, doi: 10.1002/2014WR015608, 2014.

10 Couvreur, V., Vanderborght, J., and Javaux, M.: A simple three-dimensional macroscopic root water uptake model based on the hydraulic architecture approach, Hydrol. Earth Syst. Sc., 16, 2957-2971, doi: 10.5194/hess-16-2957-2012, 2012.

Dawson, T. E.: Hydraulic lift and water-use by plants - implications for water-balance, performance and plant-plant interactions, Oecologia, 95, 565-574, doi: 10.1007/BF00317442, 1993.

15 Farquhar, G. D., Cernusak, L. A., and Barnes, B.: Heavy water fractionation during transpiration, Plant Physiol., 143, 11-18, 2007.

Gaj, M., Beyer, M., Koeniger, P., Wanke, H., Hamutoko, J., and Himmelsbach, T.: In-situ unsaturated zone stable water isotope ($^2$H and $^{18}$O) measurements in semi-arid environments using tunable off-axis integrated cavity output spectroscopy, hydrol. Earth Syst. Sci., 20, 715-731, doi: 10.5194/hess-20-715-2016, 2015.

20 Javaux, M., Couvreur, V., Vander Borght, J., and Vereecken, H.: Root Water Uptake: From Three-Dimensional Biophysical Processes to Macroscopic Modeling Approaches, Vadose Zone J., 12, doi: DOI 10.2136/vzj2013.02.0042, 2013.

Oerter, E. J., Perelet, A., Pardyjak, E., and Bowen, G.: Membrane inlet laser spectroscopy to measure H and O stable isotope compositions of soil and sediment pore water with high sample throughput, Rapid Commun

25 Mass Spectrom, doi: 10.1002/rcm.7768, 2016. doi: 10.1002/rcm.7768, 2016.

Ogle, K., Wolpert, R. L., and Reynolds, J. F.: Reconstructing plant root area and water uptake profiles, Ecology, 85, 1967-1978, doi: 10.1890/03-0346, 2004.

Orlowski, N., Breuer, L., and McDonnell, J. J.: Critical issues with cryogenic extraction of soil water for stable isotope analysis, Ecohydrology, 9, 3-10, doi: 10.1002/eco.1722, 2016.

30 Parnell, A. C., Phillips, D. L., Bearhop, S., Semmens, B. X., Ward, E. J., Moore, J. W., Jackson, A. L., Grey, J., Kelly, D. J., and Inger, R.: Bayesian stable isotope mixing models, Environmetrics, 24, 387–399, doi: 10.1002/env.2221, 2013.

Phillips, D. L. and Gregg, J. W.: Uncertainty in source partitioning using stable isotopes, Oecologia, 127, 171-179, doi: 10.1007/s004420000578, 2001.

35 Pratt, D. L., Lu, M., Barbour, S. L., and Hendry, M. J.: An evaluation of materials and methods for vapour measurement of the isotopic composition of pore water in deep, unsaturated zones, Isotopes Environ. Health Stud., doi: 10.1080/10256016.2016.1151423, 2016. doi: 10.1080/10256016.2016.1151423, 2016.

Romero-Saltos, H., Sternberg Lda, S., Moreira, M. Z., and Nepstad, D. C.: Rainfall exclusion in an eastern Amazonian forest alters soil water movement and depth of water uptake, Am. J. Bot., 92, 443-455, doi:

40 10.3732/ajb.92.3.443, 2005.

Rothfuss, Y., Merz, S., Vanderborght, J., Hermes, N., Weuthen, A., Pohlmeier, A., Vereecken, H., and Brüggemann, N.: Long-term and high frequency non-destructive monitoring of water stable isotope profiles in an evaporating soil column, Hydrol. Earth Syst. Sci., 19, 4067-4080, doi: 10.5194/hess-19-4067-2015, 2015.

45 Sprenger, M., Volkmann, T. H. M., Blume, T., and Weiler, M.: Estimating flow and transport parameters in the unsaturated zone with pore water stable isotopes, Hydrol. Earth Syst. Sci. Discuss., 19, 2617–2635, doi: 10.5194/hess-19-2617-2015, 2015.

[revised manuscript text omitted]

---

## Referee Report (RR1)

**Review: *"Isotopic approaches to quantify root water uptake: a review and comparison of methods"* by Rothfuss and Javaux**

**General comments:**

The authors did a great job in editing the manuscript. The structure of the paper is much better now and easier to follow. However, the authors should check the tenses they use and be more concise.

After considering my following suggestions, I would consider the manuscript ready for publication.

**Specific comments:**

p. 1 l. 20ff: This sentence is rather confusing. Consider editing the sentence structure.

p. 2 l. 8: "…hormonal stress signals…"

l. 17ff: "…the root's ability to extract water…"; "…to fulfill the plant's water demand…"; "…between soil and roots."

l. 21: "…, and exudation."

p. 6 l. 4 ff: add the study by Martín-Gómez (2016) about evaporative enrichment of xylem water in woody stems to this paragraph

l. 15: "…fall on an "evaporative line…"; consider defining "evaporative line"

l. 17: Chose a more descriptive rather than a too general section title here.

p. 7 l. 10: "In a third class (…), …"

l. 14: "…in section 3.2 of this paper."

l. 18: wording

l. 31: "…in section 4 of this paper."

l. 32: wording

p. 8 l. 13: "In the example presented in Fig. 2b…"

l. 19: "…when RLD is constant over depth…"

l. 20: "The graphical inference method may not…"

p. 9 l. 2: Two end-member mixing models

l. 23-24: sentence structure: "…or by taking additional errors…and vacuum distillation into account."

l. 26: "…should have greatest possible isotopic dissimilarities with a low standard error of x."

p. 10 l. 13: "Figure 3 also illustrates…"

l. 19: between 0.3 and 0.4 meter depths

l. 21: "…in dual isotope space."

l. 20 ff: Several studies observed differences in terms of uptake depth when considering either the one or the other isotope. This is worth mentioning.

p. 11 l. 4: "…not likely or possibly…"

l.7: "…depends on the value of the contributing increment…"

l. 20: delete "by"…IsoSource

l. 23: "…and to (ii) provide a

l. 24: "For this study, they used…"

p. 12 l. 22: "The reader is referred to…"; "…to Appendix B2 on how it was implemented for our intercomparison."

p. 13 l. 3: "They"; who is "they"?

l. 4: Why was this plant chosen explicitly?

l. 19: "…were fixed to 10 and 0.25%,…"; also no space character between numbers and %; applies to the whole manuscript

l. 22: "Finally,…"

l. 24: "…methodology, the reader is referred to…"

l. 26: "simulated" -> repetition

l. 27: Couvreur et al. (2012)

p. 14 l. 2-3: "…term of Eq. (1) was proportionally larger (i.e., Scomp….)."; sentence structure

l. 6: "were" instead of " could be"

l. 8: "uppermost" layer (0–0.225m)

l. 21 ff: Wording/punctuation

p. 15 l. 10: "…via the analytical model"

l. 12: punctuation

l. 13: "However,…"; this applies to the whole manuscript

l. 29: "…should therefore be…"

p. 16 l. 1-2: "…for theoretically improving the different methods outputs…"

l. 3: "However, this was never the case when considering results of…"

l. 3 ff: wording; edit this sentence

l. 8-9: "…with an auger from few centimeters down to meters, …"

l. 13: You want to increase sampling intervals close to the soil surface: "minimal" is confusing here. Rewrite this and the previous sentence.

l. 17 ff: "…factor, thus, can be sampled multiple times to overcome or characterize…"

l. 22 ff: Confusing sentence structure. Chose a different word for "organ", maybe green plant parts

l. 24-25: What do you mean by "non-conducting" → water-bearing?

p. 17 l. 9: Why do you mention the azeotropic distillation method here? There are multiple other methods available. Which method was commonly used in your reviewed literature?

l. 20: add "in-situ" and "high-frequency"

l. 31: "…is not limited for isotope analysis."

l. 33: spectroscopes

p. 18 l. 1: "greatly sensitive"-> wording

l. 2: "…which was addressed by Pratt et al. (2016) and Orlowski et al. (2016b)." The study by Pratt et al. (2016) only applies to vapor measurements.

l. 6-7: "This should be further tested for other…"

l. 8 ff: Mention the study by Martín-Gómez (2016) here.

l. 20: deep-rooting

p. 19 l. 14: "For example, Rothfuss et al. (2012) ran an…"

l. 17: "…of the transpired water was…"

l. 19 ff: The authors are using "also" way too often. This applies to the entire manuscript.

p. 20 l. 6: "…and, thus,…"

l. 8: "…as they allow for a reduction of…"

l. 11 ff: The authors should not forget about the time and costs of such experiments.

p. 21 l. 5: "…is a powerful…"

l. 9 "In this review,…"

l. 10 ff: The authors should include a few sentences about the methodology before they jump into the description of the results.

l. 14: How was the performance? good/bad/great?; Parnell et al. (2010)

l. 18: "…call for (i) further developments…"

l. 19: replace "organs" by another word

---

## Author Response (AR2)

**Answers to comments from referee #3**

Here I am reviewing 'only' the revised version. The manuscript is basically split into two parts: 1. testing three different isotopic methods for determining root water uptake with artifical data, and 2. recommendations for sampling water isotopes.

I was expecting something very different reading the title: 1. redistribution was not quantified; 2. methods were not reviewed or compared for redistribution; and 3. there was no full review of all available methods using isotopes for root water uptake but rather a review on the three selected methods.

The title should hence be more specific.

I think the paper would gain a lot by concentrating on a single task, which would be the comparison of three methods to determine root water uptake. I would hence delete section 5 and the paragraph page 4, lines 6-9. This is corroborated that in the conclusions 17 lines talk about root water uptake and there are only three lines for redistribution. As written above, I also think that the manuscript is no review of methods. It describes three methods and applies them.

*The title was changed to: "Isotopic approaches to quantify root water uptake: a review and comparison of methods", i.e.,*

*1."redistribution" is no longer part of the title and no longer treated in the manuscript (however still mentioned as future challenge in section 5);*

*2."review" is still part of the title. The authors followed the comment made by the reviewer and associate editor and substantiate the manuscript with a proper literature review (see section 3 introductive paragraphs and supplementary material). From the literature review, it clearly appears that the three selected methods which are presented and compared are the most used ones (they account for about 96% of the published studies).*

The authors argue against the use of soil water content alone because it can be redistributed: "Rather, the full soil water flow equation accounting for root uptake and soil water redistribution must be solved in an inverse mode [...]". This should be true for isotopes as well and is not included in the presented methods.

*The authors agree with this statement that soil water isotopic redistribution must also be solved. However in the (graphical/statistical) methods presented in this study, this particular isotopic flow in the soil is considered to be negligible. This is now written P7L20: "One important feature of the three first classes of methods is that they consider soil water isotopic transport flow to be negligible for the duration of the experiment. Numerical models such as HYDRUS-1D and SiSPAT-Isotope on the other hand take this into account in the computation of RWU profiles."*

I have to admit that I did not fully follow the Couvreur et al. (2012) model but from what I got, isotopes are only diagnostic so if the model works for isotopes than it should work for water content as well.

It is not well described how the virtual experiments were produced. The delta-values in the tables 2 and 3 suddenly appear from nowhere. I have thought that the authors have at least two highly sophisticated models at hand that can produce isotope profiles in the soil. Why do I need Couvreur et al. (2012)?

*The model of Couvreur et al. (2012) provides physically based RWU distribution and therefore physically sound values of tiller isotopic composition to be fed to the graphical/statistical method. This is explained now P12L17-18: "Mean RWU depths (provided by the GI method) and $x_j$ distribution (provided by the two end-member and multi-source mixing models) were determined from soil and xylem water oxygen isotopic composition distributions. While the former information was prescribed to the different methods, the latter was calculated with the physically based analytical RWU model (referred to as "Couv") of Couvreur et al. (2012)"*

*The authors could have – indeed – used for instance SiSPAT-Isotope to produce the soil and tiller water isotopic input data for the other approaches. At the end, however, we would have played with boundary conditions and parameters until reaching similar scenarios. The idea was also, rather than using a complex numerical model, to provide for the readers means to calculate "by hand" RWU profiles.*

In the text, we now (P12L26) clearly mention how isotopic input data was produced: "We developed eight virtual plausible scenarios of soil-plant systems under different environmental conditions. This synthetic information was based on (i) experimental data and (ii) expert-knowledge.".

More specific comments are:

1.  Use less abbreviations in the text. Up to the end, I could not remember what meant DeDr_IT. Not very talkative. All the MSPG, MSPa, etc. makes the manuscript hard to read.

    *Use of abbreviations is now avoided as much as possible in the text. In addition, MSPG was replaced simply by "IsoS" (for "IsoSource") and MSPa by "SIAR". "AM" was also replaced by "Couv" for easier reference to the model of Couvreur et al. (2012). Done.*

2.  Figures 4 and 6 have no legends, which makes it hard to follow. I have to read the very, very, lengthy captions to find the meaning of a certain line. One line was only explained after 15 lines of caption.

    *Figures 4 and 6 have now descriptive legends. Caption length was significantly reduced. Done.*

3.  The figure captions include too much superfluous information that is or should be in the text.

    *Figures captions were significantly reduced. Done.*

4.  Figure 1 would benefit from the root and soil moisture profiles. In the current state, it does not really help me understanding anything.

    *RLD was added. Done.*

5. I would put only equations 2 and 4b. 3 is only a unit conversion and 4a is the same as 2.

    *This is true, however the authors report the series of equation in that order on purpose: for us it is important to (i) introduce the mixing equation in terms of absolute concentrations and not deltas and (ii) evocate the conversion problem 3D → 1D and state the implication when doing such a thing (e.g., lateral homogeneity is assumed etc.)*

6. page 7, line 24: this is wrong. delta=0 is an isotope ratio of V-SMOW so not nothing. Or put it like this $J_{Ti}$ must include S<0, otherwise you mass balance is not right.

    *You are right! This text part has been removed since it was about hydraulic redistribution which is not treated anymore.*

7. The passage about Zimmermann is too lengthy. Does not add much. I guess it is Barnes and Allison who first included the importance of vapour transport.

    *Barnes and Allison (1983) extended the (nevertheless straightforward) formulation of Zimmermann et al. (1967) to the situation with a dry surface layer. This is why the authors first talk about the study of Zimmermann et al. (1967), then this of Barnes and Allison (1983).*

8. page 3, line 25: I have read as well on Wikipedia that the Richards equation should be attributed first to Richardson. However, it is called Richards equations and not Richardson equation. Otherwise, nobody knows what you are talking about. And the review of Vereecken et al. (2016) is not mentioning it so remove the reference.

    *The reference of Richardson and Vereecken et al. (2016) were removed from the text.*

9. The authors cite themselves that the evaporative front is not at the point of maximal isotope value but at the point of maximal derivative. This is a hypothesis in their earlier paper but it was not shown theoretically. This could be done, probably, with one of their isotope in soil models

    *We thank you for this observation and keep the idea in mind!*

[revised manuscript text omitted]

---

## Author Response (AR4)

**Comment from the associate editor, Michael Bahn**

Dear authors,

both reviewers think that your manuscript has gained substantially and recommend its acceptance after another round of revisions. While reviewer #2 made a number of useful editorial suggestions, reviewer #3 thinks that section 5 and your conclusions require further focus. I agree with the reviewer that some streamlining should be useful here, but personally do not mind you keeping some key ideas conveyed in the sections 5.1-5.3 in a strongly condensed form.

> **Section 5.1 and 5.2 are now merged together and were significantly condensed. Section 5.3 was entirely removed from the manuscript.**

> **In an effort to streamline the paper even more, a novel figure (now Fig. 3) was added to illustrate the main findings of the literature review, thereby giving more weight to Section 3.**

Accordingly, I would be ok if you were to remove just the first point (i) of the last sentence in your conclusion.

> **We now only refer to new part 5.2 ("Call for a coupled experiments-modelling approach for determination of plant water sources on the basis of isotopic data") in the introduction (now P1 L19-20). Done**

Reviewer #3 also had some problems concerning your use of the model by Couvreur et al. 2012. Note that some further clarification to this end will make your paper also more accessible to its readers at large.

> **We hope that the use of the model of Couvreur et al. (2012) is now clearer to the associate editor and reviewer #3.**

**Comment from Anonymous referee #2,**

**General comments:**

The authors did a great job in editing the manuscript. The structure of the paper is much better now and easier to follow. However, the authors should check the tenses they use and be more concise. After considering my following suggestions, I would consider the manuscript ready for publication.

> **Dear referee, we *greatly* appreciate the tremendous effort in detecting the remaining typos in the document! The authors.**

**Specific comments:**

p. 1 l. 20ff: This sentence is rather confusing. Consider editing the sentence structure.

p. 2 l. 8: "…hormonal stress signals…"

> **Done**

l. 17ff: "…the root's ability to extract water…"; "…to fulfill the plant's water demand…"; "…between soil and roots."

> **Done**

l. 21: "…, and exudation."

**Done**

p. 6 l. 4 ff: add the study by Martín-Gómez (2016) about evaporative enrichment of xylem water in woody stems to this paragraph

**This (very nice) study does not study fractionation *during* RWU, only *after* RWU has taken place.**

l. 15: "…fall on an "evaporative line…"; consider defining "evaporative line"

**Done**

l. 17: Chose a more descriptive rather than a too general section title here.

**We would like to keep the title "LITERATURE REVIEW" as it echoes the manuscript first part of the title, i.e., "Isotopic approaches to quantify root water uptake: a *review* and comparison of methods". Similarly, the title of part 4 "INTER-COMPARISON OF METHODS" refers to the second part of the title.**

p. 7 l. 10: "In a third class (…)**,** …"

**Done**

l. 14: "…in section 3.2 of this paper."

**Done**

l. 18: wording

**Done**

l. 31: "…in section 4 of this paper."

**Done**

l. 32: wording

**Done**

p. 8 l. 13: "In the example presented in Fig. 2b…"

**Done**

l. 19: "…when RLD is constant over depth…"

**Done**

l. 20: "The graphical inference method may not…"

**Done**

p. 9 l. 2: Two end-member mixing models

**We would like to keep the acronym in the title as we think it will be convenient for the readers, right?**

l. 23-24: sentence structure: "…or by taking additional errors…and vacuum distillation into account."

**Done**

l. 26: "…should have greatest possible isotopic dissimilarities with a low standard error of x."

**Done. Thanks!**

p. 10 l. 13: "Figure 3 also illustrates…"

**Done**

l. 19: between 0.3 and 0.4 meter depths

**Done. Thanks**

l. 21: "…in dual isotope space."

**Done**

5 l. 20 ff: Several studies observed differences in terms of uptake depth when considering either the one or the other isotope. This is worth mentioning.

**Yes, but, to our knowledge (see supplementary materials), differences where observed when considering either $\delta^2H$ or $\delta^{18}O$ measurements by authors who used other methods (i.e., not the two end-member mixing model). This is mentioned early in the text (Section 3 introduction, now P7 L30-33):**

10        **"In the remaining studies, both isotopic compositions were measured and used to provide two separate estimates of relative contribution distributions even though $\delta^2H$ or $\delta^{18}O$ distributions were strongly linked (see section 2). This last approach is in the present study referred as "double single" (see supplementary materials)."**

p. 11 l. 4: "…not likely or possibly…"

15        **Done**

l.7: "…depends on the value of the contributing increment…"

**Done**

l. 20: delete "by"…IsoSource

**Done**

20 l. 23: "…and to (ii) provide a

**Done**

l. 24: "For this study**,** they used…"

**Done**

p. 12 l. 22: "The reader is referred to…"; "…to Appendix B2 on how it was implemented for our

25 intercomparison."

**Done**

p. 13 l. 3: "They"; who is "they"?

**Done ("All eight scenarios relied on …", now P13 L22)**

l. 4: Why was this plant chosen explicitly?

30        **This plant was chosen because we had access to information on $K_{plant}$ of use for modeling purposes.**

l. 19: "…were fixed to 10 and 0.25%,…"; also no space character between numbers and %; applies to the whole manuscript

**Done**

l. 22: "Finally**,**…"

**Done**

35 l. 24: "…methodology, the reader is referred to…"

**Done**

l. 26: "simulated" -> repetition

**Thanks ! Done**

l. 27: Couvreur et al. (2012)

5 **Done**

p. 14 l. 2-3: "…term of Eq. (1) was proportionally larger (i.e., Scomp….)."; sentence structure

**Done. Sentence was split in two (now P14 L22-25).**

l. 6: "were" instead of " could be"

**Done**

10 l. 8: "uppermost" layer (0–0.225m)

**Done**

l. 21 ff: Wording/punctuation

**Done**

p. 15 l. 10: "…via the analytical model"

15 **We would prefer to keep the conjunction "by"**

l. 12: punctuation

**Done. Sentence was rearranged.**

l. 13: "However,…"; this applies to the whole manuscript

**Done**

20 l. 29: "…should therefore be…"

**Done**

p. 16 l. 1-2: "…for theoretically improving the different methods outputs…"

**The theoretical basis underlying the different methods is not impacted by the quality of the data, only their precision.**

25 l. 3: "However, this was never the case when considering results of…"

**Done**

l. 3 ff: wording; edit this sentence

**Done**

l. 8-9: "…with an auger from few centimeters down to meters, ..."

30 **Done**

l. 13: You want to increase sampling intervals close to the soil surface: "minimal" is confusing here. Rewrite this and the previous sentence.

**Done. "interval" was replaced with "resolution", therefore "minimal" is now "maximal", and vice versa. (now P17 L5-8)**

l. 17 ff: "…factor, thus**,** can be sampled multiple times to overcome or characterize…"

> **Done**

l. 22 ff: Confusing sentence structure. Chose a different word for "organ", maybe green plant parts

> **Done**

5   l. 24-25: What do you mean by "non-conducting" ☐ water-bearing?

> **Sentence was erased from the text for streamlining. (see our answer to referee #3)**

p. 17 l. 9: Why do you mention the azeotropic distillation method here? There are multiple other methods available. Which method was commonly used in your reviewed literature?

> **Sentence was erased from the text for streamlining. (see our answer to referee #3)**

10   l. 20: add "in-situ" and "high-frequency"

> **Done**

l. 31: "…is not limited for isotope analysis."

> **Done**

l. 33: spectroscopes

15   > **We would like to stick to "spectrometers" here.**

p. 18 l. 1: "greatly sensitive"-> wording

> **We use now "significantly sensitive". Done**

l. 2: "…which was addressed by Pratt et al. (2016) and Orlowski et al. (2016b)." The study by Pratt et al. (2016) only applies to vapor measurements.

20   > **Done**

l. 6-7: "This should be further tested for other…"

> **Done**

l. 8 ff: Mention the study by Martín-Gómez (2016) here.

> **The authors could not find this particular study.**

25   l. 20: deep-rooting

> **We would like to stick to "deep-rooted" here.**

p. 19 l. 14: "For example, Rothfuss et al. (2012) ran an…"

> **Done**

l. 17: "…of the transpired water was…"

30   > **Done**

l. 19 ff: The authors are using "also" way too often. This applies to the entire manuscript.

> **Done. "also" is now written 13 times (instead of 25 initially).**

p. 20 l. 6: "…and, thus,…"

**Done**

l. 8: "…as they allow for a reduction of…"

**Done**

l. 11 ff: The authors should not forget about the time and costs of such experiments.

5 **We do not. Done**

p. 21 l. 5: "…is a powerful…"

**Done and thanks!!**

l. 9 "In this review**,**…"

**Done**

10 l. 10 ff: The authors should include a few sentences about the methodology before they jump into the description of the results.

**Done**

l. 14: How was the performance? good/bad/great?; Parnell et al. (2010)

**Done**

15 l. 18: "…call for (i) further developments…"

**Done**

l. 19: replace "organs" by another word

**Done**

**Comment from Anonymous referee #3,**

The manuscript has improved tremendously. Well done.

> **Thanks!**

There are basically two items, which I would ask to consider:

5   1.I would delete sections 5.1-5.3. They are not adding to the manuscript. This would also delete the last paragraph of the conclusions and the last sentence of the abstract.

> **Section 5.1 and 5.2 are now merged together and were significantly condensed. Section 5.3 was entirely removed from the manuscript. We now only refer to new part 5.2 ("Call for a coupled experiments-modelling approach for determination of plant water sources on the basis of isotopic data") in the introduction.**

> **The last sentence of the abstract was replaced with (now P1 L19-20):**

>> **"Finally, the authors call for a development of approaches coupling physically based RWU models with controlled conditions experimental setups."**

If experimental precision were an issue than it should be treated in the results section with a virtual experiment, just as the two disjoint soil layers. Otherwise this is just the usual muttering that we need higher resolution observations.

> **This is why the authors not only mention this problem but emphasize the need to take into account the measurement uncertainty (through the different standard errors coefficients present in Eqs. 8) when using the two-end members mixing model. This is rarely the case from the literature review.**

Section 5.4, however, is interesting and not enough elaborated. If a model were inverted, one needs to identify quite a few extra variables such as hydraulic conductivities, etc. The information content in the data is probably not enough to infer all required information.

> **This is true. Note that the authors mention this problem in the text already (now P19 L9-10):**

>> **"We recognize that in comparison with the statistical and conceptual methodologies presented in this review, using a physical (analytical or numerical) model implies the measurements of additional state variables to be fed as input to the model, and of one parameter (Kplant) (when considering the assumption Kplant = Kcomp valid, see Appendix B). Some of these variables are laborious to obtain (e.g., RLD) or not straightforward to measure (HS, HL, and T) – especially in the field – but are mandatory to be able to determine contributions to T across a set of identified water sources."**

So the method, which quite a few people would find best, might not be superior to the Bayesian approaches presented here. It is unclear to what extent this plays into the presented methods. For example, the preferred Bayesian inversions depend on strongly on the priors and hence to some assumed soil properties. Did the authors try different priors, e.g. flat priors, Jeffreys prior, etc.?

**We also argue in the text that the Bayesian method is very efficient in determining relative RWU profiles, while it needs less input data than the, e.g., approach of Couvreur et al. (2012).**

**Regarding prior information, the authors now mention this in text (now P16 L25-28):**

> **"Note that no prior information on the relative contribution to $T$ from the different soil layers was used when running the SIAR program, i.e., the authors opted for flat priors. This can be changed by the user, based on additional collected data such as, for instance, information of root architecture and function across the soil profile or information on soil hydraulic properties and water status."**

2. I still do not understand the use of Couvreur et al. (2012). These are virtual experiments so running a full isotope-enabled soil model gives all the information for the virtual experiments.

**Paragraph 4.1.1 ("Scenario definition") was substantially rewritten for clarifying the point (P13 L10-27):**

> **"We developed eight virtual plausible scenarios of soil-plant systems under different environmental conditions. For each scenario, we set one total soil water potential ($H_S$) profile and one soil water oxygen isotopic composition ($\delta_S$) profile. These profiles resulted from the combination of a lower boundary condition, i.e., the depth of the groundwater table, and an upper boundary condition, i.e., the soil surface water status. The groundwater table (of water isotopic composition equal to -7‰) was either shallow at -1.25 m depth (prefix "Sh") or deep at -6 m depth (prefix "De"). The soil water potential was considered to be at static equilibrium below the groundwater level. The soil surface was either dry under evaporative conditions (suffix "Dr"), or wet, e.g., shortly after a rain event (suffix "We"). For instance for the scenario "ShDr", we set the δS profile to be maximal at the surface, due to evaporation, and minimal from –0.5 m downward, due to the shallow groundwater table location. For the scenario "DeWe", on the other hand, the increase of $\delta_S$ towards the surface was not monotonic due to a recent precipitation event (of water isotopic composition equal to -7‰). Finally, we tested two different values of plant transpiration rate (T) and leaf water potential ($H_L$) with each of these four combinations (i.e., ShDr, ShWe, DeDr, and DeWe). The transpiration rate was either low (e.g., relevant at night, $T = 0.01$ mm h-1, suffix "_lT") or high (T = 0.30 mm h-1, suffix "_hT"). All eight scenarios relied on a common measured root length density vertical distribution of**

**Festuca arundinacea. Table 2 reports the input data. Note that, as hypothetized in Eq. (4b), transpiration and sap flow rates (i.e., per unit of surface area [L T-1]) were considered as equal.**

**The objective was not to use an advanced numerical model such as, e.g., SiSPAT-Isotope or Soil-litter-iso, to produce these scenarios, but rather use synthetic information based on (i) experimental data and (ii) expert-knowledge which would ideally illustrate the performances or limitations of the different methods."**

**Plus, the idea was to encourage the readers to use much simpler, however still complex RWU models, such as this of Couvreur et al. (2012).**

Here, I have to believe that Couvreur et al. (2012) gives the right answer. Is this so?

**The model of Couvreur et al. (2012) gives a physically-sound answer. We assume that it gives the right answer, indeed.**

**The following clarification was added to the text (Introduction of section 4, now P12-13 L29-4):**

**"It has been proved (Couvreur et al., 2012) that this model gives similar results than a 3D physically based model with detailed descriptions of the root architecture and of the water flow in soil and roots. In that sense, this is the best current model existing nowadays to simulate water fluxes in a soil-plant system (based on biophysical considerations). Other current models make assumptions or use empirical relations to predict RWU, which are not based on bio-physical considerations only (Jarvis, 2011; Simunek and Hopmans, 2009). Obviously, we do not mean that the model of Couvreur et al. (2012) gives the reality but rather the best estimate of the water flow based on our physical knowledge."**

Or I do not understand the method. It is NOT well described: "Mean RWU depths (provided by the GI method) and xj distribution (provided by the two end-member and multi-source mixing models) were determined from soil and xylem water oxygen isotopic composition distributions. While the former information was prescribed to the different methods, the latter was

calculated with the physically based analytical RWU model (referred to as "Couv") of Couvreur et al. (2012)."

How is GI entering in this? Where are the "soil and xylem water oxygen isotopic composition distributions" coming from? What is "the former" and what "the latter"? What are you referring to?

**This sentence was reformulated (indeed, this was clear enough!) as such (now P12 L25-28):**

**"Mean RWU depths (provided by the GI method) and xj distribution (provided by the two end-member and multi-source mixing models) were determined from the $\delta_S$ profile and the $\delta_{Ti}$ value. For**

each virtual experiment the $\delta_S$ profile was prescribed to the different methods while $\delta_{Ti}$ was calculated with the physically based analytical RWU model (referred to as "Couv") of Couvreur et al. (2012).".

**We hope it is now clearer.**

How I understood virtual experiments: you run the most advanced physical model and add some noise on the output. Then you use this noisy output to infer properties, states or fluxes, which were calculated in the physical model. It seems to be different here.

**This is not what we have been doing because the objective was not to find the best method but rather to compare outputs of methods. Therefore we did not need to have a "real" or "simulated" case with the "most advanced physical model". Please see for this our answer above. Doing the virtual experiment you propose is a good objective for a next study, when a novel 3D physical model for root water uptake, soil water flow and stable water isotope fate will exist (what you call "most advanced physical model").**

Minor comments:

a) page 1, lines 17-20: I cannot understand the sentence. Please reformulate.

**The sentence now reads (now P1 L16-17):**

> **"The benchmarking of these methods illustrates the limitations of the graphical and statistical methods while it underlines the performance of one Bayesian mixing model."**

b) page 4, 20-23: theory says something different and Rothfuss et al. (2015) only proposed that it might be the point of the steepest gradient. Please reformulate.

**Text now reads (P4 L20-23):**

> **"Finally, note that Rothfuss et al. (2015) argued that, at transient state ($\delta_E \neq \delta_{source}$), the maximal isotopic enrichment in the soil profile might not point to the location of the evaporation front. Instead, they proposed that the depth where the steepest gradient in the isotopic profile is observed corresponds to the evaporation front."**

**Done**

c) page 9, 8: i should be superscript and A, B subscript.

**Thanks! Done**

d) page 14, 31: you call Figure 6 still Figure 5 from this point on.

**Thanks! Done**

e)  page 15, 32: 8 layers were not shown but only mentioned. If you claim that it might be a good way forward then it might warrant a figure or including it in Fig. 6.

**The results of the Bayesian and analytical modelling methods are shown for eight soil layers in Table 3. Nevertheless, it is true that the methods of Phillips and Gregg (2003) and this of Parnell et al. (2013) are not compared in the text for eight soil layers. The text was therefore modified accordingly (now P16 L22-25):**

> **"As highlighted in this series of virtual experiments, the Bayesian method showed for the case of two and three soil layers much more convincing results than the method of Phillips and Gregg (2003). The Bayesian method was particularly efficient in the case of eight soil layers, illustrating the interest of reaching the best vertical resolution and maximizing the number of identified potential sources (Table 3)."**

**Associate Editor Decision: Publish subject to technical corrections (04 Apr 2017) by Michael Bahn**

**Comments to the Author:**

Dear Drs. Rothfuss and Javaux,

I am satisfied with your revisions and accept your manuscript for publication in Biogeosciences. I think that this will be a very useful contribution to the field.

> **Thanks!  We also hope so.**

Please make the following improvements before uploading the final version of your manuscript:

* In your final statement of the conclusion you restrict the relevance of the approach to the fields of agronomy and agriculture, which appears justified especially when targeting RWU as a function of plant genotype. However, it would be useful to conclude with a broader perspective addressing the potential of the approach for understanding and quantifying RWU also in seminatural and natural ecosystems.

> **Now the final statement reads:**

> **"This type of approach could be used in agronomy to quantify RWU as a function of plant genotype and soil structure. It also has great potential to quantifying RWU in seminatural and natural ecosystems for understanding the mechanisms underlying the vegetation feedbacks to the atmosphere in the contexts of land cover and climate changes."**

* The legend of Fig. 3 should be self-explaining, so please add information on what precisely was searched and where. In the Figure, panels d) to f) could all be similarly sized.

> **Figure3 panels d) to f) are now similarly sized and the legend caption reads:**

[revised manuscript text omitted]